



# Surface [Urban] Energy and Water Balance Scheme (v2020a) in non-urban areas: developments, parameters and performance

Hamidreza Omidvar[1,✉], Ting Sun[1], Sue Grimmond[1], Dave Bilesbach[2], Andrew Black[3], Jiquan Chen[4], Zexia Duan[5], Zhiqiu Gao[5,6], Hiroki Iwata [7], Joseph P. McFadden[8]

[1] Department of Meteorology, University of Reading, Reading, RG6 6BB, UK
[2] Biological Systems Engineering Department, University of Nebraska, Lincoln, NE, 68588, USA
[3] Faculty of Land and Food System, University of British Columbia, Vancouver, BC, V6T 1Z4, CA
[4] Center for Global Change and Earth Observation, Department of Geography, Michigan State University, East Lansing, MI, 48824, USA
[5] Collaborative Innovation Centre on Forecast and Evaluation of Meteorological Disasters, School of Atmospheric Physics, Nanjing University of Information Science and Technology, Nanjing, 210044, China
[6] State Key Laboratory of Atmospheric Boundary Layer Physics and Atmospheric Chemistry, Institute of Atmospheric Physics, Chinese Academy of Sciences, Beijing, 100029, China
[7] Department of Environmental Science, Faculty of Science, Shinshu University, Nagano 390-8621, Japan
[8] Department of Geography, University of California, Santa Barbara, CA, 93106 USA

✉ *Correspondence to*: *h.omidvar@reading.ac.uk; c.s.grimmond@reading.ac.uk*

**ORCID IDs:**

Hamidreza Omidvar: *https://orcid.org/0000-0001-8124-7264*
Ting Sun: *https://orcid.org/0000-0002-2486-6146*
Sue Grimmond: *https://orcid.org/0000-0002-3166-9415*
Dave Bilesbach: *https://orcid.org/0000-0001-8661-9178*
Andrew Black: *https://orcid.org/0000-0001-9292-1146*
Jiquan Chen: *https://orcid.org/0000-0003-0761-9458*
Zexia Duan: *https://orcid.org/0000-0003-2822-7066*
Zhiqiu Gao: *https://orcid.org/0000-0001-8256-005X*
Hiroki Iwata: *https://orcid.org/0000-0002-8962-8982*
Joseph P. McFadden: *https://orcid.org/0000-0002-5869-7774*





***Abstract***
This paper extends the applicability of the SUEWS (Surface [Urban] Energy and Water Balance Scheme)
to extensive pervious areas (deciduous trees, evergreen trees, grass, croplands, soil and water) outside
cities. It can be used either offline or online (i.e., coupled to weather/climate models). The required
parameters to simulate the turbulent latent heat (or evaporative) flux are derived using observations. Both
the parameters (leaf area index (LAI), albedo, roughness parameters and surface conductance) and the
surface energy balance fluxes are evaluated at independent sites and/or different periods at the same
site. Methods to obtain parameters and guidance to apply SUEWS are provided. Results demonstrate the
impacts from differences in LAI dynamics and albedo for various types of vegetation. The relation
between LAI and albedo is explored. Deciduous, evergreen, and grass land covers all have long periods
of LAI maxima, but croplands normally have a short sharp peak due to harvesting. For most of the
vegetation types studied the maximum albedo coincides with the maximum LAI period, but for some
evergreen trees the maxima are associated with leaves changing colour (needles/leaves get darker as
they age during autumn and winter). Ensuring these dynamics are captured is important for assessing
urban-rural differences (e.g. canopy layer air temperature).
*Keywords*: SUEWS, pervious land cover, leaf area index, albedo, evaporation flux, roughness parameters
**1    Introduction**
Key to advancing our knowledge of planetary boundary layer behaviour is understanding
surface-atmosphere interactions. Various land surface models (LSM) simulate these energy and
water exchanges (Ek et al., 2003; Levis et al., 2004; Krinner et al., 2005; Kowalczyk et al.,
2006). 'Urban' land use is amongst the most diverse (e.g. high-rise central business district to
one-storey single family residential areas) with many land-cover types (e.g. paved roads,
buildings, parks with trees and grass) influencing energy and water surface-atmosphere
exchange through a wide range of complex biophysical processes. The complexity of urban
systems have grown substantially with urbanization (United Nations 2018). A number of LSMs
have been designed for urban areas (Grimmond et al., 2010), to capture processes such as
heat and water released by anthropogenic activities (Grimmond et al., 1986; Grimmond, 1992;
Masson, 2000; Kusaka et al., 2001; Martilli et al., 2002).
The Surface [Urban] Energy and Water Balance Scheme (SUEWS, Grimmond *et al.,* 1986,
1991, Grimmond & Oke 1991, Järvi *et al.,* 2011) characterises the heterogeneity of urban
surfaces using seven land covers split between impervious (buildings, paved) and pervious


(evergreen trees/shrubs, deciduous trees/shrubs, grass, soil, water) types. SUEWS has been
evaluated in multiple cities globally (e.g. Karsisto *et al.,* 2016,  Ward *et al.,* 2016,  Ao *et al.,*
2018, Kokkonen *et al.,* 2018, Harshan *et al.,* 2018) with varying mixes of integrated impervious-
pervious land covers. However, when extensive areas of one type of pervious land cover (e.g.
deciduous trees) occurs (e.g. in rural areas) some parameters are expected to differ from
integrated-urban values (i.e. obtained for built-up areas). Most notably, there will be differences
in parameters that are associated with the surface resistances for latent heat flux calculations
because of differences in sub-grid-scale advection processes (Spronken-Smith et al., 2000).
Thus, new parameters need to be determined from observations.
Our objective is to bridge this gap by deriving values for several latent heat flux related
parameters (*viz*, leaf area index (LAI), albedo, roughness parameters and surface resistance)
for extensive non-urban pervious areas and assess their seasonal variability. This improves
SUEWS regional applicability with rural areas with forests, farms, and grasslands (*etc*). For
reproducibility and applicability to other data sets parameter derivation is implemented in Python
Jupyter notebooks (Omidvar et al., 2020). The SUEWS model (Sect. 2.1, Appendix A) is used
with observations (Sect. 2.2) from numerous sites. Methods address both obtaining the
parameters and their evaluation (Sect. 2.3). The derived parameters (Sect. 3) are evaluated
(Sect. 4), allowing conclusions to be drawn (Sect. 5).
## 2    Methods
### 2.1    SUEWS and its vegetation-related sub-models
The details of how SUEWS computes the surface energy, water and carbon fluxes are given in
Järvi *et al.* (2011), Ward *et al.* (2016), and Järvi *et al.* (2019). The surface energy and water
balances are directly linked by the turbulent latent heat flux ($Q_E$) or its mass equivalent
evaporation (*E*):
$$Q^* + Q_F = Q_H + Q_E + \Delta Q_S \tag{1}$$
$$P + I_e = E + R + \Delta S \tag{2}$$
where $Q^*$ is the net all-wave radiation flux, $Q_F$ is the anthropogenic heat flux, $Q_H$ is the turbulent
sensible heat flux, $\Delta Q_S$ the net storage heat flux, and $P$, $I_e$, $\Delta S$ and $R$ are precipitation, external
water use, net change in the canopy water storage and runoff, respectively. As we focus on
extensive (non-urban) pervious areas the anthropogenic heat flux ($Q_F$) is assumed to be 0 W
m$^{-2}$.





Vegetation phenology changes key model parameters, most notably, leaf area index (*LAI*). Leaf-
out and senescence impact the albedo ($\alpha$) and therefore surface radiative exchanges. LAI
changes also modify both aerodynamic roughness parameters (roughness length ($z_0$), zero
plane displacement height ($z_d$)) (e.g. Kent *et al.,* 2017) and surface resistance ($r_s$) . The former
impacts aerodynamic resistance ($r_a$) while $r_s$ directly moderates $Q_E$ (Sect. 2.1.4).
Model parameters need to be internally consistent for land cover type *i.* This allows different
types of vegetation (e.g. a crop) to be simulated*.* All the parameters needed for a vegetated
surface and those addressed in this paper are given in Table 1. SUEWS allows parameters to
vary between individual grids (Järvi *et al.,* 2019, Sun *et al.,* 2020) and thus can represent a high
degree of spatial heterogeneity (e.g. different heights of trees).
Table 1: Parameters that SUEWS uses (and can be set) for pervious surface types by first
associated process (i.e. most impact multiple variables). Those determined (D) in this
study (*) and the values used (given in Table: T#, Sect.: S#) in individual equations (E).

| Category | Symbol | Definition | Value | E | D |
|---|---|---|---|---|---|
| *Radiation* | $\alpha_{LAI_{\min}}$ | Albedo at $LAI_{min}$ | T4 | 6 | * |
| | $\alpha_{LAI_{\max}}$ | Albedo at $LAI_{max}$ | T4 | 6 | * |
| | $\varepsilon_0$ | Emissivity | T2 | | |
| *Leaf Area Index (LAI)* | $LAI_{min}$ | LAI Minimum | T4 | 4,5 | * |
| | $LAI_{max}$ | LAI Maximum | T4 | 4.5 | * |
| | $T_{BaseSDD}$ | Base temperature senescence degree days (*SDD)* | T4 | 4 | * |
| | $T_{BaseGDD}$ | Base temperature for growing degree days | T4 | 4 | * |
| | $GDD_v$ | GDD from the start of the crop vegetative phase | T4 | 5 | * |
| | $GDD_{LAI_{max}}$ | Growing degree days until $LAI_{max}$ | T4 | 5 | * |
| *Roughness* | $H_v$ | Vegetation height | T3 | | |
| | $z_{0m}$ | Roughness length for momentum | S2.1.4 | 9 | * |
| | $z_d$ | Zero plane displacement | S2.1.4 | 9 | * |
| *Surface resistance* | G2-G6 | Coefficients | T5 | 12 | * |
| | $G_{max}$ | Coefficients | T5 | 12 | * |
| | $T_H, T_L$ | Temperature limits for switching off evaporation | S2.1.4 | 15 | |
| | $s_1$ | Coefficient related to wilting point | S2.1.4 | 16 | |
| | $K_{\downarrow,\max}$ | Maximum observed incoming shortwave | S2.1.4 | 13 | |
| *Storage heat flux* | $a_1$-$a_3$ | Coefficient for storage heat flux | T2 | 7 | * |
| *Water storage* | $S_i$ | Canopy water storage capacity | T2 | 19 | |

**2.1.1   Leaf Area Index (*LAI*)**
In SUEWS, $LAI$ for the current day ($d$) is calculated using cumulative growing degree days
($GDD$) and senescence degree days ($SDD$) of the previous day ($d - 1$) for vegetation type *i.* For
forests and grass we use (Järvi *et al.,* 2011):

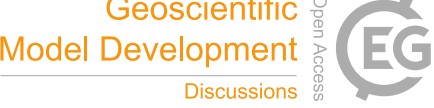


$$LAI_{d,i} = \begin{cases} \min(LAI_{\max,i}, LAI_{d-1,i}^{\omega_1} GDD\ \omega_2 + LAI_{d-1,i}), & T_{BaseSDD} < T_d < T_{BaseGDD} \\ \max\big(LAI_{\min,i}, LAI_{d-1,i}^{\omega_1} SDD\ \omega_2 + LAI_{d-1,i}\big), & T_{BaseGDD} < T_d < T_{BaseSDD} \end{cases} \tag{3}$$

with $\omega_1 = 30 \times 10^{-3}$ and $\omega_2 = 0.5 \times 10^{-3}$. The base temperatures associated with the initiation
of leaf-on ($T_{BaseGDD}$) and leaf-off ($T_{BaseSDD}$, units °C) periods are used relative to a mean air
temperature $T_d$ derived from the daily maximum ($T_a^{\max}$) and minimum ($T_a^{\min}$) for the current day:

$$T_d = \frac{T_a^{\max} + T_a^{\min}}{2} \tag{4}$$

The model requires the maximum and minimum *LAI* values ($LAI_{max,i}$, $LAI_{min,i}$) for each
vegetation type. Eq. 3 has fewer calibration parameters than Eq. A1 of Järvi *et al. (*2014) as
$T_{BaseGDD}$ and $T_{BaseSDD}$ are determined for each site (Sect. 2.3). If $T_{BaseGDD}$ and $T_{BaseSDD}$ are
available for a site, one should account for day-length and photoperiod for more northerly sites
(Bauerle et al., 2012; Gill et al., 2015).
For crops (e.g. rice, wheat) *LAI* also depends on the planting date. However, as crops are
grown to be harvested, the period of *LAI_max* is short (cf. e.g. forests as parametrised in Eq. 3).
We propose:

$$LAI_{d,crop} = \begin{cases} \min\left(LAI_{\max}, \dfrac{LAI_{max} - LAI_{min}}{GDD_{LAI_{max}} - GDD_v}(GDD_p - GDD_v) + LAI_{\min}\right) & GDD_p \leq GDD_{LAI_{max}} \\ \max\left(LAI_{\min}, -\dfrac{LAI_{max} - LAI_{min}}{GDD_{LAI_{max}} - GDD_v}(GDD_p - GDD_v) + 2LAI_{max} - LAI_{\min}\right) & GDD_p > GDD_{LAI_{max}} \end{cases} \tag{5}$$

where $GDD_p$ is the *GDD* accumulated from the day of planting; $GDD_{LAI_{max}}$ is associated with
*LAI_max* and $GDD_v$ is the start of crop vegetative phase. Note that $GDD$, $SDD$, and $GDD_p$ (in Eq. 4
and 5) change with time from their base temperatures ($T_{BaseGDD}$ for $GDD$, $T_{BaseSDD}$ for $SDD$, 0 °C
for $GDD_d$). Using a different base temperature (than 0 °C) to calculate $GDD_d$, $GDD_v$, and
$GDD_{LAI_{max}}$ in Eq. 5 leads to same $LAI_{d,crop}$ results as only the difference values $GDD_{LAI_{max}} -$
$GDD_v$ and $GDD_p - GDD_v$ are important. Here, these crop specific coefficients are obtained for
rice and winter wheat (Sect. 2.3).
**2.1.2 Albedo ($\alpha$)**
In SUEWS, the albedo varies with daily *LAI* between the minimum ($\alpha_{LAI_{\min}}$) and maximum
($\alpha_{LAI_{\max}}$) by vegetation type:

$$\alpha_{d,i} = \alpha_{d-1,i} + \big(\alpha_{LAI_{\max,i}} - \alpha_{LAI_{\min,i}}\big)\frac{LAI_{d,i} - LAI_{d-1,i}}{LAI_{\max,i} - LAI_{\min,i}} \tag{6}$$



The maximum albedo does not necessarily occur with the maximum *LAI* because of change in
leaf/needle colour (Sect. 3.1). Here we focus on snow-free conditions, albeit a snow module is
available in SUEWS (Järvi *et al.,* 2014). Bare soil and water albedo are assumed to be constant
in a model run (Sect. 3.2).
The observed (30 min) incoming and outgoing shortwave radiation are used to calculate each
albedo from 10:00 to 14:00 (local standard time). From this, one mean albedo for each day is
calculated. The two model parameters (Eq. 6, Table 1) are selected from those that minimize
the mean absolute error (MAE, Sect. 2.4) of the albedo prediction at a calibration site.
Within SUEWS the albedo is used with the observed incoming shortwave radiation to obtain $Q^*$.
In the current analyses, the observed incoming longwave ($L_\downarrow$) and modelled outgoing longwave
radiation ($L_\uparrow = (1 - \varepsilon_0)L_\downarrow + \varepsilon_0 \sigma T_s^4$ where $\varepsilon_0$ is the surface emissivity, $\sigma$ is the Stefan Boltzmann
constant (W m$^{-2}$ K$^{-4}$), and $T_s$ is the surface temperature (K), Appendix A.1) are used. Table 2
gives the emissivity values used.
To determine $\alpha_{LAI_{\min}}$ and $\alpha_{LAI_{\max}}$ for each individual vegetated site (excluding snow) we analyse
observational data for snow free periods. Although SUEWS has a snow option, this option is
disabled in all runs to verify "no snow" scenarios. We assume precipitation is snow if $T_a < 0$ °C
(Järvi *et al.,* 2014), and that snow remains until the 5-day moving average of air temperature is
above 5 °C. Although this method will not flag all the snow-covered days (e.g. duration of snow
cover also depends on snow depth), it provides a rough estimate of when the albedo is affected
by snow.
*Table 2: Pervious surface OHM storage heat flux (a₁, a₂, a₃) coefficients are derived (this study*
*O20, methods: Sect. 2.1.3), except for tree and grass areas which are derived from literature*
*sources (D85= Doll et al. (1985), M85= McCaughey (1985)) by Grimmond et al. (1991) and*
*Grimmond and Oke (1999)); canopy water storage (Sᵢ, Eq. 19) from B03= Breuer et al.*
*(2003), and emissivity from sources in W16 - Ward et al. (2016).*

| Vegetation Type | $a_1$ | $a_2$ | $a_3$ | Source | $S_i$ (mm) | Source | Emissivity | Sources in |
|---|---|---|---|---|---|---|---|---|
| Deciduous trees/shrubs | 0.215 | 0.325 | -19.9 | M85 | 1.3 | B03 | 0.98 | W16 |
| Evergreen trees/shrubs | 0.215 | 0.325 | -19.9 | M85 | 0.8 | B03 | 0.98 | W16 |
| Grass | 0.215 | 0.325 | -19.9 | D85 | 1.9 | B03 | 0.93 | W16 |
| Rice | 0.185 | 0.615 | -18.0 | O20 | 1.9 | B03 | 0.95 | Water |
| Wheat | 0.283 | 0.784 | -18.0 | O20 | 1.9 | B03 | 0.93 | Grass |
| Soil | 0.210 | 0.902 | -20.4 | O20 | 1.9 | B03 | 0.93 | W16 |
| Water | 0.880 | 0.370 | -85.4 | O20 | - | - | 0.95 | W16 |



### 2.1.3 Storage heat flux ($\Delta Q_S$)

Storage heat flux is simulated with the objective hysteresis model (OHM, Grimmond *et al.,* 1991):

$$\Delta Q_S = \sum_i f_i \left[ a_{1,i} Q^* + a_{2,i} \frac{\partial Q^*}{\partial t} + a_{3,i} \right] \tag{7}$$

where $f_i$ is the plan area (or 3d, Grimmond et al. 1991, Grimmond and Oke 1999) fraction of surface $i$ and $a_{1-3}$ are the OHM coefficients (Table 2). To obtain $a_{1-3}$ from observations of $Q^*$ and $\Delta Q_S$ (as the residual of Eq. 1, in extensive pervious sites $Q_F$ = 0 W m$^{-2}$) regression is used. As the sites are assumed to be extensively the same pervious land cover type $f_i$ = 1 in each case. We determine one set of OHM coefficients per site, hence assuming they are constant and ignoring soil wetness effects and other variations.

### 2.1.4 Latent heat flux ($Q_E$)

In SUEWS, a modified Penman-Monteith equation (Penman, 1948; Monteith, 1965) is used to compute $Q_E$ with $Q_F$ = 0 W m$^{-2}$ in non-urban areas (e.g. this paper) and greater than zero for cities (Grimmond & Oke 1991):

$$Q_E = \frac{s(Q^* + Q_F - \Delta Q_S) + \frac{\rho c_p V}{r_a}}{s + \gamma \left( 1 + \frac{r_s}{r_a} \right)} \tag{8}$$

The atmospheric state is obtained from the slope of saturation vapour pressure curve with respect to temperature ($s$, units: Pa K$^{-1}$), density of air ($\rho$, kg m$^{-3}$), specific heat of air at constant pressure ($c_p$, J K$^{-1}$ kg$^{-1}$), vapour pressure deficit ($V,$ Pa), psychrometric 'constant' ($\gamma$,: Pa K$^{-1}$), and the aerodynamic resistance for water vapour ($r_a$, units: s m$^{-1}$). The latter is obtained from Ulden & Holtslag (1985) and Järvi *et al.* (2011):

$$r_a = \frac{\left[ \ln \left( \frac{z_m - z_d}{z_{0m}} - \psi_m(\zeta) \right) \right] \left[ \ln \left( \frac{z_m - z_d}{z_{0v}} - \psi_v(\zeta) \right) \right]}{\kappa^2 u}, \tag{9}$$

where $z_m$ is the measurement height for mean wind speed ($u)$ and $\kappa$ the von Kármán constant (0.4 assumed); the aerodynamic parameters $z_d$ (zero plane displacement height) and $z_{0m}$ (roughness length for the momentum) are estimated as a function of canopy height which varies for different *LAI* states of each surface, as discussed in Appendix B (Garratt, 1994; Grimmond and Oke, 1999). For water and soil surfaces they are estimated to be $z_{0m}$ = 0.0005 m and 0.002 m respectively with $z_d$ = 0 m (Moene and van Dam, 2013). Canopy height for the different surface types is given in Table 3. The stability scale $\zeta$ ($= (z_m - z_d)/L$) depends on $L$ the Obukhov length. SUEWS is modified (Appendix A) so that for completely pervious surfaces the





roughness length for vapour ($z_{0v}$) is calculated as $z_{0v} = 0.1z_{0m}$ (Brutsaert, 1982) and assumed to
be the same as for sensible heat. The atmospheric stability functions of momentum ($\psi_m$) and
water vapour ($\psi_v$) for unstable condition are (Campbell and Norman, 1998):
$$\psi_v = 2 \ln\left[\frac{1 + (1 - 16\zeta)^{1/2}}{2}\right],$$
$$\psi_m = 0.6\psi_v$$
(10)

and for stable condition (Campbell and Norman, 1998; Högström, 1988):
$$\psi_v = -4.5 \ln(1 + \zeta)$$
$$\psi_m = -6 \ln(1 + \zeta)$$
(11)

For completely wet surfaces, the surface resistance ($r_s$) is assumed to be 0 s m$^{-1}$ (i.e. potential
evaporation is calculated from Eq. 8). Otherwise $r_s$, or its inverse surface conductance ($g_s$), is
modelled (Ward *et al.,* 2016):
$$r_s^{-1} = g_s = \sum_i \left(g_{\max,i} f_i\right) g(LAI_i) g(K_\downarrow) g(\Delta q) g(T_a) g(\Delta\theta_{soil}).$$
(12)

To reduce the number of coefficients in Ward *et al.*'s (2016), $G_1$ (their Eq. 9) is removed from the
first term (of Eq. 12) leaving $g_{\max,i}$ (maximum surface conductance, units: m s$^{-1}$) and $f_i$. For
'homogeneous' sites (Sect. 2.2) $f_i$ =1. Phenological state is critical: $g(LAI_i) = \frac{LAI_i}{LAI_{\max,i}}$. For bare
soil surfaces (i.e. no vegetation), when *LAI* is irrelevant $g(LAI_i) = 1$. The remaining terms are
related to meteorology (incoming shortwave radiation $K_\downarrow$, specific humidity deficit $\Delta q$, air
temperature $T_a$), and soil moisture deficit ($\Delta\theta_{soil}$, difference between soil moisture and soil water
capacity); using Grimmond & Oke (1991), Järvi *et al. (*2011), and Ward *et al. (*2016):
$$g(K_\downarrow) = \frac{\frac{K_\downarrow}{G_2 + K_\downarrow}}{\frac{K_{\downarrow,\max}}{G2 + K_{\downarrow,\max}}}$$
(13)

where $K_{\downarrow,\max}$ is the maximum observed incoming shortwave radiation (= 1200 W m$^{-2}$);
$$g(\Delta q) = G_3 + (1 - G_3)G_4^{\Delta q}$$
(14)

$$g(T_{air}) = \frac{(T_{air} - T_L)(T_H - T_a)^{T_c}}{(G_5 - T_L)(T_H - G_5)^{T_c}}$$
(15)

where $T_c = \frac{T_H - G_5}{G_5 - T_L}$ is a function of the lower ($T_L = -20$ °C) and upper ($T_H$ = 55 °C) limits that
determine when the evaporation switches off in SUEWS. Here we extended $T_L$ from $-10$ °C
(from Ward *et al.,* 2016) to $-20$ °C to ensure that the temperature limit covers all climates (Table
3) studied here. Note $Q_E$ is negligible (Appendix C) when $T_a < -20$ °C.



The soil moisture control considers the wilting point ($\Delta\theta_{WP} = \frac{s_1}{G_6}$, with $s_1 = 5.56$, see Järvi *et al.*
2011) using $G_6$ to vary with soil and plant type:
$$g(\Delta\theta_{soil}) = \frac{1 - \exp(G_6(\Delta\theta_{soil} - \Delta\theta_{WP}))}{1 - \exp(-G_6\Delta\theta_{WP})} \tag{16}$$

To obtain the $G_2\ to\ G_6$ and $g_{max}$, a so-called 'observed' $g_s$ is obtained by rearranging Eq. 8,
when the surface is dry (and both $Q_H$ and $Q_E$ are > 0 W m$^{-2}$):
$$\frac{1}{g_s} = r_s = \left[\frac{s}{\gamma}\frac{Q_H}{Q_E} - 1\right]r_a + \frac{\rho c_p V}{\gamma Q_E}. \tag{17}$$

The $g_s$ related parameters (Eq. 12) are obtained using non-linear regression with the observed
values (Eq. 17). We use a Python package Platypus (Hadka, 2015) with a multi-objective
evolutionary algorithm (Zhou et al., 2011) so that we capture: (1) *variations of $g_s$*: difference
between standard deviation of $g_s$ from model and observations (normalized by standard
deviation of observations); and (2) *magnitude of $g_s$*: mean absolute difference between $g_s$ from
model and observations.
SUEWS has a running water balance that accounts for the multiple surface types. The amount
of water on the canopy of each surface ($C_i$) (Grimmond & Oke 1991) is used to vary the surface
resistance between dry and wet ($r_s = 0$ s m$^{-1}$) by replacing $r_s$ with $r_{ss}$ (Shuttleworth 1978):
$$r_{ss} = \left[\frac{W}{r_b(s/\gamma + 1)} + \frac{(1 - W)}{r_s + r_b(s/\gamma + 1)}\right]^{-1} - r_b(s/\gamma + 1), \tag{18}$$

where *W* is a function of the relative amount of water present on each surface to its water
storage capacity ($S_i$, Table 2):
$$\begin{aligned} W &= 1 & C_i \geq S_i \\ W &= \frac{K - 1}{K - S_i/C_i} & C_i < S_i \end{aligned} \tag{19}$$

*K* depends on the aerodynamic and surface resistances:
$$K = \frac{(r_s/r_a)/(r_a - r_b)}{r_s + r_b(s/\gamma + 1)}, \tag{20}$$

where $r_b$, the boundary layer resistance, is a function of friction velocity $u_*$ (Shuttleworth 1983):
$$r_b = 1.1u_*^{-1} + 5.6u_*^{\frac{1}{3}}. \tag{21}$$

Equations 18-21 ensure that the surface resistance $r_{ss}$ has a smooth transition from 0 (a
completely wet surface) to $r_s$ (a dry surface).



## 2.2 Observations and sites


To determine parameters for non-urban surfaces (Table 1), and to evaluate their performance,
observations from "homogeneous" sites with long term radiation fluxes and eddy covariance
measurements are required (Table 3). The 30 min meteorological observations used are air
temperature, incoming shortwave radiation, upwelling shortwave radiation, station pressure,
relative humidity, wind speed, precipitation, net all-wave radiation, sensible heat flux and
evaporation flux. The precipitation data are used to select dry periods and are required by
SUEWS to calculate $\Delta\theta_{soil}$ and the surface state ($C_i$).
The site land cover characteristics are provided by their key references (Table 3). The observed
LAI data are from the NASA Moderate Resolution Imaging Spectroradiometer (MODIS,
Nishihama *et al.,* 1997) four-day composite product MCD15A3H (Myneni *et al.,* 2015) with 500
m resolution.
The sites (Fig. 1) in North America are part of the AmeriFlux network (Baldocchi *et al.,* 2001)
and two of the Asian sites are part of AsiaFlux (AsiaFlux, data access: 2020-01-22). Seven land
cover types are analysed (Table 3) using one to three sites:
(1) Deciduous trees (DBF): three sites
(2) Evergreen trees (ENF): three sites
(3) Grass (GRA): two sites
(4) Rice (RIC): two sites
(5) Winter wheat (WHT): one site
(6) Water (WAT): one continuous and two intermittent flood irrigated rice sites (Table 3)
(7) Bare soil (BSV): two sites (up to four weeks after rice planting).
Irrigation ($I_e$, Eq. 2) modifies both the soil moisture deficit and surface state and is critical for the
growth of many plants. Notably rice has flood irrigation for a period when a site-specific depth
(Table 3) is maintained. At CN-DNT (Table 3) this occurs until 5 weeks before harvest, whereas
at PH-IRI there are only 2 weeks without irrigation. The CN-DNT wheat field is kept saturated
but not flooded for the entire time (Duan *et al.*, 2020). To account for this, sufficient water is
added by SUEWS to satisfy these conditions (Appendix A.2 gives details).
*Table 3: Analysed pervious land cover types (DBF: Deciduous Broadleaf Forests, ENF: Evergreen*
*   Needleleaf Forests, GRA: Grasslands, CRP: crops, BSV: bare soil, WAT: Water) at different sites and*
*   periods. Key references and DOI provide the details of the observations and each site. The sites*
*   elevation (elev) above sea level (asl), vegetation height (Hᵥ) above ground level (agl) and height of*


wind speed measurement ($H_U$). Sites are in Canada (CA), China (CN), Japan (JP), Philippines (PH)
and USA (US). At CN-DNT both rice (RIC) and wheat (WHT) are grown.

| Site | Name | Type | Mean Temp. (°C)[1] | Elev. (m asl) | $H_v$ (m agl) | $H_U$ (m agl) | Lat. (°N) | Lon. (°) | Calibration year | Test years | DOI | Key Reference |
|------|------|------|------|------|------|------|------|------|------|------|------|------|
| US-MMS | Morgan Monroe State Forest | DBF | 13.2 | 275.0 | 25.0 | 46.0 | 39.32 | −86.41 | 2017 | 2010,2012, 2016 | 10.17190/AMF/1246080 | Schmid *et al.* (2000) |
| US-UMB | Univ. Michigan Biological Station | DBF | 7.1 | 234.0 | 20.0 | 46.0 | 45.56 | −84.71 | 2008 | 2010,2014, 2016 | 10.17190/AMF/1246107 | Curtis *et al.* (2002) |
| US-Oho | Oak Openings | DBF | 11.0 | 230.0 | 24.0 | 34.0 | 41.55 | −83.84 | 2010 | 2011,2012, 2013 | 10.17190/AMF/1246089 | Noormets *et al.* (2008) |
| CA-Obs | Saskatchewan - Western Boreal, Mature Black Spruce | ENF | 1.3 | 628.9 | 7.2 | 26.0 | 53.99 | −105.12 | 2008 | 2003,2005, 2006 | 10.17190/AMF/1375198 | Bergeron *et al.* (2007) |
| CA-Qcu | Quebec - Eastern Boreal, Black Spruce /Jack Pine Cutover | ENF | 1.6 | 392.3 | 13.8 | 24.0 | 49.27 | −74.04 | 2010 | 2005,2008, 2009 | 10.17190/AMF/1246828 | Bergeron *et al.* (2007) |
| US-Blk | Black Hills | ENF | 6.6 | 1718.0 | 13.0[2] | 24.0 | 44.16 | −103.65 | 2005 | 2004,2006, 2008 | 10.17190/AMF/1246031 | - |
| US-KUT | KUOM Turfgrass Field | GRA | 8.0 | 301.0 | 0.07 | 1.35 | 44.99 | −93.19 | 2008 | 2006,2007 | 10.17190/AMF/1246145 | Peters et al. (2011) |
| US-AR1 | ARM USDA UNL OSU Woodward Switchgrass 1 | GRA | 15.6 | 611.0 | 1.0[3] | 2.84 | 36.43 | −99.42 | 2012 | 2010,2011 | 10.17190/AMF/1246137 | - |
| CN-DNT | Rice-wheat rotation cropland Dongtai country, Jiangsu | CRP, BSV[4] | 15.1 | 4.0 | 0.6 (R) 0.5 (W) | 10.0 | 32.76 | 120.47 | 2015 (R)[5] 2015-16 (W)[6] | 2016 (R) 2014-15 (W) | - | Duan *et al.* (2020) |
| JP-SWL | Suwa Lake site Suwa city, Nagano | WAT | 14.6 | 759.0 | - | 3.0 | 36.04 | 138.10 | April 2015 | May-Dec 2015 | www.asiaflux.net | Iwata *et al.* (2018) |
| PH-IRI | Los Banos, Laguna | CRP, BSV[4] | 27.5 | 21.0 | 1.0 | 2.25 | 14.2 | 121.3 | 2014[7] | 2013 | www.asiaflux.net | Alberto *et al.* (2009) |

**1** For years used in this study
**2** Source: Keyser *et al.* (2008)
**3** Estimated from Porter (1966)
**4** First 4 weeks after planting rice - considered as soil surface.
**5** Rice planted and harvested: Jun 20-Nov 7 in 2015 and Jun 16-Nov 5 in 2016. Field flooded (0.15 m) until 5 weeks prior to harvest.
**6** Wheat planted and harvested: Dec 15-May 31 in 2014-15 and Dec 10-May 25 in 2015-16. Field kept saturated the entire period.
**7** Rice planted and harvested: Jun 27-October 22 in 2013 and Jun 17-Oct 1 in 2014. Field flooded (0.3 m) until 2 weeks prior to
harvest.

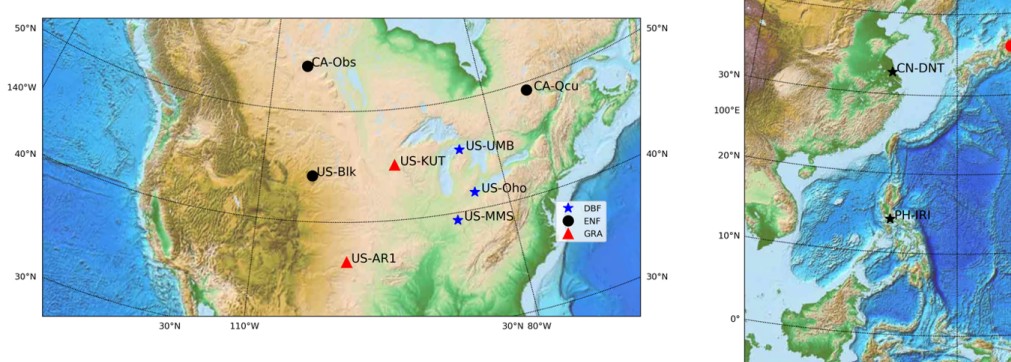


*Figure 1: Location of sites (Table 3) analysed by vegetation type deciduous trees (DBF), evergreen trees*
*(ENF), grass (GRA), water (WAT) and crops (CRP). Source of base maps: Basemap (2012)*





### 2.3 Determination of SUEWS parameters for pervious surfaces


The processes and parameters of interest (Sect. 2.1, Table 1) are not completely independent
for vegetated surfaces as both $LAI$ and albedo influence surface conductance, hence $Q_E$. As
$LAI$ varies with vegetation type, season and climate (e.g. latitude, local site characteristics), this
should be determined prior to albedo, surface conductance and $Q_E$; whereas neither bare soil
nor water surfaces require $LAI$ (Fig. 2). At each site, the $LAI$ and albedo model parameters are
derived with one year of data ('calibration') and evaluated with other years ('test') (Table 3, Fig.
3). Given limited data for the water site (JP-SWL, Table 3) the albedo is determined for April
2015 and evaluated for the remaining months (Fig. 3). Calibration data are used to derive $z_0$ and
$z_d$ (Eq. 9) using the methods in Appendix B. These values are used in the $Q_E$ evaluation.
To assess the generality of the derived parameters (chosen based on minimized MAE) for a
surface type, most are evaluated against both (Fig. 3): (a) another year at the same site, and (b)
two independent sites using one year of data. However, lack of data prevents this for bare soil,
crop, and water sites.
The Python package SuPy v2020.3.18 (Sun & Grimmond 2019) with the calculation kernel
SUEWS v2020a (Sun et al., 2020 ,Appendix A) is used for all simulations. The 5-min
simulations are averaged to 30-min for consistency with the eddy covariance observations
(Table 3). The complete Python source code (with comments) are provided at Omidvar et al.

302 (2020).


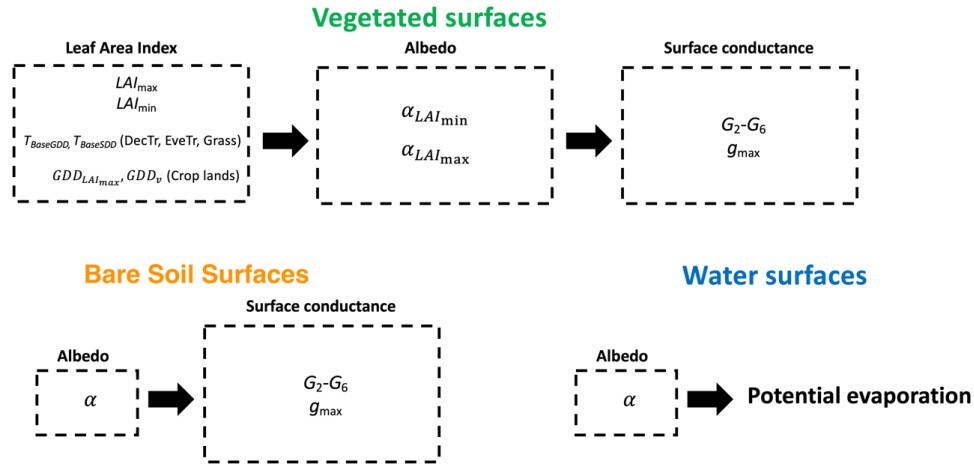

*Figure 2: To obtain SUEWS parameters for LAI, albedo and surface conductance, the order indicated is*
*required to be used. For notation see Table 1*

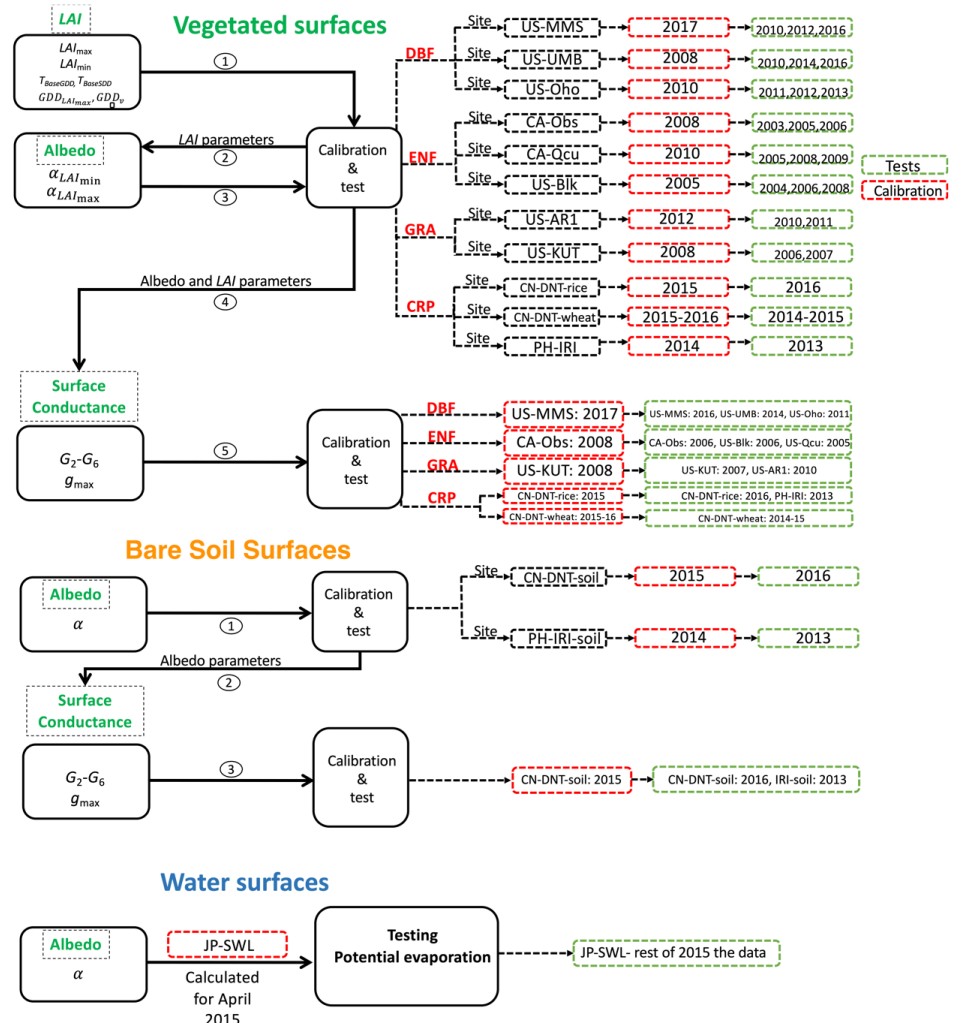


*Figure 3: Sites and periods (Table 3) used to derive (calibrate) and evaluate (test) the parameters related*

*to LAI, albedo and surface conductance for non-urban land types. Numbers in circles indicate*

*order of calculation. Notation defined in Table 1.*

## 2.4 Evaluation metrics

To evaluate the model output ($Y_{mod}$) with observations $(Y_{obs})$ for a number ($N$) of data points the following metrics are used:

1) mean absolute error (MAE):

$$MAE = \frac{\sum_{i=1}^{n}|Y_{mod} - Y_{obs}|}{N} \qquad (22)$$





2)   mean bias error (MBE):
$$\text{MBE} = \frac{\sum_{i=1}^{n}(Y_{mod} - Y_{obs})}{N}$$   (23)
Both the MAE and MBE are ideally 0 (with units of parameter/variable assessed).
3) normalised MAE (nMAE):
$$\text{nMAE} = \frac{\text{MAE}}{\text{MAE}_{calib}}$$   (24)
This is used to assess the model performance relative to data used to derive the parameters
(calib). If nMAE > 1 the performance is poorer with the test data set than the calibration set (and
vice versa).

To evaluate the evaporation, data are stratified by *LAI* phenology: (1) leaf off/leaf on/transition
for DBF, ENF and GRA sites (Sect. 3.1, 3.3) and (2) vegetative/reproductive/ripening for crops.
Crop dates are available for CN-DNT (Table 3) but not for PH-IRI-rice. As different states are
not available for the PH-IRI-rice, BSV and WAT sites $Q_E$ evaluation uses the entire period.
**3   Results and Discussion**
**3.1   *LAI* parameters**
Fig. 4 shows how different parameters control the *LAI* dynamics (Eq. 3) at the deciduous forest
site US-MMS (Table 3, Fig. 1) in 2017. At this site, *LAI* begins to increase from its minimum (0.5
$m^2\ m^{-2}$) as the daily mean air temperature ($T_d$) increases. As $T_d$ increases above $T_{BaseGDD}$ *LAI*
increases to its maximum (i.e. 5). *LAI* remains constant until $T_d$ goes below $T_{BaseSDD}$ when *LAI*
starts decreasing until it reaches the minimum (i.e. 0.5). Whereas for rice (Fig. 5, CN-DNT site)
the *LAI* evolution from planting has a short peak period with almost symmetric ascending and
descending parts. Given this different behaviour in *LAI* evolution between crops and other
vegetation types, two different forms (Eq. 3 and 5) are used.
Across all sites and years, the calculated *LAI* (Eq. 3, Table 4 parameters) have good agreement
with the MODIS *LAI* product (Sect. 2.1.1) (Fig. 6, 7). Based on entire years, all MAE are less
than 0.67 $m^2\ m^{-2}$ and the MBE are between −0.36 and 0.16 $m^2\ m^{-2}$ (Table D1). The largest
deviation from the MODIS *LAI* occurs at a grassland site (US-AR1) in 2011. A possible
explanation for this may be a lack of rain, as in 2011, US-AR1 received half the rainfall of the
other years, leading to larger soil moisture deficits (cf. 2010, 2012 (calibration)) (Fig. 8). This





important role of rainfall and soil moisture in moderating *LAI* dynamics with shallow vegetation
roots is also found by Bobée *et al.* (2012).
As expected, deciduous tree (DBF) sites have the largest variation in *LAI* among the vegetated
areas whereas grass has the smallest (Fig. 6, Table 4). However, the *LAI* variation at the
evergreen sites (ENF) indicates that assuming a constant *LAI* would result in poor predictions of
albedo and consequently turbulent heat fluxes. Consistent with Liu *et al.* (2013) and Alemu &
Henebry (2016), for each vegetation type $T_{BaseGDD}$ and $T_{BaseSDD}$ generally decrease with
increase in latitude (Table 3, 4). However, CA-Obs has slightly larger values than CA-Qcu
despite its higher latitude.
For both rice and wheat Eq. 5 performs well (Fig. 9, Table D1). The sharp decrease of *LAI* after
its peak in both rice and wheat is captured (Fig. 9a-c). For these crops, the MAE is < 0.53 $m^2$ $m^{-2}$
$^2$ and MBE is between -0.31 and 0.19 $m^2$ $m^{-2}$.
Generally, both Eq. 3 and 5 perform similarly when derived and evaluated (Fig. 7c, 9d). This
suggests the *LAI* calibration parameters from other years can be used. Recommended values
are given in Sect. 4. Although Eq. 3 performance varies between calibration and test sites with
phenology, no general trend is found (Fig. 7c).

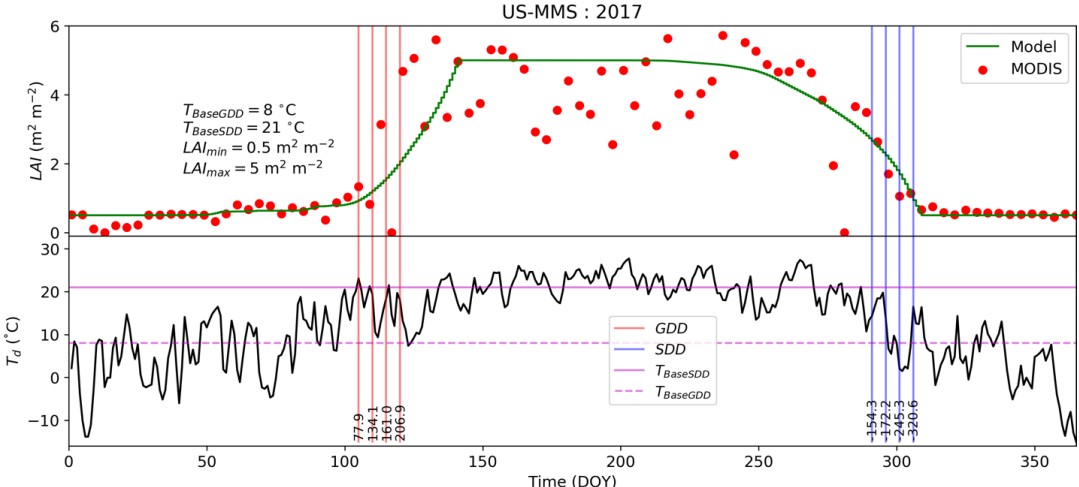


*Figure 4: Deciduous forest (US-MMS 2017, Table 3) (a) LAI ($m^2$ $m^{-2}$) modelled (Eq. 3) and from MODIS*
*(Myneni et al. 2015) with values of $T_{BaseSDD}$, $T_{BaseGDD}$, LAI$_{min}$ and LAI$_{max}$ ; and (b) $T_d$ (Eq. 4). Vertical*
*lines (5 days apart) give GDD (red) and SDD (blue) values (°C) relative to $T_{BaseGDD}$ (solid) and*
*$T_{BaseSDD}$ (dashed) (horizontal purple lines, °C). Notation is given in Table 1.*



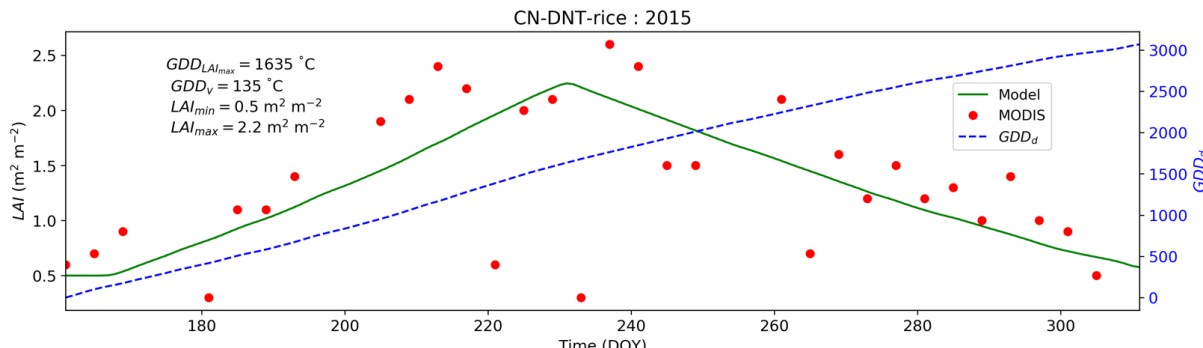


*Figure 5: Rice field (CN-DNT-2015 from June 10 (planting) to November 7 (harvest)) LAI: modelled (Eq.*


*5) and MODIS (Myneni et al. 2015) with values of $GDD_{LAI_{max}}$ (°C), $GDD_{LAI_{min}}$ (°C), LAI$_{min}$ (m$^2$ m$^{-2}$)*


*and LAI$_{max}$ (m$^2$ m$^{-2}$); and GDD$_d$ (right axis, °C). Notation is given in Table 1.*


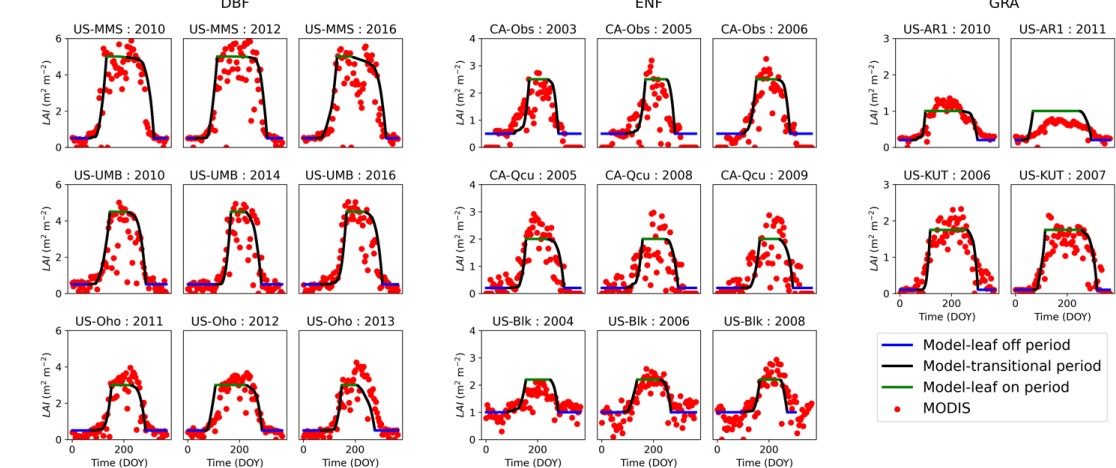


*Figure 6: Comparison of LAI (m$^2$ m$^{-2}$) calculated (lines, Eq. 3, Table 4 parameters) and MODIS (dots,*


*Myneni et al. 2015) for deciduous (DBF), evergreen (ENF) and grass (GRA) sites (Table 3, Fig.3)*


*for different years with modelled maxima (green), leaf off (blue), and transitional*


*growth/senescence (black) periods shown.*




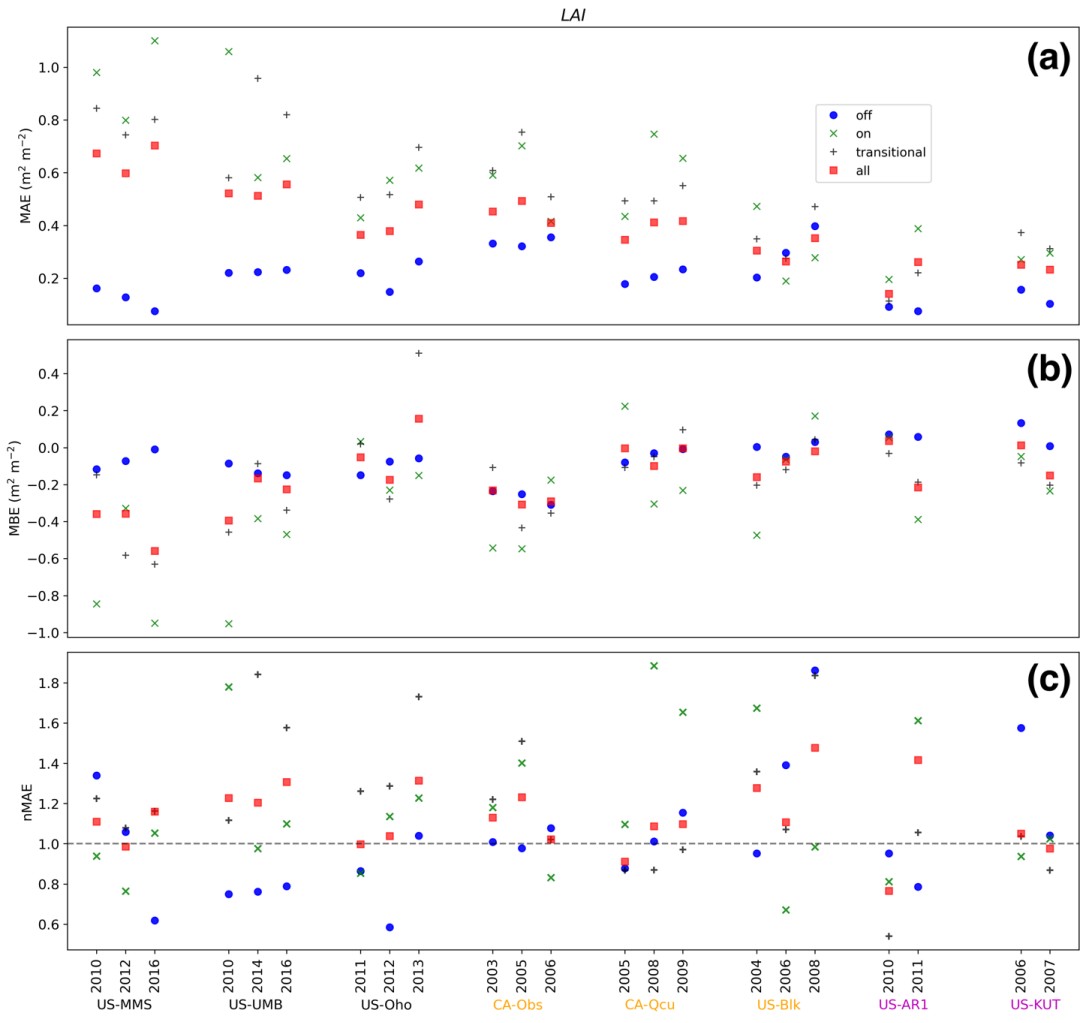


Figure 7: Modelled LAI (Eq. 3, Table 4 parameters) evaluated using MODIS (Myneni et al., 2015) for
        entire year (all), leaf on period (maxima), leaf off period (minima), and transitional period
        (growth/senescence period) for DBF (black x-axis label), ENF (yellow) and GRA (purple) sites.
        Performance metrics (Sect. 2.4) **a**: MAE **b**: MBE **c**: nMAE (Sect. 2.4)



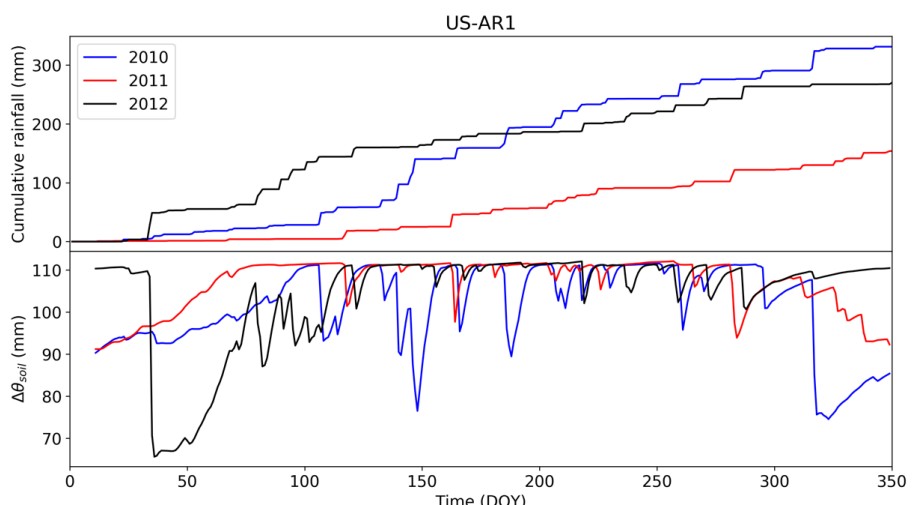


Figure 8: Grassland site (US-AR1) (a) cumulative rain and (b) SUEWS modelled soil moisture deficit
($\Delta\theta_{soil}$) for three years: 2012 (calibration), 2010 and 2011 (test years). SUEWS spun up for each
year using the same year data until the soil moisture deficit converges.

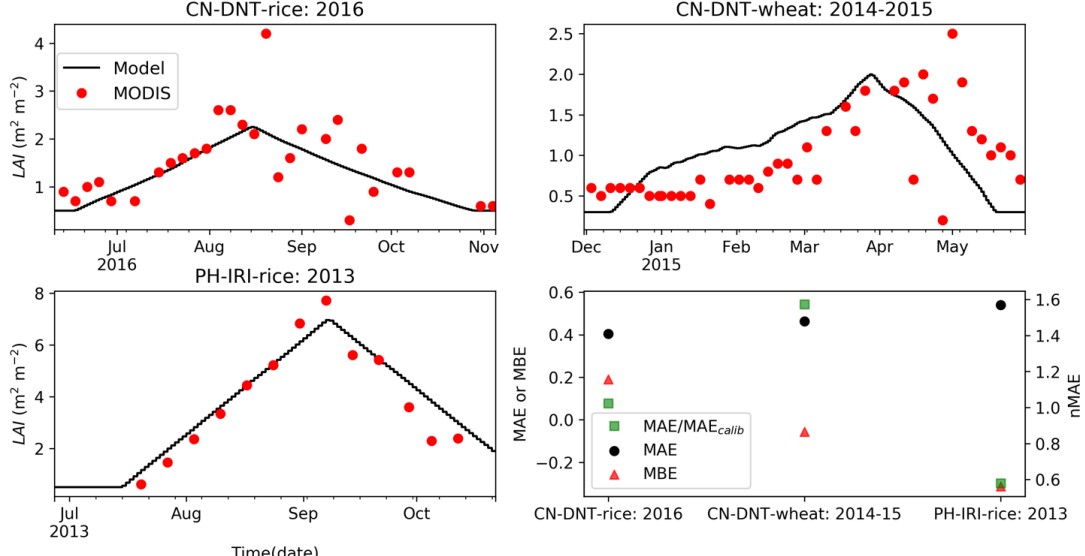


Figure 9: Crop site (Table 3) LAI ($m^2\ m^{-2}$) results **a-c**: using Eq. 5 with Table 4 parameters (lines) and
MODIS (dots, Myneni et al., 2015) by time and site (Table 3, Fig.3); and **d:** evaluation statistics
(Sect. 2.4).






Table 4: Parameters derived for LAI using Eq. 3 (DBF, ENF, GRA) and Eq. 5 (RIC, WHT) and albedo (Eq. 6). Bare soil (BSV) $\alpha_{LAI_{min}}$ is derived from the first 4 weeks of CN-DNT-rice. Water (WAT) is only for JP-SWL. Crop site air temperature at planting ($T_{plant}$) * is the 5 day mean.

| Site | Cover | $LAI_{min}$ | $LAI_{max}$ | $T_{BaseSDD}$ | $T_{BaseGDD}$ | | $\alpha_{LAI_{min}}$ | $\alpha_{LAI_{max}}$ |
|---|---|---|---|---|---|---|---|---|
| | | m² m⁻² | m² m⁻² | °C | °C | | - | - |
| US-MMS | DBF | 0.5 | 5.0 | 21 | 8 | | 0.10 | 0.14 |
| US-UMB | DBF | 0.5 | 4.5 | 20 | 6 | | 0.10 | 0.14 |
| US-Oho | DBF | 0.5 | 3.0 | 21 | 8 | | 0.10 | 0.14 |
| CA-Obs | ENF | 0.5 | 2.5 | 15.0 | 5 | | 0.08 | 0.07 |
| CA-Qcu | ENF | 0.2 | 2.0 | 11 | 2 | | 0.08 | 0.15 |
| US-Blk | ENF | 1.0 | 2.2 | 16 | 5 | | 0.08 | 0.07 |
| US-AR1 | GRA | 0.2 | 1.0 | 20 | 5 | | 0.14 | 0.19 |
| US-KUT | GRA | 0.1 | 1.7 | 13 | 3 | | 0.18 | 0.21 |
| | | | | $T_{plant}$ | $GDD_v$ | $GDD_{LAI_{max}}$ | | |
| | | | | °C | °C | °C | | |
| CN-DNT | RIC | 0.5 | 2.25 | 22.5* | 135 | 1635 | 0.10 | 0.17 |
| CN-DNT | WHT | 0.3 | 2.0 | 9.0* | 90 | 770 | 0.12 | 0.18 |
| PH-IRI | RIC | 0.5 | 7 | 29.0* | 475 | 1970 | 0.09 | 0.18 |
| JP-SWL | WAT | - | - | - | - | - | 0.05 | - |
| CN-DNT | BSV | - | - | - | - | - | 0.10 | - |

## 3.2 Albedo parameters

The daily albedo simulated with Eq. 6 (Table 4 parameters) clearly shows similar intra-annual evolution as the observations (Fig. 10, 11, snow-free periods, α< 0.3). Some sites (e.g. CA-Qcu) have an α ~ 0.85 during snow. Although the snow flags (Sect. 2.1.2) do not identify all snow days (i.e. high albedo), they approximately indicate snow periods.

As our sites are snow-free between May and October (Fig. 10, F1), the independent evaluations use this period (except for crops). The crops are evaluated between planting and harvest (Fig. 11). Overall, the modelled and observed albedos are in good agreement (Fig. 12, Table D2) during the snow-free periods (May-October for AmeriFlux sites, and entire period for other sites) with $\text{MAE} < 0.025$, $-0.012 < \text{MBE} < 0.025$ and $0.5 < \text{nMAE} < 1.6$ (Fig. 12). Water (0.05) and bare soil ( 0.10) albedo are treated as constants (Table 4, consistent with Gascoin et al. (2009) and Nunez et al. (1972)).


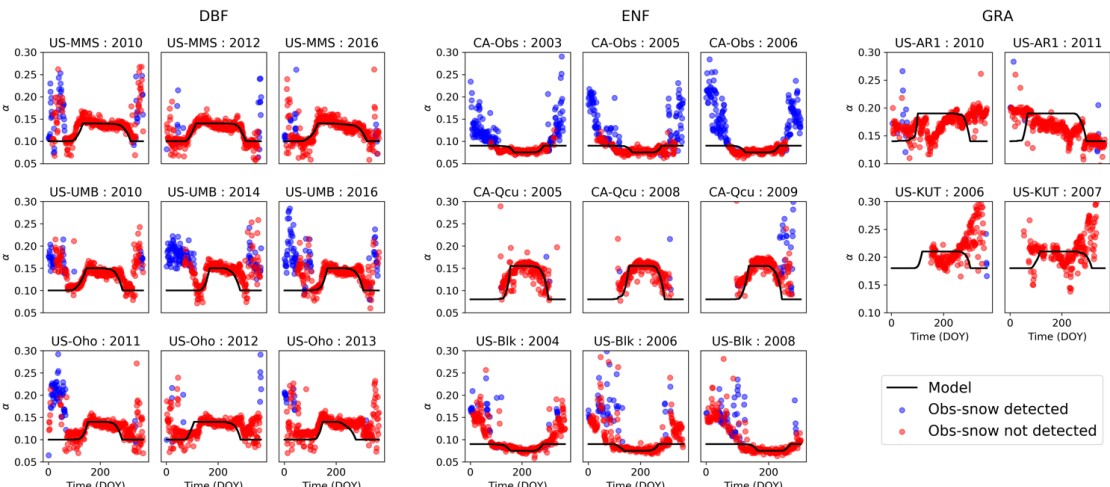

Figure 10: Modelled (Eq. 6, lines) and observed (Sect. 2.1.2) daily albedo by time (day of year, red –
408         snow free, blue - snow-covered, Sect. 2.3) for different vegetation types (Table 3). Note y-axis
409         (albedo) varies between vegetation types.


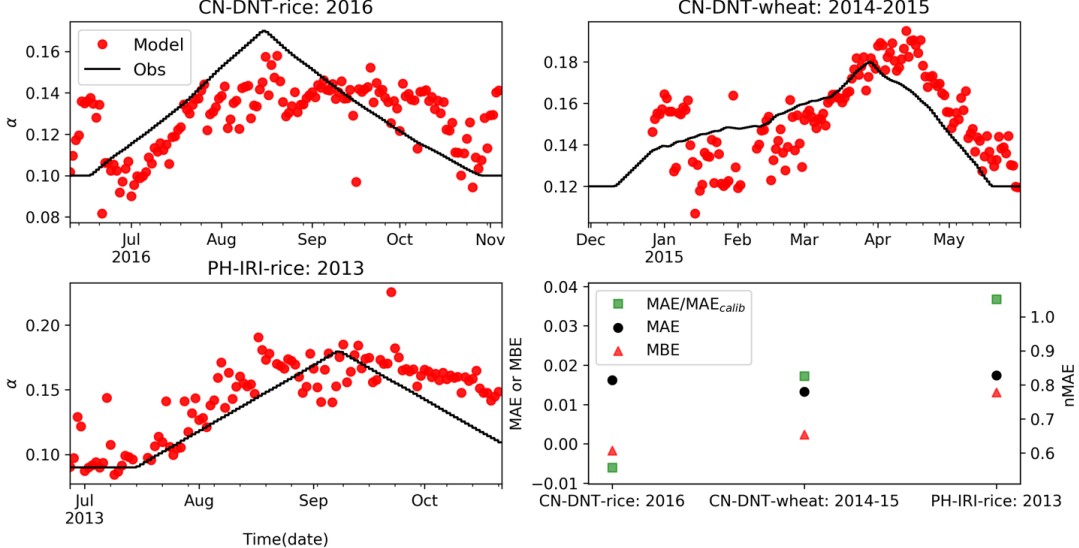

Figure 11: Daily crop albedo **a-c:** modelled (Eq. 6 with Table 4 parameters, lines) with observations by
412         time (date) for three cases (Table 3), and **d:** evaluation statistics.



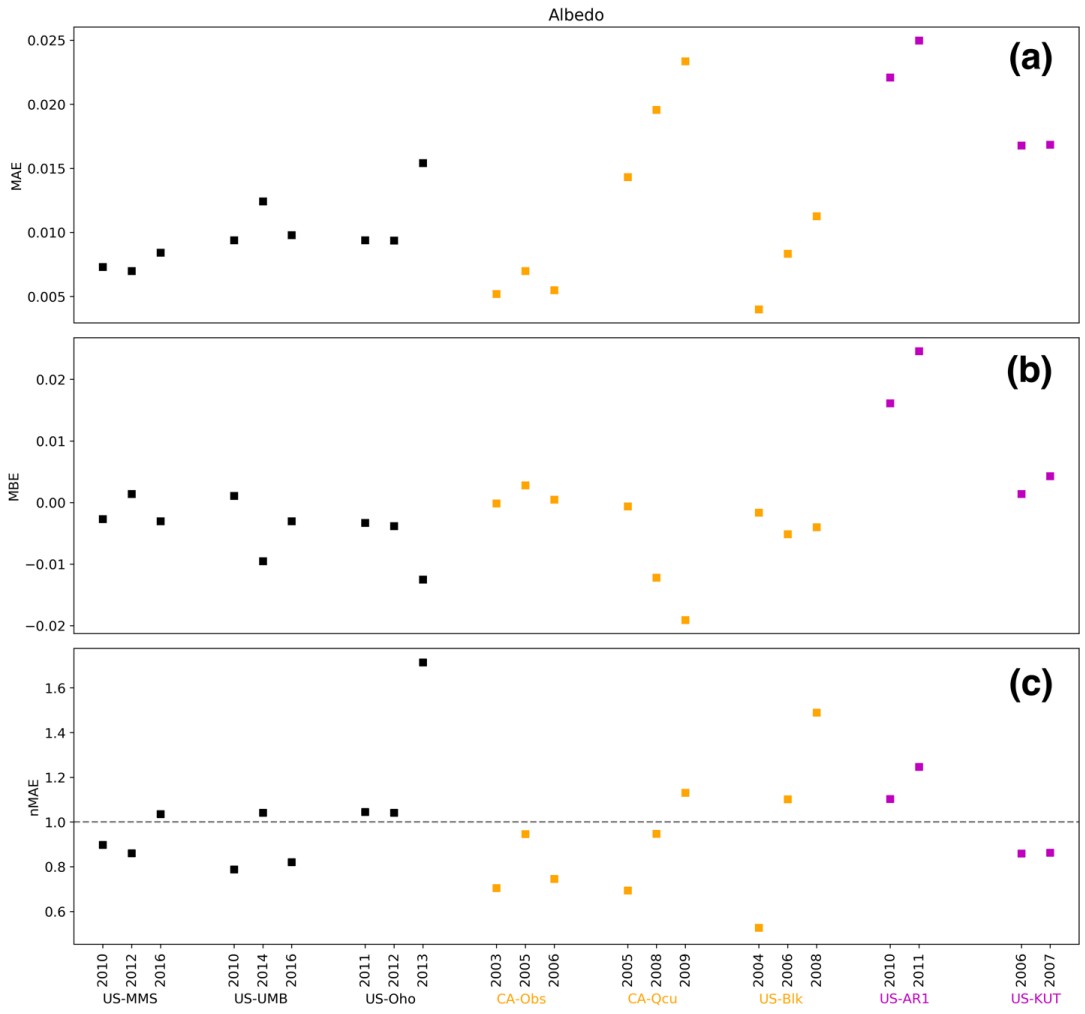

Figure 12: As Fig. 7, but for albedo assessed during a snow free period (May- October).

## 3.3 Surface conductance parameters

To model $Q_E$, $g_{\max}$ and $[G_2 - G_6]$ are essential (Sect. 2.1.4). Here observed $K_\downarrow$, $T_a$, $\Delta q$ and modelled $LAI$ (Fig. 2, 3) and $\Delta \theta_{soil}$ are used when fitting the parameters (Table 5). The values obtained for different pervious land cover types are summarised in Table 5. $G_2$ (related to $K_\downarrow$), $G_5$ (related to $T_a$) and $G_4$ (related to $\Delta q$) do not vary substantially among different land types. $G_6$ (related to $\Delta \theta_{soil}$) is quite similar for DBF, ENF and GRA but varies for other land cover types. However, $g_{\max}$ varies between all the land cover types.

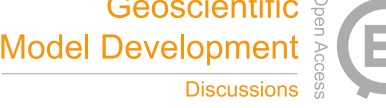

Using the derived parameters (Table 5), the SUEWS ability to predict $Q_E$ is assessed across the
test sites and years (Fig. 13, 14; Table D3). In general, for the DBF, ENF and GRA sites the
MAE (for all *LAI* states) is less than 58.5 W m$^{-2}$ (Table D3) with a slight overestimation for most
of the sites in the leaf-on period (e.g. US-MMS, CA-Obs, CA-Qcu; MBE: 8.8 to 40.4 W m$^{-2}$; Fig.
14, Table D3). $Q_E$ is overestimated in the leaf-transitional period at US-AR1 (MBE = 8.1 W m$^{-2}$)
and underestimated at US-KUT (MBE = $-18.2$ W m$^{-2}$). For CRP, WAT and BSV, the MAE of
$Q_E$ is generally less than 44.5 W m$^{-2}$. For WAT, the smaller nocturnal overestimation of $Q_E$ may
result from overestimation of nocturnal storage heat flux.
Multiple factors influence the $Q_E$ performance: over/under estimation of *LAI* (modifying albedo
and conductance) at vegetated sites; over/under prediction of storage heat flux (from for
example, missing moisture feedbacks); and/or assuming homogeneous fetch around each site.
Compared with using urban specific parameters, such as those derived for London and
Swindon, Ward *et al.,* 2016), those derived for non-urban land covers (Table 5) improve
SUEWS $Q_E$ performance (Appendix E): MAE is reduced (cf. $\mathrm{MAE}_{Ward}$) and nMAE is less than
one for all the sites (Fig. E2).
*Table 5: Surface conductance (Eq. 13-16) parameters (sites, Fig. 3) derived for different land cover types.*
*Note individual site values are not reported.*

| Land cover | $g_{max}$ (m s$^{-1}$) | $G_2$ (W m$^{-2}$) | $G_3$ | $G_4$ | $G_5$ (°C) | $G_6$ (mm$^{-1}$) |
|---|---|---|---|---|---|---|
| DBF | 89.9 | 104.10 | 0.16 | 0.57 | 25.92 | 0.028 |
| ENF | 14.9 | 104.64 | 0.70 | 0.63 | 36.62 | 0.022 |
| GRA | 24.2 | 104.85 | 0.49 | 0.61 | 36.63 | 0.022 |
| RIC | 234.8 | 105.13 | 0.97 | 0.75 | 36.91 | 0.046 |
| WHT | 747.5 | 104.45 | 0.16 | 0.70 | 37.37 | 0.048 |
| BSV | 10.9 | 108.93 | 0.93 | 0.96 | 42.26 | 0.041 |



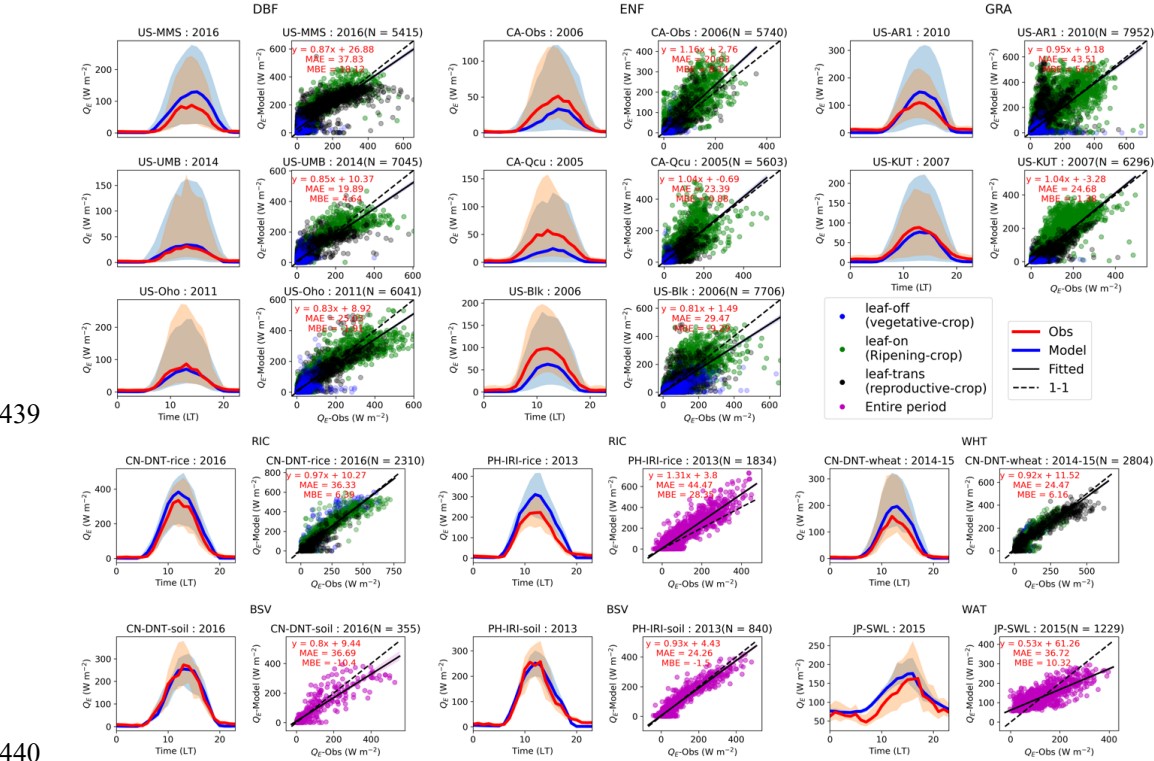

Figure 13: Latent heat flux for different sites (Table 3) calculated by SUEWS (Eq. 8 with Table 5 parameters) with annual diurnal pattern (median (lines) and interquartile range (shading)) for observed (red) and model (blue)) and scatter (dots, colour for LAI period, and N is the number of data points). Note for water and flood period for rice, $r_s$ in Eq. 12 is zero and potential evaporation is calculated.



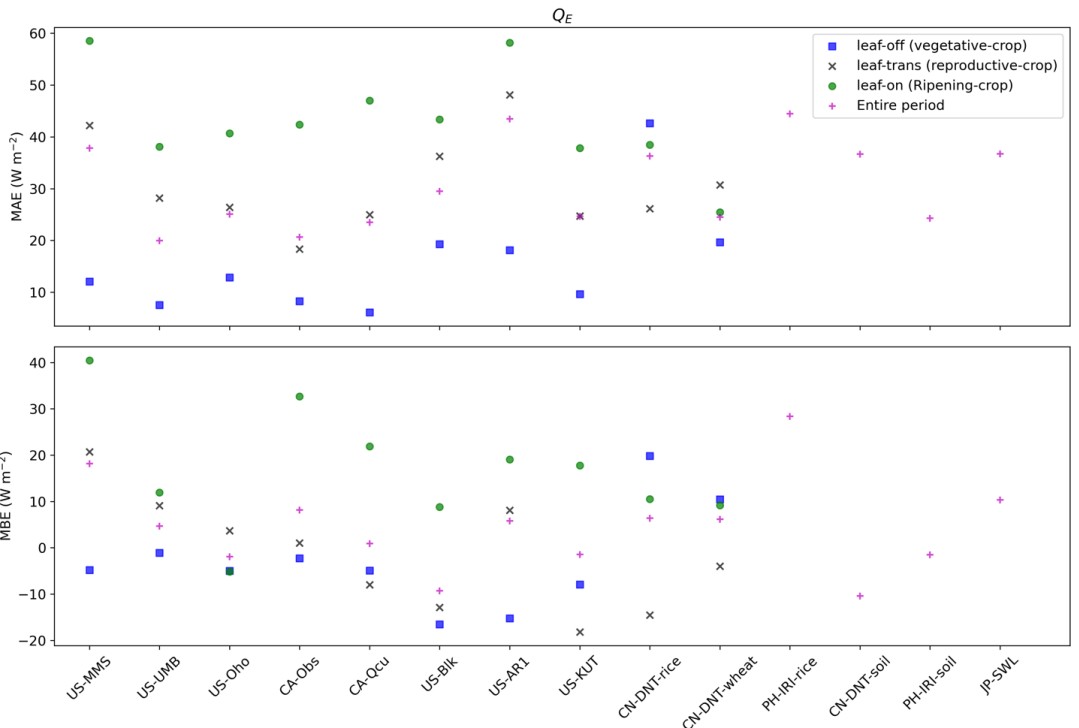


*Figure 14:  As Fig. 7, but for $Q_E$ (Eq. 8, Table 5 parameters). Units W m$^{-2}$*
**4    Concluding remarks**
New SUEWS parameters to simulate *LAI*, albedo and latent heat flux for different extensive
pervious (i.e. non-urban) land covers are derived and independently evaluated. The Python
Jupyter Notebooks protocol to derive the parameters is provided (Fig. 2, GitHub repository in
Omidvar *et al.,* 2020). This can be applied to other sites (or to other time periods at these sites).
The order of parameter determination is critical (*LAI* → albedo → surface resistance/
conductance, Fig 2, 3) to ensure appropriate values are obtained.
Recommended values are given in Table 6 based on the variability of different parameters
derived in this paper. In agreement with previous studies (e.g. Bobée *et al.,* 2012), we find that
soil moisture impacts *LAI* for vegetation with shallow roots (e.g. grass). This feedback should be
considered in future LAI modelling for SUEWS.
Using the derived (Table 2, 4, 5, 6) parameters or obtaining new values from the protocol
(Omidvar *et al.,* 2020) gives broader applicability of SUEWS in non-urban areas and thus





improves the model performance (cf. SUEWS runs using urban specific resistances, assuming
$f_i$= 1). Use of these derived parameters in online SUEWS applications should improve
representation of land-atmosphere interactions.
Table 6: Recommended values for SUEWS parameters (Table 1) for pervious land cover where ranges in
*LAI*, albedo and roughness parameters indicate regional variations.

| Cover | | | DBF | ENF | GRA | RIC | WHT | BSV | WAT |
|---|---|---|---|---|---|---|---|---|---|
| **LAI**(Table 4) | | | | | | | | | |
| $LAI_{min}$ | | m² m⁻² | 0.5 | 0.2-1.0 | 0.2-1.0 | 0.5 | 0.2 | - | - |
| $LAI_{max}$ | | m² m⁻² | 3.0-5.0 | 2.0-2.5 | 1.0-1.7 | 1.7-7.0 | 2.0 | - | - |
| $T_{BaseSD}$ | | °C | 20-21 | 11-16 | 13-20 | - | - | - | - |
| $T_{BaseGD}$ | | °C | 6-8 | 2-5 | 3-5 | - | - | - | - |
| $GDD_v$ | | °C | - | - | - | 135-475 | 90 | - | - |
| $GDD_{LA}$ | | °C | - | - | - | 1635-1970 | 770 | - | - |
| **Albedo**(Table 4) | | | | | | | | | |
| $\alpha_{LAI_{min}}$ | | - | 0.1 | 0.8 | 0.14-0.18 | 0.09-0.10 | 0.12 | 0.1 | 0.05 |
| $\alpha_{LAI_{max}}$ | | - | 0.14 | 0.07-0.15 | 0.19-0.21 | 0.17-0.18 | 0.18 | - | - |
| **Surface conductance**(Table 5) | | | | | | | | | |
| $g_{max}$ | | m s⁻¹ | 33.5 | 21.8 | 13.8 | 276.8 | 660.8 | 10.9 | - |
| $G_2$ | | W m⁻² | 104.82 | 104.38 | 104.47 | 104.71 | 105.08 | 108.93 | - |
| $G_3$ | | - | 0.53 | 0.51 | 0.79 | 0.19 | 0.17 | 0.93 | - |
| $G_4$ | | - | 0.61 | 0.77 | 0.59 | 0.57 | 0.68 | 0.96 | - |
| $G_5$ | | °C | 36.3 | 36.28 | 37.24 | 36.46 | 36.76 | 42.26 | - |
| $G_6$ | | mm⁻¹ | 0.03 | 0.023 | 0.025 | 0.049 | 0.044 | 0.041 | - |
| **OHM storage heat flux**(Table 2) | | | | | | | | | |
| $a_1$ | | - | 0.215 | 0.215 | 0.215 | 0.185 | 0.283 | 0.210 | 0.880 |
| $a_2$ | | s | 0.325 | 0.325 | 0.325 | 0.615 | 0.784 | 0.902 | 0.370 |
| $a_3$ | | W m⁻² | -19.9 | -19.9 | -19.9 | -18.0 | -18.0 | -20.4 | -85.4 |
| **Canopy water storage capacity**(Table 2) | | | | | | | | | |
| $S_i$ | | mm | 1.3 | 0.8 | 1.9 | 1.9 | 1.9 | 1.9 | - |
| **Aerodynamic roughness** (Table B1) by phenological state with $f_0$ (Eq. B2) and $f_d$ (Eq. B3) parameters | | | | | | | | | |
| $z_{0m}$ | m | Leaf-off/ vegetative \| 0.16 | 3.2-5.2 | 0.3-5.1 | 0.01-0.03 | 0.24 | 0.12 | 0.002 | 0.0005 |
| | | Trans./ reproductive\| 0.18 | 3.9-5.5 | 0.3-2.6 | 0.01 | 0.19 | 0.2 | | |
| | | Leaf-on/ ripening    \| 0.18 | 3.2-5.4 | 1.8-2.4 | 0.02-0.03 | 0.55 | 0.38 | | |
| $z_d$ | m | Leaf-off /vegetative \| 0.5 | 7.2-19.2 | 3.7-6.5 | 0.06-0.83 | 0.32 | 0.14 | 0 | 0 |
| | | Trans/ reproductive \| 0.44 | 8.2-15.4 | 2.2-6.6 | 0.06-0.9 | 0.39 | 0.45 | | |
| | | Leaf-on/ripening     \| 0.42 | 8.0-10.9 | 3.8-6.6 | 0.06-0.83 | 0.88 | 0.65 | | |






### Appendix A: SUEWS developments included in v2020a

#### A.1 SUEWS surface temperature ($T_s$) calculation

At each time step, the surface temperature $T_s$ is calculated iteratively. First $T_s$ is estimated by NARP (net-all radiation parameterization) (Offerle *et al.,* 2003; Loridan *et al.,* 2011) as a function of air temperature $T_a$ (i.e., $T_s^{NARP} = NARP(T_a)$), then $T_s^{NARP}$ is used to calculate $Q^*$ (via outgoing longwave radiation $L_\uparrow$). At the end of this iteration (*j*), $T_s$ is updated using sensible heat flux $Q_H$ and $T_a$ based on Monin-Obukhov similarity theory (MOST) to give a new value $T_{s,j}$ (*j*=1, initial iteration). In subsequent iterations, the NARP-based estimation of $T_s$ is skipped and $T_{s,j-1}$ (i.e., previous iteration) is used in the $Q^*$ calculation and updated to $T_{s,j}$ (i.e. current iteration) using MOST. Once $\left| T_{s,j-1} - T_{s,j} \right| <$ a prescribed tolerance, then $T_s = T_{s,j}$ and iteration stops (or for *j*= 20)

#### A.2    SUEWS irrigation scheme for crops

Automatic irrigation can be set (`WaterUseMethod=1` in RunControl.nml file) to maintain the water availability at a specified level $h_m$ (e.g. a certain depth of ponding water for flood irrigation of rice or a particular soil moisture state of other crops; by setting column $h_m$ of SUEWS_Irrigation.txt file, in mm). When it is a positive value it allows for flood irrigation (e.g. rice); otherwise, the soil moisture is maintained by irrigation at the maximum soil storage capacity minus $h_m$. The running water balance considering precipitation, irrigation, evaporation and runoff rates and the net change in storage is used to determine the irrigation needed (cf. Eq. 2) taking $h_m$ in to account. The irrigation needed ($I_N$) is determined at the last time step of the day. The $I_e$ water is applied the next day if needed (i.e. for $I_N$ > 0 mm) based on the rates specified by the user via the SUEWS automatic irrigation profile $f_a$. The automatic irrigation profile allows water to be supplied at the appropriate times of the day and intensity for the region. If the water is applied too rapidly (e.g. all in one 5 min timestep) unrealistic runoff will occur. At each time step, the $I_N$ is checked to confirm that water is still needed (as determined by $I_N$ at the end of the previous day). If there is need remaining at the end of the day this will be included in the end of day water balance calculation for the next day.

#### A.3    SUEWS land cover adaptive $z_{0v}$ scheme

A new option RoughLenHeatMethod (choice 5) is included (RunControl.nml) that allows different $z_{0v}$ schemes to be used depending on the land cover characteristics. If no impervious cover exists in a grid ($f_{prv} = 1$) then Brutsaert's (1982) method is used:





$$z_{0v} = 0.1 z_{0m} \qquad\qquad (A1)$$

Otherwise ($f_{prv}$ <1) Kawai *et al.* (2009) is used:
$$z_{0v} = z_{0m} \exp\left(2 - \left(1.2 - 0.9 f_{prv}^{0.29}\right)\left(\frac{u_* z_{0m}}{\mu}\right)^{0.25}\right) \qquad (A2)$$

where $z_{0m}$ is the roughness length for momentum, $u_*$ the friction velocity, and $\mu$ the molecular
diffusivity of air.

**Appendix B: Roughness length and zero-displacement height for momentum**
**B.1 Methods**
The roughness parameters ($z_{0m}$, $z_d$) are derived during neutral stability ($|(z_m$ - $0.7h_v)/L)|$ <0.01, i.e.
assuming initially $z_{d=}0.7h_v$) using observed $u_*$ and $u$ (Monin and Obukhov, 1954):
$$u = \frac{u_*}{\kappa} \ln\left[\frac{z_m - z_d}{z_{0m}}\right] \qquad\qquad (B1)$$

These can also be obtained simply using a rule of thumb (Garratt 1991, Grimmond and Oke

510 1999):

$$z_{0,m} = f_{0,i} h_{v,i} \qquad\qquad (B2)$$

$$z_d = f_{d,i} h_{v,i} \qquad\qquad (B3)$$

where $h_{vi}$ *is the vegetation height for type i and* $f_{0,i}$ and $f_{d,i}$ depend on porosity of the vegetation
type.
The same multi-objective evolutionary algorithm used to determine $G_2$ - $G_6$ (Sect. 2.1.4), is
applied with two objectives to optimize Eq. B1: (1) to minimize the normalized (n) standard
deviation (SD) of observations (obs) *of u*:
$$\text{nSD} = \frac{SD(u_{mod}) - SD(u_{obs})}{SD(u_{obs})} \qquad\qquad (B4)$$

(2) to minimize the *MAE of u (Eq. 22).*

As *LAI* state changes both $z_{0m}$ *and* $z_d$ (e.g. Kent et al. 2017b ) by modifying the porosity of the
canopy, the three phenological states (leaf off, on and transition, Sect. 3.1) are considered.
However, sufficient data (> 20, Grimmond et al. 1998) need to be used. By undertaking analysis
by wind direction for sites that appeared to have variable results (based on modelled *u*) it is also
possible to identify sites that have varying fetch by wind direction (e.g. CA-Qcu). This can be



confirmed using visible wavelength satellite imagery. As CA-Qcu's fetch is found to vary, data
are analysed with 10° direction bins (for each *LAI* state) with the median $z_{0m}$ and $z_d$ used.

To obtain the parameters $f_{0,i}$ and $f_{d,i}$ (Eq. B2 and B3) vegetation heights are needed (Table 3).
As crop height varies substantially through a season, where heights are available (e.g. C-DNT,
Duan *et al.* 2020) these are used. However, for others only one height is used and $z_{0m}$, $z_d$ are
calculated for the entire growth period (e.g. PH-IRI). The training years (Table 3) are used to
derive $z_{0m}$, $z_d$ and subsequently $f_{0,i}$ and $f_{d,I}$.
**B.2 Results**
Analysis of 'observed' $z_{0m}$ and $z_d$ *(Eq.* B1*)* with height suggests $f_0$ and $f_d$ (Eq. B2) vary between
0.16 - 0.18 and 0.42-0.5 (Fig. B.1) across phenological states. These values for each *LAI* state
are used to derive $z_{0m}$, $z_d$ of test sites.

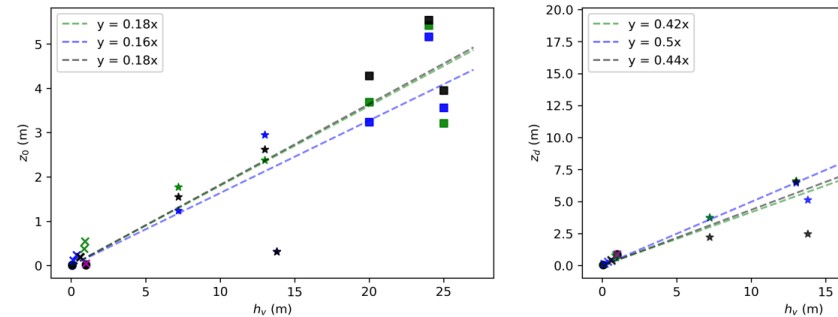


*Figure B1*: Micrometeorologically derived (Eq. B1) aerodynamic roughness parameters and vegetation
height (*Table 3) calculated for* the *calibration years by phenological state (Sect. 3.1)*: *(a)* $z_{0m}$ and
*(b)* $z_d$. The crop states heights vary for CN-DNT (Duan et al. 2020) [rice: vegetative =0.39 m,
reproductive=0.64 m, ripening=0.93 m; wheat: vegetative=0.15 m, reproductive = 0.58 m
ripening=0.86 m] but not for Ph-IRI.

The ability to predict *u is assessed* for different sites with LAI/crop state using Eq. B1 and the
derived $z_{0m}$ and $z_d$ values (Table B1). These are in generally good agreement with observations
(Fig. B2), with MAE < 1.32 m s$^{-1}$ and -1.03 < MBE <1.22 m s$^{-1}$.



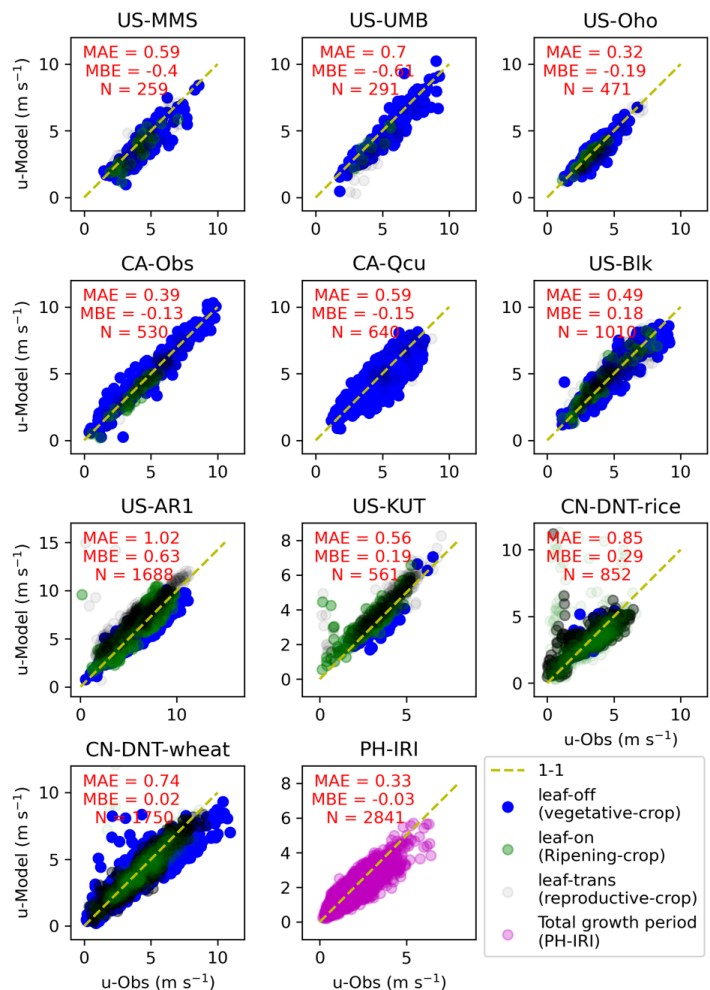


*Figure B2: Comparison of observed u (u-Obs) to modelled u (u-Model, $z_{0m}$ and $z_d$ Eq. B1, Table B1) at the vegetated sites for all training years (Table 3) with number (N) data points and phenology (Sect. 3.1) and CN-DNT crop states (Duan et al. 2020).*

*Table B1: Micrometeorological (Eq. B1) $z_{0m}$ and $z_d$ for vegetation sites (Table 3) for number (N) data points for different phenology periods (Sect. 3.1) and for CN-DNT crop states (Duan et al. 2020). MAE (m s⁻¹) and MBE (m s⁻¹) are calculated for u.*

| site | Leaf-off | | | | | Leaf-trans | | | | | Leaf-on | | | | |
|---|---|---|---|---|---|---|---|---|---|---|---|---|---|---|---|
| | $z_{0m}$ (m) | $z_d$ (m) | MAE | MBE | N | $z_{0m}$ (m) | $z_d$ (m) | MAE | MBE | N | $z_{0m}$ (m) | $z_d$ (m) | MAE | MBE | N |
| US-MMS | 3.56 | 19.19 | 0.58 | -0.38 | 145 | 3.95 | 8.2 | 0.66 | -0.40 | 89 | 3.21 | 10.9 | 0.54 | -0.41 | 25 |
| US-UMB | 3.24 | 7.22 | 0.58 | -0.44 | 221 | 4.28 | 15.36 | 1.03 | -1.03 | 48 | 3.69 | 9.06 | 0.50 | -0.36 | 22 |
| US-Oho | 5.16 | 8.18 | 0.33 | -0.19 | 354 | 5.54 | 9.54 | 0.46 | -0.38 | 90 | 5.42 | 7.99 | 0.16 | 0.0 | 27 |
| CA-Obs | 1.24 | 3.73 | 0.41 | 0.12 | 371 | 1.55 | 2.22 | 0.39 | -0.16 | 124 | 1.77 | 3.76 | 0.39 | -0.36 | 35 |
| CA-Qcu | 0.32 | 5.14 | 0.58 | -0.14 | 602 | 0.31 | 2.47 | 0.72 | -0.16 | 38 | - | - | - | - | - |
| US-Blk | 2.95 | 6.46 | 0.45 | 0.12 | 727 | 2.62 | 6.56 | 0.58 | 0.22 | 230 | 2.38 | 6.63 | 0.44 | 0.20 | 53 |
| US-AR1 | 0.03 | 0.83 | 0.82 | 0.28 | 938 | 0.01 | 0.9 | 1.32 | 1.22 | 442 | 0.03 | 0.83 | 0.91 | 0.38 | 308 |





| US-KUT | 0.01 | 0.06 | 0.59 | -0.38 | 54 | 0.01 | 0.06 | 0.57 | 0.52 | 330 | 0.02 | 0.06 | 0.5 | 0.43 | 177 |
|---|---|---|---|---|---|---|---|---|---|---|---|---|---|---|---|
| | Vegetative-crop | | | | | Reproductive-crop | | | | | Ripening-crop | | | | |
| CN-DNT-rice | 0.24 | 0.32 | 0.42 | 0.05 | 112 | 0.19 | 0.39 | 0.84 | 0.26 | 318 | 0.55 | 0.88 | 1.30 | 0.55 | 422 |
| CN-DNT-wheat | 0.12 | 0.14 | 0.91 | 0.02 | 754 | 0.2 | 0.45 | 0.48 | -0.14 | 575 | 0.38 | 0.65 | 0.85 | 0.19 | 421 |
| | Total Growth Period | | | | | | | | | | | | | | |
| PH-IRI-rice | 0.05 | 0.89 | 0.33 | -0.03 | 2841 | | | | | | | | | | |

555

## Appendix C: Latent heat flux ($Q_E$) in extremely cold conditions

As the surface conductance (Eq. 12) has an air temperature dependency with limits (i.e. $T_L$ in Eq. 15), we investigate the $T_L$ limit with $Q_E$ below -10°C for different sites (Fig. C1). Given this we use $T_L = -20°C$ for all sites as the limit when evaporation switches off in SUEWS.

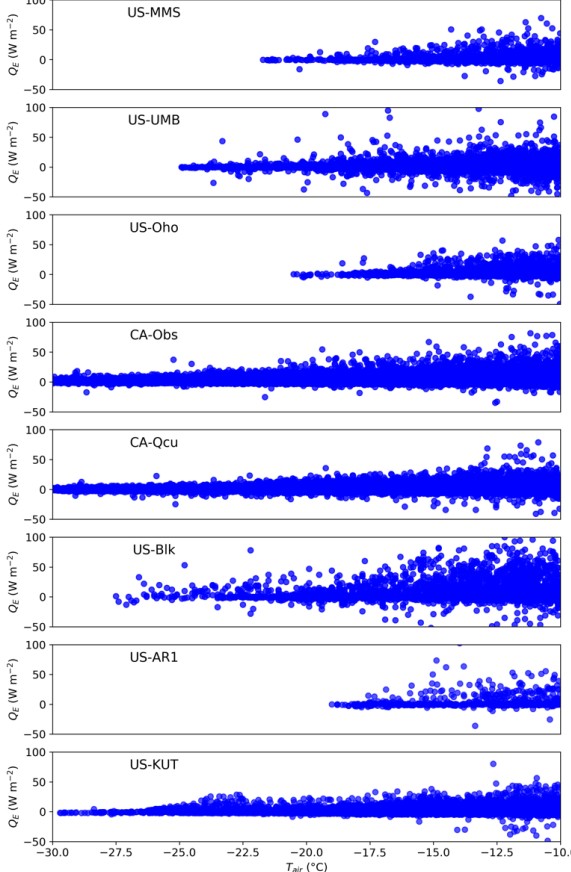


*Figure C1: Latent heat flux $Q_E$ variation with air temperature $T_a$ when $T_a < -10°C$ at eight sites (Table 3).*






**Appendix D: *Model evaluation statistics***
Sites (Table 3) used to evaluated (metrics Sect. 2.4) the parameters assessed LAI (Table D1),
albedo (Table D2) and latent heat flux (Table D3).
*Table D1: Evaluation of SUEWS modelled LAI (Eq. 3 or 5, with parameters Table 4) and MODIS LAI*
*product (Myneni et al., 2015) for entire year (all), leaf on period (model LAI - maxima), leaf off*
*period (LAI model - minima), and transitional period (growth/senescence period).Units: $m^2\ m^{-2}$.*
*N is the number of data points in each period for each site*

| site | year | All | | | Leaf-off | | | Transitional | | | Leaf-on | | |
|---|---|---|---|---|---|---|---|---|---|---|---|---|---|
| | | MAE | MBE | N | MAE | MBE | N | MAE | MBE | N | MAE | MBE | N |
| US-MMS | 2010 | 0.67 | −0.36 | 94 | 0.16 | −0.12 | 30 | 0.84 | −0.15 | 44 | 0.98 | −0.84 | 20 |
| | 2012 | 0.60 | −0.36 | 92 | 0.12 | −0.05 | 23 | 0.88 | −0.70 | 38 | 0.80 | −0.33 | 31 |
| | 2016 | 0.70 | −0.56 | 91 | 0.08 | −0.01 | 24 | 0.80 | −0.63 | 51 | 1.10 | −0.95 | 16 |
| US-UMB | 2010 | 0.52 | −0.39 | 92 | 0.22 | −0.09 | 38 | 0.58 | −0.46 | 36 | 1.06 | −0.95 | 18 |
| | 2014 | 0.51 | −0.17 | 92 | 0.22 | −0.14 | 48 | 0.96 | −0.09 | 29 | 0.58 | −0.38 | 15 |
| | 2016 | 0.56 | −0.23 | 90 | 0.23 | −0.15 | 30 | 0.82 | −0.34 | 41 | 0.65 | −0.47 | 19 |
| US-Oho | 2011 | 0.36 | −0.05 | 94 | 0.22 | −0.15 | 40 | 0.51 | 0.02 | 34 | 0.43 | 0.03 | 20 |
| | 2012 | 0.38 | −0.17 | 93 | 0.15 | −0.08 | 37 | 0.52 | −0.28 | 25 | 0.57 | −0.23 | 31 |
| | 2013 | 0.48 | 0.16 | 93 | 0.24 | −0.04 | 40 | 0.74 | 0.54 | 37 | 0.62 | −0.15 | 16 |
| CA-Obs | 2003 | 0.45 | −0.23 | 91 | 0.33 | −0.24 | 47 | 0.61 | −0.11 | 25 | 0.59 | −0.54 | 19 |
| | 2005 | 0.49 | −0.31 | 95 | 0.32 | −0.25 | 46 | 0.75 | −0.43 | 28 | 0.70 | −0.55 | 21 |
| | 2006 | 0.41 | −0.29 | 93 | 0.35 | −0.31 | 46 | 0.51 | −0.35 | 25 | 0.42 | −0.17 | 22 |
| CA-Qcu | 2005 | 0.35 | −0.03 | 92 | 0.20 | −0.13 | 38 | 0.48 | −0.13 | 32 | 0.43 | 0.22 | 22 |
| | 2008 | 0.42 | −0.13 | 94 | 0.23 | −0.08 | 43 | 0.48 | −0.08 | 38 | 0.75 | −0.31 | 23 |
| | 2009 | 0.42 | −0.04 | 93 | 0.26 | −0.05 | 42 | 0.54 | 0.07 | 41 | 0.66 | −0.23 | 20 |
| US-Blk | 2004 | 0.30 | −0.16 | 93 | 0.20 | 0.01 | 43 | 0.35 | −0.20 | 26 | 0.47 | −0.47 | 24 |
| | 2006 | 0.26 | −0.07 | 94 | 0.30 | −0.05 | 47 | 0.27 | −0.12 | 24 | 0.19 | −0.06 | 23 |
| | 2008 | 0.35 | −0.02 | 92 | 0.40 | 0.03 | 51 | 0.47 | 0.04 | 21 | 0.28 | 0.17 | 20 |
| US-AR1 | 2010 | 0.13 | 0.01 | 94 | 0.06 | 0.03 | 29 | 0.11 | −0.07 | 28 | 0.20 | 0.05 | 37 |
| | 2011 | 0.21 | −0.18 | 93 | 0.04 | 0.01 | 25 | 0.21 | −0.17 | 22 | 0.34 | −0.34 | 46 |
| US-KUT | 2006 | 0.25 | 0.01 | 93 | 0.16 | 0.13 | 24 | 0.37 | −0.08 | 23 | 0.27 | −0.05 | 36 |
| | 2007 | 0.23 | −0.15 | 92 | 0.10 | 0.01 | 31 | 0.31 | −0.20 | 25 | 0.30 | −0.23 | 36 |
| CN-DNT-rice | 2016 | 0.40 | 0.19 | 28 | - | - | - | - | - | | - | - | - |
| CN-DNT-wheat | 2014-15 | 0.46 | -0.06 | 42 | - | - | - | - | - | | - | - | - |
| PH-IRI | 2013 | 0.53 | -0.31 | 13 | - | - | - | - | - | | - | - | - |

*Table D2: As Table D1, but albedo (Eq. 6, Table 4) during snow-free periods (May- October; but the*
*entire period for crops).*

| site | year | MAE | MBE | N |
|---|---|---|---|---|
| US-MMS | 2010 | 0.007 | −0.003 | 177 |
| | 2012 | 0.007 | 0.001 | 183 |
| | 2016 | 0.008 | −0.003 | 182 |
| US-UMB | 2010 | 0.009 | −0.001 | 182 |
| | 2014 | 0.012 | −0.010 | 182 |
| | 2016 | 0.010 | −0.003 | 183 |
| US-Oho | 2011 | 0.009 | −0.003 | 183 |





| | | | | |
|---|---|---|---|---|
| | 2012 | 0.009 | −0.004 | 183 |
| | 2013 | 0.015 | −0.012 | 183 |
| CA-Obs | 2003 | 0.005 | −0.000 | 183 |
| | 2005 | 0.007 | −0.003 | 175 |
| | 2006 | 0.005 | −0.000 | 171 |
| CA-Qcu | 2005 | 0.014 | −0.001 | 180 |
| | 2008 | 0.020 | −0.012 | 180 |
| | 2009 | 0.023 | −0.019 | 183 |
| US-Blk | 2004 | 0.004 | −0.002 | 160 |
| | 2006 | 0.008 | −0.005 | 176 |
| | 2008 | 0.011 | −0.004 | 182 |
| US-AR1 | 2010 | 0.022 | 0.016 | 183 |
| | 2011 | 0.025 | 0.025 | 183 |
| US-KUT | 2006 | 0.017 | 0.001 | 145 |
| | 2007 | 0.017 | 0.004 | 148 |
| CN-DNT-rice | 2016 | 0.016 | -0.001 | 123 |
| CN-DNT-wheat | 2014-15 | 0.013 | 0.002 | 174 |
| PH-IRI | 2013 | 0.017 | 0.013 | 113 |


*Table D3: As Table D1, but for $Q_E$ (Eq. 8 with Table 5 parameters). Units: W m$^{-2}$*

| | | Leaf-off | | | Leaf-trans | | | Leaf-on | | | All | | |
|---|---|---|---|---|---|---|---|---|---|---|---|---|---|
| site | year | MAE | MBE | N | MAE | MBE | N | MAE | MBE | N | MAE | MBE | N |
| US-MMS | 2016 | 12.1 | -4.8 | 1330 | 42.2 | 20.7 | 3028 | 58.5 | 40.4 | 1057 | 37.8 | 18.1 | 5415 |
| US-UMB | 2014 | 7.5 | −1.1 | 3410 | 28.2 | 9.1 | 2347 | 38.1 | 11.9 | 1288 | 19.9 | 4.6 | 7045 |
| US-Oho | 2011 | 12.8 | -5.0 | 2255 | 26.4 | 3.7 | 2198 | 40.6 | −5.2 | 1588 | 25.0 | -1.9 | 6041 |
| CA-Obs | 2006 | 8.3 | -2.3 | 2352 | 18.3 | 1.0 | 1844 | 42.4 | 32.7 | 1544 | 20.6 | 8.1 | 5740 |
| CA-Qcu | 2005 | 6.1 | −4.9 | 2134 | 25.0 | -8.0 | 2007 | 47.0 | 21.9 | 1462 | 23.4 | 0.9 | 5603 |
| US-Blk | 2006 | 19.3 | −16.5 | 3867 | 36.2 | -12.9 | 1907 | 43.3 | 8.8 | 1938 | 29.5 | -9.3 | 7709 |
| US-AR1 | 2010 | 18.1 | −15.2 | 2338 | 48.1 | 8.1 | 2323 | 58.2 | 19.1 | 3291 | 43.5 | 5.8 | 7952 |
| US-KUT | 2007 | 9.6 | -7.9 | 2059 | 24.7 | -18.2 | 1899 | 37.8 | 17.8 | 2338 | 24.6 | -1.5 | 6296 |
| | | Vegetative | | | Reproductive | | | Ripening | | | | | |
| CN-DNT-rice | 2016 | 42.6 | 19.8 | 610 | 26.1 | -14.5 | 605 | 38.4 | 10.5 | 1095 | 36.3 | 6.4 | 2310 |
| CN-DNT-wheat | 2014 | 19.6 | 10.4 | 1153 | 30.7 | -4.0 | 752 | 25.4 | 9.2 | 899 | 24.5 | 6.2 | 2804 |
| PH-IRI-rice | 2013 | - | - | - | - | - | - | - | - | - | 44.5 | 28.3 | 1834 |
| CN-DNT-soil | 2016 | - | - | - | - | - | - | - | - | - | 36.7 | -10.4 | 355 |
| PH-IRI-soil | 2013 | - | - | - | - | - | - | - | - | - | 24.3 | -1.5 | 840 |
| JP-SWL | - | - | - | - | - | - | - | - | - | - | 36.7 | 10.3 | 1229 |


## Appendix E: $Q_E$ simulated with London and Swindon parameters

To demonstrate the necessity and benefit of using appropriate parameters to estimate $Q_E$ in
SUEWS, we compare $Q_E$ simulated at DBF, ENF and GRA pervious sites using Ward *et al.'s*





(2016) $g_{max}$ and $G_2$-$G_6$ parameters (derived for London and Swindon) (Fig. E1) to those derived
here (Table 5). In all cases the performance is improved using pervious area surface
parameters (e.g. *LAI,* albedo, surface conductance) than using the suburban/urban parameters
(assuming $f_i$=1 of the pervious area) (Fig. E2).

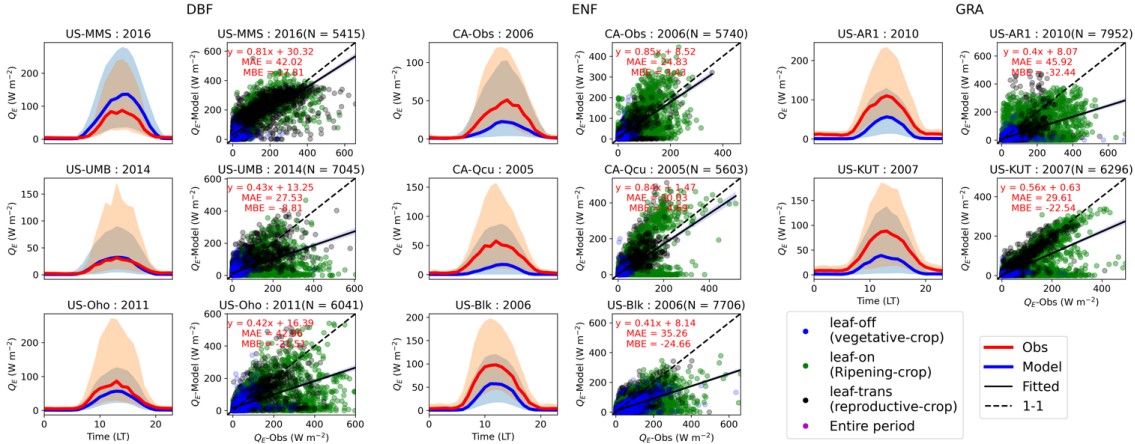


*Figure E1: As Fig. 15, but using the parameters from Table A1 in Ward et al. (2016).*

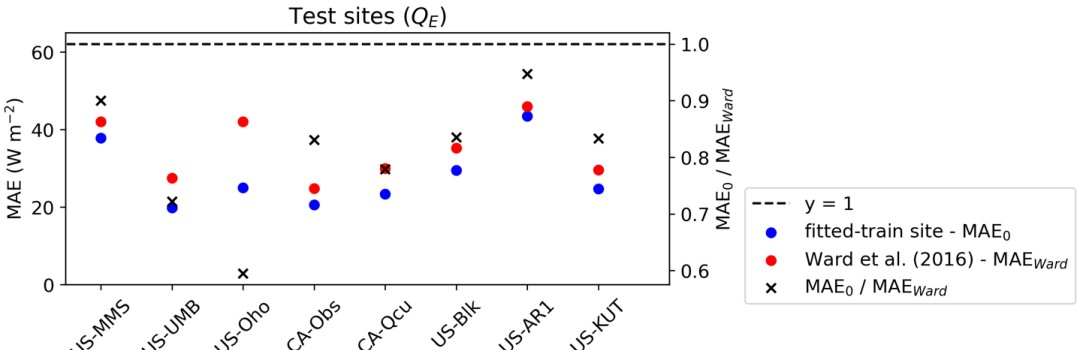


*Figure E2:* MAE *for* $Q_E$ *(Eq. 5) calculated with site-specific surface conductance parameters for sites*

586        *(Table 3) (i.e. Table 5, $MAE_0$) and with the Ward et al. (2016) parameters (their Table A1)*

587        *($MAE_{W16}$), and the ratio of $MAE_0$ and $MAE_{W16}$ for these sites.*

**Appendix F:** *Albedo for May to October period*

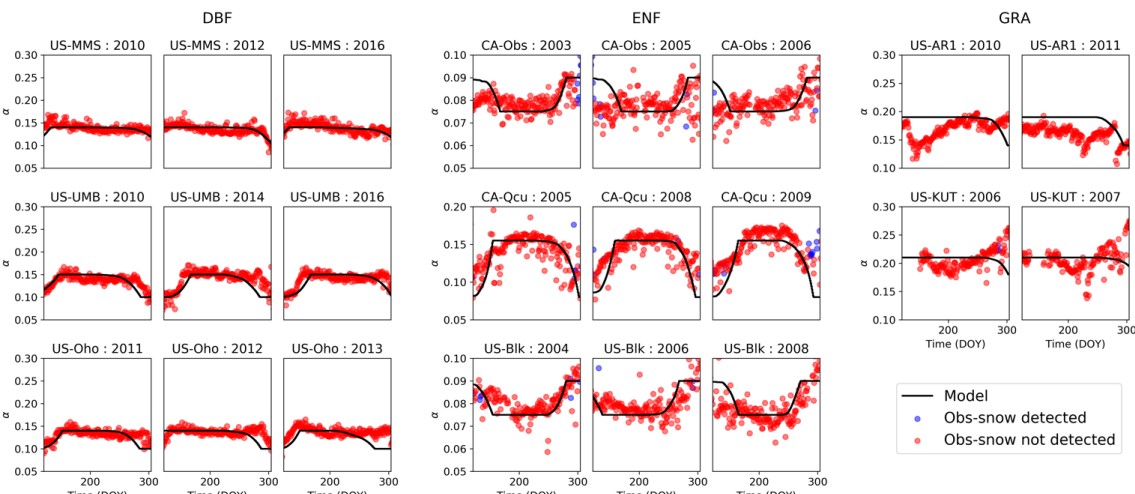

*Figure F1: As Fig. 10, but only for May–October period.*
**Code and data availability**
All source codes (Jupyter notebooks and Python scripts), input and output data are archived on
Zenodo (https://doi.org/10.5281/zenodo.3831233, Omidvar et al., 2020)
**Author contribution**
HO, TS and SG contributed to data preparation, model development, running simulations and
writing the paper. All other authors (DB, AB, JC, ZD, HI, and JM) provided data, interpreted the
results, and reviewed the manuscript.
**Competing interest**
The authors declare that they have no conflict of interest.
**Acknowledgments**
This work is funded by NERC-COSMA project (NE/S005889/1), Newton Fund Met Office CSSP-
China (SG), NERC Independent Research Fellowship (NE/P018637/1), National Natural
Science Foundation of China (41875013; Zhiqiu Gao and Zexia Duan). We thank PIs for
providing the data: Kim Novick and Rich Phillips (US-MMS); Christopher Gough, Gil Bohrer and
Peter Curtis (US-UMB); Hank A. Margolis (CA-Qcu); Tilden Meyers (US-Blk); James Bradford,
Margaret Torn (US-AR1); and Maricar Alberto, Caesar Arloo Centeno, Reiner Wassmann (PH-
IRI)



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

Index/FPAR 4-day L4 Global 500m SIN Grid V006. NASA EOSDIS Land Processes DAAC,
782    2015.

Nations, U.: 2018 revision of world urbanization prospects, 2018.
Nishihama, M., Wolfe, R., Solomon, D., Patt, F., Blanchette, J., Fleig, A. and Masuoka, E.:



MODIS Level 1A Earth Location: Algorithm Theoretical Basis Document By the MODIS Science
Data Support Team, Greenbelt, Md., 1997.
Noormets, A., McNulty, S. G., DeForest, J. L., Sun, G., Li, Q. and Chen, J.: Drought during
canopy development has lasting effect on annual carbon balance in a deciduous temperate
forest, New Phytol., 179(3), 818–828, doi:10.1111/j.1469-8137.2008.02501.x, 2008.
Nunez, M., Davies, J. A. and Robinson, P. J.: Surface albedo at a tower site in Lake Ontario,
Boundary-Layer Meteorol., 3(1), 77–86, doi:10.1007/BF00769108, 1972.
Offerle, B., Grimmond, C. S. B. and Oke, T. R.: Parameterization of Net All-Wave Radiation for
Urban Areas, J. Appl. Meteorol., 42(8), 1157–1173, doi:10.1175/1520-
0450(2003)042<1157:PONARF>2.0.CO;2, 2003.
Omidvar, H., Sun, T. and Grimmond, C. S. B.: Assets for SUEWS Parameters calculation, ,
doi:10.5281/zenodo.3831233, 2020.
Penman, H. L.: Natural evaporation from open water, hare soil and grass, Proc. R. Soc. Lond.
A. Math. Phys. Sci., 193(1032), 120–145, doi:10.1098/rspa.1948.0037, 1948.
Peters, E. B., Hiller, R. V. and McFadden, J. P.: Seasonal contributions of vegetation types to
suburban evapotranspiration, J. Geophys. Res. Biogeosciences, 116(1), G01003,
doi:10.1029/2010JG001463, 2011.
Philip Bloomington, R. and Novick Bloomington, K.: AmeriFlux US-MMS Morgan Monroe State
Forest, , doi:10.17190/AMF/1246080, 2016.
Porter, C. L.: An Analysis of Variation Between Upland and Lowland Switchgrass, Panicum
Virgatum L., in Central Oklahoma, Ecology, 47(6), 980–992, doi:10.2307/1935646, 1966.
Schmid, H. P., Grimmond, C. S. B., Cropley, F., Offerle, B. and Su, H. B.: Measurements of
CO2 and energy fluxes over a mixed hardwood forest in the mid-western United States, Agric.
For. Meteorol., 103(4), 357–374, doi:10.1016/S0168-1923(00)00140-4, 2000.
Shuttleworth, W. J.: A simplified one-dimensional theoretical description of the vegetation-
atmosphere interaction, Boundary-Layer Meteorol., 14(1), 3–27, doi:10.1007/BF00123986,
811  1978.

Shuttleworth, W. J.: Evaporation models in the global water budget., Var. Glob. water Budg.,
147–171, doi:10.1007/978-94-009-6954-4_11, 1983.





Spronken-Smith, R. A., Oke, T. R. and Lowry, W. P.: Advection and the surface energy balance
across an irrigated urban park, Int. J. Climatol., 20(9), 1033–1047, doi:10.1002/1097-
0088(200007)20:9<1033::AID-JOC508>3.0.CO;2-U, 2000.
Sun, T. and Grimmond, S.: A Python-enhanced urban land surface model SuPy (SUEWS in
Python, v2019.2): development, deployment and demonstration, Geosci. Model Dev, 12, 2781–
2795, doi:10.5194/gmd-12-2781-2019, 2019.
Sun, T., Järvi, L., Omidvar, H., Theeuwes, N., Lindberg, F., Li, Z. and Grimmond, S.: Urban-
Meteorology-Reading/SUEWS: 2020a Release, , doi:10.5281/zenodo.3828525, 2020.
Van Ulden, A. P. and Holtslag, A. A. M.: Estimation of atmospheric boundary layer parameters
for diffusion applications., J. Clim. Appl. Meteorol., 24(11), 1196–1207, doi:10.1175/1520-
0450(1985)024<1196:EOABLP>2.0.CO;2, 1985.
Ward, H. C., Kotthaus, S., Järvi, L. and Grimmond, C. S. B.: Surface Urban Energy and Water
Balance Scheme (SUEWS): Development and evaluation at two UK sites, Urban Clim., 18, 1–
32, doi:10.1016/j.uclim.2016.05.001, 2016.
Zhou, A., Qu, B.-Y., Li, H., Zhao, S.-Z., Suganthan, P. N. and Zhang, Q.: Multiobjective
evolutionary algorithms: A survey of the state of the art, Swarm Evol. Comput., 1(1), 32–49,
doi:10.1016/j.swevo.2011.03.001, 2011.