# Peer review of "Surface [Urban] Energy and Water Balance Scheme (v2020a) in non-urban areas: developments, parameters and performance"

_Geoscientific Model Development, 2020_

## Referee Comment (RC1) · Anonymous Referee #1 · 29 Jul 2020

**Review of GMD-2020-148**

**Surface [Urban] Energy and Water Balance Scheme (v2020a) in non-urban areas: developments, parameters and performance, by Omidvar et al.**

This work concerns use of the SUEWS model in non-urban areas. The manuscript includes some recent developments to the SUEWS model, some analysis of observed data from eddy covariance sites, estimation of SUEWS model parameters relevant to latent heat fluxes using the observed datasets and assessment of model performance for different time periods. This study includes many components, yet the overall purpose of the paper is unclear. Is it new model developments, new parameter values, a new method for parameter derivation or an assessment of model performance? It seems likely the authors would like to tackle several, if not all, of these aspects here, but lack of a clear structure makes the manuscript very hard to follow and, in my opinion, none of these aspects are covered in sufficient detail.

One of the main problems is that the manuscript is not well organised. It often reads more like notes than a journal article and it is very hard for the reader to follow what has been done and why. It is not necessary to have six appendices when the main text is only about 450 lines! The figures and tables should also be improved (many are not especially useful and there seems to be a lot of repetition).

This work would be of much greater use if the findings were analysed at a deeper level and set in context against the literature. In general, more evidence of awareness of the literature is necessary. As an example, the authors refer mainly to the body of work on the SUEWS model, in particular the recent model development papers of Järvi et al. (2011) and Ward et al. (2016). One question that arises is why the current study is needed at all, given that much of SUEWS was originally based on non-urban models and parameters. Here the authors take the recent 'urbanised' sub-models and parameters and seem to 'un-urbanise' them again.

More detailed suggestions for improvement are given below, along with questions about various parts of the methodology. Providing these are given consideration, I believe the manuscript can be substantially improved. I therefore recommend publication after major revisions.

**General comments**

**Structure and presentation**

Throughout the Methods and Results section, the various aspects of the study are mentioned interchangeably so it is not clear whether observations, model output, calibration or evaluation is being discussed. Currently the model description is spread throughout the Methods section.

- I suggest first adding a section where the model is described along with the equations, including the new developments currently in Appendix A (these seem quite important, especially for a GMD paper, and I'm not sure why these are in the appendix whereas LAI is in the main text). For this to be a standalone publication, more general details about the SUEWS model also need to be given (e.g. what are the required inputs, what are the outputs, what scale does the model operate at). This section should simply describe what the model does, without including any methodological details about the parameterisation or evaluation approaches in this study.
- The readability of the section describing the observations should be improved so that the reader quickly gets an overview of the sites and starts to feel familiar with their different characteristics. For example, mentioning the site names in the text, not just the table, and

giving a very brief description. It would also help if the abbreviations of the sites contained the land cover code instead of the country (which is relatively unimportant). Some details about quality control of observed data should be given. It would also be useful here to mention the representativeness of the site years (e.g. the low rainfall mentioned in L342).
- Then there should be the section on how each of the model parameters were derived. Seeing as this is a key part of the manuscript, more details are needed about how these parameters were fitted. It is currently not at all clear how this was done (using multiple SUEWS runs with various parameter values and minimising the MAE?). What range of parameter values was considered? Was more than one parameter allowed to vary at once to allow for interdependencies? Was any bootstrapping done?
- There also needs to be a section describing the approach used for the SUEWS runs, e.g. spin-up, initial conditions, forcing variables used.

Having so many appendices for a short paper is not helpful. Much of the material in the appendices is not useful anyway and should be edited as strictly as in the main text. Here are some suggestions.

- Appendix A seems to be important and should be in the main text (in the new model description section). If I understood correctly, SUEWS now calculates Q* based on Ts which is based on the sensible heat flux and MOST. This has the potential to cause large errors in Q*, especially in SUEWS since the sensible heat flux is calculated as the residual of the energy balance, and also because of the poor performance of MOST and all the uncertainties of the roughness length for heat, etc. The effect of this change needs to be shown here, and makes it more important that the model's ability to calculate Q* is dealt with.
- Appendix B: this is probably appropriate as an appendix but needs to be rewritten as it is not at all clear what has been done here. Start with one or two sentences describing the purpose of this analysis. It needs to be made clear that this section concerns observed data (not SUEWS). How do the results obtained compare to rule-of-thumb values? How do the roughness length and displacement height values compare to the literature? What do the lines in Fig B1 represent? The discussion of fetch in L523-527 is very unclear and needs rewriting. Make the points smaller and axes ticks consistent in Fig B2.
- Appendix C should be moved to the appropriate place in the main text. Perhaps adding boxplots for different temperature bins would help support your decision. I would suggest rephrasing as there does not seem to be a point where evaporation 'switches off' and three of the sites have very little data below the suggested cut-off. It should be made clear that this is observed data, not model output.
- Appendix D: what do the authors want the reader to take away from these three tables? L564-565 makes no sense. Suggest deleting.
- Appendix E: as explained before, this comparison does not make sense. The Ward et al. (2016) parameters were derived for bulk urban surfaces, and were not intended to be used for non-urban areas. Suggest deleting.
- Appendix F: What does this plot add to what is already shown (and more) in Fig 10? Suggest deleting.

In addition, general readability could be improved by:

- using fewer cross-references: the reader has to work very hard to follow the text when we are constantly directed to Equation/Table/Figure/Appendix X. Use cross-references where necessary and helpful, but try to ensure the reader knows what variable/site you're talking about;

- avoiding vague language; instead specify what you mean (particularly with respect to this study versus previous studies and what is generally true/what is true in the model/what is done here);
- using more words so that the text flows more naturally and is therefore more easily understandable to the reader. This is particularly true for the table and figure captions, many of which don't really make sense;
- don't include key methodological information in captions instead of the text (e.g. L381-382).

A lot of space (Figures and Tables) is given to presenting MAE, MBE and nMAE for the different sites at different times and different states of vegetation. However, there is little insight gained and very little discussion in the text. I therefore suggest removing these figures and, if necessary, compiling the information into a single figure or table

**Non-urban SUEWS in context**

The background of this work seems to be the SUEWS model – i.e. an urban land surface scheme that has been developed by 'urbanising' sub-models developed over non-urban environments. The latent heat flux calculation is based on the Penman-Monteith equation with the Jarvis formulation of the surface conductance. This work seems to 'start' from SUEWS and then 'un-urbanise' the equations again by setting the anthropogenic heat flux to zero and fitting parameters for non-urban sites. In many places, the manuscript needs adjusting to reflect that these non-urban forms exist – and in fact existed long before SUEWS!

Not only is this acknowledgement missing in the model description, but also in the Introduction, Results and Conclusion. In the Introduction, the motivation for this work needs to be set in the wider context of land surface modelling – i.e. at least a paragraph describing previous work that has been done on albedo, LAI and evaporation in forest, grassland, agricultural environments and for water and bare soil surfaces too. In the Results section, the results obtained here should be compared to previous results obtained in some of these previous non-urban studies. In Table 6, how do the non-SUEWS specific parameters (albedo, LAI, roughness length and displacement height) obtained here compare to the body of literature over the last few decades and why should future users of SUEWS use the values presented here instead of those in the literature? Are the roughness length and displacement height values given in Table 6 really useful? Wouldn't it be more reasonable to use the rule of thumb relating these parameters to vegetation height at the site?

For SUEWS applications in the urban environment, Järvi et al. (2011) and Ward et al. (2016) derived parameters for the surface conductance using datasets collected in urban areas with the aim of better capturing latent heat fluxes in urban environments. The comparison using the Ward et al. (2016) parameter values therefore does not make sense at these urban sites, as those parameter values were never intended to be used for non-urban surfaces! It would make more sense to compare the current results to those derived over non-urban surfaces, such as Ogink-Hendriks (1995) or Stewart (1988) over forest.

A comparison of SUEWS model performance over these non-urban surfaces with previously published results of SUEWS model performance over urban surfaces could also be useful.

**Depth of analysis**

The analysis/interpretation is generally superficial and needs to be developed substantially. For example, in L353-355 the timing of the decrease in LAI for wheat is much worse than for rice but this is not discussed. In Fig 10 (and even Fig F1), the results for grass seem to show quite poor agreement

but this is not discussed in the text at all. For a more complete paper and, crucially, to avoid drawing misinformed conclusions, the observed data must also be analysed. What is the explanation for the observed variation in the albedo of grassland (are the data even reliable)? How is the variation in albedo at US-AR1 related to the variation in LAI – more explanation is required, i.e. what is the mechanism proposed behind the low rainfall in 2011 mentioned in L341-343? Why are the results for water and bare soil not shown? What is the reason for the very different annual cycles for evergreen trees seen in Fig 10. Why are the conductance parameters for the individual sites not shown (L438)? These could potentially provide some important insight into the robustness of the results. Some physical interpretation of the G2-G6 values should be attempted (i.e. why did the fitting procedure result in these values and what does it tell us?).

**Specific comments**

The Abstract is quite vague and suggests some topics will be analysed more deeply than they are. Some suggestions:

- add 'Here' at the start of the sentence on L34 to make clear this is what you did in this study and is not a general feature of SUEWS
- add 'from multiple sites' to the end of this sentence
- meaning of 'guidance to apply SUEWS are provided' (L38) is unclear
- L39: 'impacts' on what?
- 'The relation between LAI and albedo is explored' (L39-40) – I'm not sure this is really covered in the analysis
- Add 'in the model' after 'captured' in L44
- L44-45: the meaning is unclear and there is no discussion in the manuscript of how latent heat fluxes affects modelled canopy-layer air temperature.

The treatment of snow cover is rather strange. Judging from Fig 10, either the detection of snow cover is not appropriate or the modelled albedo does not capture the true seasonal variability in vegetation characteristics. As snow cover is not considered in this work, perhaps additional sites with much less snow during leaf-off periods would be more valuable.

Why are albedo and LAI parameters derived for each site but surface conductance parameters derived for each land cover type? Analysis of the variation in conductance parameters between sites would be informative and may help to inform about applicability to other sites.

The paper is missing a balanced consideration of shortcomings of this study. For example,

- Would these non-urban parameters be expected to be appropriate for e.g. deciduous trees in residential areas (L66) given the possibility of increased urban temperatures or advective effects?
- Considering the importance placed on seasonal variation the impact of assuming constant OHM coefficients should be addressed.
- By not including snow cover much of the leaf-off periods are not useable, and the significance of accurately capturing seasonal variation in LAI is reduced. If snow cover is not addressed here, these sites seem like a strange choice.
- Previous shortcomings of LSMs have highlighted that parameters derived for particular conditions can lead to bias in model results at other sites. Although this study aims to provide new, generalised parameter values the range of sites used for each land cover type is not very large and does not cover a large geographical area. The limitations of this should at least be mentioned.

- Keep in mind the appropriateness of the statistics used when comparing different sites and different seasons, where the variables may have different magnitudes and different amounts of data available. Depending on the analyses that end up in the revised paper, this may not be an issue, but it is important to consider. Was any consideration given to other statistical measures (e.g. the correlation coefficient would indicate how well the model reproduces the variability, even if the magnitude is wro)?

No uncertainty estimates are provided making it very difficult to judge the robustness and accuracy of these results. This should be rectified in the revised manuscript.

**More minor comments**

L56-58: Why only mention anthropogenic heat and water here? Urban LSMs have many more additions than this.

L62: make it clear that these land covers can occur in a single grid. Somewhere the intended grid size for SUEWS needs to be mentioned.

L64, 67: what is meant by 'integrated'?

L70: Here would be a good place to bring in some of the non-urban literature.

L71 'bridge this gap' – you have not really talked about a gap so this doesn't really make sense.

L76-79: This overview does not make sense (perhaps reflecting the confused structure)!

L93: 'phenology changes key model parameters' – changes what to what?

L93-102: these paragraphs do not really make sense. Suggest rephrasing and incorporating at the appropriate stage in the model description section

Table 1: I personally find the last 3 columns unhelpful. There are details missing from the Definition column (GDD abbreviation, 'coefficients' alone is not informative, 'shortwave radiation', OHM) but possibly this table could be deleted as everything should be defined and explained in the text anyway.

L163: what is t?

L165-169: This methodology should be separated from the model description and more details should be added, including what timestep and what type of regression. What is the justification for ignoring variation in these OHM coefficients, particularly for this paper on seasonal variation – was any observed? At least a couple of sentences with references should be added here.

L181: measurement height for wind speed appears to be Hu in L271

L187: 'stability scale' → 'stability parameter'

L201: 'Phenological state is critical' – what is the justification for this statement? Presumably all functions are important if they are not correctly parameterised.

L120: It may be easier to follow if this new LAI equation was presented along with Fig 5 so the reader understands why a different parameterisation is needed. Some justification for this equation would be helpful.

L135: In reality or in the model?

L141-142: Meaning unclear. Which 'model parameters'? How does this paragraph fit with L148-154?

L143: And also the longwave radiation components

Table 2: Without discussion in the text this table is not much use. The caption does not make sense. Why were OHM coefficients for some surfaces derived here and others from the literature?

L218: How was it ensured that the surface was dry?

L253-263: Table 3 is vaguely referenced 4 times here. Why not provide the appropriate information in the text if necessary?

Table 3: Why are the DOIs given here? They should be included in the Reference list.

Table 3: How were the study years chosen and then how was it decided which to use for calibration and which for testing? Without information, a cynical reader may suspect the combination was selected which gives the best results…

Figure 2: This figure does not seem very useful. Suggest deleting unless it is better described in the text and provides more insight.

L334-334: Units of LAI missing

L356-359: Meaning unclear.

Figure 7, 12, and 14 take up a lot of space, are not very easy to read and are hardly discussed in the text. Perhaps a more useful way to summarise this information could be found, or only the most relevant results displayed. The same goes for Figure 9d and Fig 11d. Note it is frustrating for the reader to have to adjust to essentially reading the same information as Fig 7 and 12 but presented in a different way.

Panel labels are missing from Fig 9 and 11.

Would it make sense to merge Fig 6 and 9a-c and Fig 10 and 11a-c so that these similar results are presented in a more comparable way? I would suggest perhaps even making the x-axis all one year long so the difference for the crops can be seen more immediately. Please use smaller points.

Table 4: Tplant, GDDv and GDDLAImax would be better as separate columns.

Fig 12 and L403: the MAE and MBE seem small for the grassland sites considering the results shown in Fig 10. Please check.

L427-429: Where is the justification for this statement? Was the performance of other fluxes checked (e.g. net radiation, storage heat flux)? If these fluxes are not modelled adequately, wouldn't they result in inappropriate conductance parameters being derived (e.g. parameters which are tuned to give the right results for the wrong reasons)?

L430: More explanation needed here. Also Q* needs to be reasonably accurate. The assumptions in the resistance approach and the uncertainties in the roughness parameters should be discussed too. The assumption of homogeneous fetch requires more explanation if it is included here.

L433-436: See above for explanation of why this comparison does not make sense.

Fig 13: Difficult to read (make full-page?). Use smaller points. The annual diurnal pattern is of limited use considering the huge seasonal variation which makes the large interquartile ranges. Consider

using daily or (if data availability is an issue) monthly evaporation totals instead, which may allow insight into when the latent heat flux is modelled well and when not.

L456-457: This cannot be concluded here as it is not demonstrated in the paper. This hypothesis is suggested as 'a possible explanation' in L341-344 but no other possibilities are discussed and there is no further analysis to substantiate or contradict this suggestion.

L459: It should be stated somewhere that (presumably) to obtain fitted parameters for a specific site observations must be available.

L558: 'Given this' – given what?

There are also a few typos that would need to be corrected at a later stage.

---

## Referee Comment (RC2) · Anonymous Referee #2 · 21 Aug 2020

Review - Surface [Urban] Energy and Water Balance Scheme (v2020a) in non-urban areas: developments, parameters and performance

The manuscript aims to extent the SUEWS model to non-urban surfaces, with the overall goal to estimate the energy-balance fluxes in such areas. Therefore, specific parameters used to estimate the surface heat fluxes are inferred from observational data sampled at energy balance stations, which includes different vegetation types and different climate zones. The modelled surface fluxes with SUEWS were compared against observational data to evaluate the performance of the SUEWS model over rural surfaces. The topic of the paper itself fits well into the journal and is of interest to the

research community, especially since reliable input of phenological data and surface data play a key role for reliable estimation of the surface-energy balance components in models. However, after extensive review, I cannot recommend the manuscript for publication until major revisions have been done and extended analysis is presented. My major concerns are outlined in the following.

Major comments

(A) You added new methods and tuned parameters to model impervious surfaces in rural areas. However, the description of newly developed parameter estimation such as for LAI or albedo is mixed with parts of model description, so that it is hardly possible to extract what is new and what has been there already before. I would recommend to first described the state-of-the-art model and describe newly developed approaches separately. Also, the manuscript provides no condense model description of SUEWS but refers to previous papers. The manuscript itself should be readable as a stand-alone paper. Hence, even though not all details need to be brought-up, the manuscript needs to provide a proper overview of the model at one place. Further, please give all information concerning model description in the text, not within the appendix.

(B) The manuscript is sometimes hard to follow due to missing logical order between sentences. In several sections, sentences appear to be disconnected from each other rather than indicating a logical order. As a consequence the text reads more like a collection of notes.

(C) The discussion of the results is not sufficient and lacks important aspects. For example, why is the bias error positive for some sites but negative for others. The authors provide the errors for all sites, but do not try to put these within the context of site-specific information. Also, one of the main problems of eddy-covariance measurements is the non-closure of the energy balance. Especially for the comparison of surface latent heat fluxes this needs to be discussed.

In this context, the manuscript need to provide also more information about the specific

EC sites. At EC stations located in heterogeneous landscapes the measured fluxes are a mixture of signals emerging at different land-surface types rather than only one type (as assumed in this study), i.e. the footprint of the stations covers several land surfaces with different properties (LAI, roughness). As a consequence the value f_i (which is assumed to be 1 in this study) is not necessarily one. To be able to evaluate the validity of the inferred parameters in this study, site specific information should be provided, e.g. the degree of surface heterogeneity, which in turn need to be correlated to the overall error in the surface latent heat flux for the individual sites.

Minor Comments

55-56: You mention that there is a number of LSM's, but you cite only one. 54-58: In my opinion this leads the reader on a wrong track, the manuscript focuses on non-urban rural sites. 63: I guess you mean "around the globe". 66: The word parameters is unclear at that point and need to be specified. Do you mean certain (bio)physical quantities such as leaf-are densities, surface or material properties, or do you mean certain values used in parametrizations? 71: Which gap does the authors mean? Please be more specific. 96: It is unclear to what does "The former" refer to. 98-99: "Model parameters ...": As a stand alone sentence this makes sense, though it becomes not directly apparent to the reader what is exactly meant. However, from this there is not obvious connection to the following sentence. With changes of the key parameters you may describe any type of vegetation, but how is this related to the statement that parameters need to be consistent? 107:108: How are GDD and SDD defined? Are these vegetation-type specific?

Eq. 3: Does the index i includes all vegetation types including or excluding crops? Eq. 3: Is LAI_max/min a function of the time of the year? If this is the case, please indicate this somehow within the equation or text. 115-116: Where does these max/min values come from? Here, a reference is required in the text. 121: The note within the parenthesis is unclear to me, how are shorter / longer LAI_max times are reflected in Eq. 3? 175/189: I guess you mean water vapor. 199/200: The authors should

elaborate why removing G1 from the first term is a valid approach. According to the text it sounds to be an arbitrary decision, though I assume there is a specific reason for this? 200-201 and following: This is not really a sentence but more a note. Also, the following sentences sound more like a note. 204: To be specific, soil moisture deficit is not really a meteorological quantity. Eq. 13,14,15: $G_2$, $G_3$, $G_4$, $G_5$ are not defined in the text. 240: Parameters itself cannot have a performance. What you mean is the performance of SUEWS using parameters for non-urban surfaces. 246: What do the authors mean with surface state? 285: What do the authors meas with "are not completely independent": among each other? 347-349: In Fig. 6 the authors show the LAI distribution over the year. It does not become clear how this indicates that a constant LAI would lead to poor radiation and surface fluxes. If the authors see a link between these two things it should be given there. Fig. 6: The LAI variation for the evergreen-tree sites is surprisingly high. The minimum LAI values for the respective Canadian sites are similar compared to the deciduous-tree sites. For evergreen trees I would expect a rather time-constant value, while here also the MODIS values indicate almost zero LAI. Could the maybe connected to snow cover on trees? Fig 7b: MBE indicates that the modelled LAI values are biased towards smaller values (not for all sites, but for many), especially during the leaf-on period. However, I miss some discussion about this in the text (line 338-345). Fig 12: Please provide a full description what is shown in the figure. To switch between the figures to find out what is shown makes the figure hardly readable.

Most of the equations: Punctuation is missing.

---

## Author Comment (AC1) · 4 Nov 2021

We thank the reviewers and editor for their constructive comments on our paper and appreciate the helpful suggestions.

This paper is concerned with developing workflows to derive parameters and the evaluation of the resulting model performance; we have clarified this in the revision and improved the paper in the following aspects:

- Expansion of data sources analysed to the 38 sites from the now available FLUXNET2015 dataset.
- Expansion of surface conductance parameters evaluation to include the both the FLUXNET2015 site derived values but also the literature-based values of NOAH (a popular land surface model used in NWPs, e.g., WRF).
- Only vegetated areas are analysed in three groups: evergreen and deciduous trees, grass (including crops). Bare soil and water are no longer included in this work as less data are available to generalise findings compared to other land cover types in an appropriate way.
- Generalised workflows are provided to derive model parameters with variability/uncertainties (e.g. standard deviations and/or inter-quantile ranges).

Our responses below refer to the **new** Section/Figure/Table/Appendix in the revised manuscript unless otherwise indicated. Given we have made substantial changes to the paper some comments are no longer applicable, so we indicate as N/A. Some comments are applicable, elsewhere in the revised manuscript, and we have taken them into account.

**Reviewer 1**

**General Comments:**

1. *This work concerns use of the SUEWS model in non-urban areas. The manuscript includes some recent developments to the SUEWS model, some analysis of observed data from eddy covariance sites, estimation of SUEWS model parameters relevant to latent heat fluxes using the observed datasets and assessment of model performance for different time periods. This study includes many components, yet the overall purpose of the paper is unclear. Is it new model developments, new parameter values, a new method for parameter derivation or an assessment of model performance? It seems likely the authors would like to tackle several, if not all, of these aspects here, but lack of a clear structure makes the manuscript very hard to follow and, in my opinion, none of these aspects are covered in sufficient detail.*

2. *One of the main problems is that the manuscript is not well organised. It often reads more like notes than a journal article and it is very hard for the reader to follow what has been done and why. It is not necessary to have six appendices when the main text is only about 450 lines! The figures and tables should also be improved (many are not especially useful and there seems to be a lot of repetition).*

3. *This work would be of much greater use if the findings were analysed at a deeper level and set in context against the literature. In general, more evidence of awareness of the literature is necessary. As an example, the authors refer mainly to the body of work on the SUEWS model, in particular the recent model development papers of Järvi et al. (2011) and Ward et al. (2016). One question that arises is why the current study is needed at all, given that much of SUEWS was originally based on non-urban models and parameters. Here the authors take the recent 'urbanised' sub-models and parameters and seem to 'un-urbanise' them again.*

*4. More detailed suggestions for improvement are given below, along with questions about various parts of the methodology. Providing these are given consideration, I believe the manuscript can be substantially improved. I therefore recommend publication after major revisions.*

Our focus is the determination of parameters for use fully vegetated areas that commonly occur adjacent to cities.

We have restructured the paper and removed the non-vegetated land covers to simplify the storyline.

- The original parameters in SUEWS for conductance were all based on urban environments, so the analysis of the non-urban parameters is all new.
- More discussions on relevant literature have been added along with expanded analysis.

**Major Comments:**

*5. Throughout the Methods and Results section, the various aspects of the study are mentioned interchangeably so it is not clear whether observations, model output, calibration or evaluation is being discussed. Currently the model description is spread throughout the Methods section.*

*a) I suggest first adding a section where the model is described along with the equations, including the new developments currently in Appendix A (these seem quite important, especially for a GMD paper, and I'm not sure why these are in the appendix whereas LAI is in the main text). For this to be a standalone publication, more general details about the SUEWS model also need to be given (e.g. what are the required inputs, what are the outputs, what scale does the model operate at). This section should simply describe what the model does, without including any methodological details about the parameterisation or evaluation approaches in this study.*

The new paper structure is:

- Section 2: Vegetation related physics in SUEWS.
- Section 4: Model parameters derivation workflows

*b) The readability of the section describing the observations should be improved so that the reader quickly gets an overview of the sites and starts to feel familiar with their different characteristics. For example, mentioning the site names in the text, not just the table, and giving a very brief description. It would also help if the abbreviations of the sites contained the land cover code instead of the country (which is relatively unimportant). Some details about quality control of observed data should be given. It would also be useful here to mention the representativeness of the site years (e.g. the low rainfall mentioned in L342). Then there should be the section on how each of the model parameters were derived. Seeing as this is a key part of the manuscript, more details are needed about how these parameters were fitted. It is currently not at all clear how this was done (using multiple SUEWS runs with various parameter values and minimising the MAE?). What range of parameter values was considered? Was more than one parameter allowed to vary at once to allow for interdependencies? Was any bootstrapping done?*

The new paper structure includes:

- Section 3: Datasets used
- 3.1 FLUXNET2015 - by using one source for the flux data for simplicity.

    Also, we keep site ID in the format "country-site" to be consistent with the FLUXNET naming convention as it is widely adopted in FLUXNET-related studies.
- 3.2 MODIS for LAI
- 3.3 SoilGrids for soil properties
- Section 4: Model parameters derivation workflows

*6. There also needs to be a section describing the approach used for the SUEWS runs, e.g. spinup, initial conditions, forcing variables used.*

The new paper structure includes:

- Section 5.1: Model configurations.

7. *Having so many appendices for a short paper is not helpful. Much of the material in the appendices is not useful anyway and should be edited as strictly as in the main text. Here are some suggestions.*

   a) *Appendix A seems to be important and should be in the main text (in the new model description section). If I understood correctly, SUEWS now calculates Q\* based on Ts which is based on the sensible heat flux and MOST. This has the potential to cause large errors in Q\*, especially in SUEWS since the sensible heat flux is calculated as the residual of the energy balance, and also because of the poor performance of MOST and all the uncertainties of the roughness length for heat, etc. The effect of this change needs to be shown here, and makes it more important that the model's ability to calculate Q\* is dealt with.*

   N/A: removed in the revised manuscript as this is an experimental development and not used in this work.

   b) *Appendix B: this is probably appropriate as an appendix but needs to be rewritten as it is not at all clear what has been done here. Start with one or two sentences describing the purpose of this analysis. It needs to be made clear that this section concerns observed data (not SUEWS). How do the results obtained compare to rule-of-thumb values? What do the lines in Fig B1 represent? The discussion of fetch in L523-527 is very unclear and needs rewriting. Make the points smaller and axes ticks consistent in Fig B2.*

   Remains as Appendix B but with the following updates:

   - New analysis using the FLUXNET2015 dataset.

   - Text rewritten and figures updated.

   - New text (Appendix B, L796–L801)

     Using the derived $z_{0m}$ and $z_d$, $f_0$ and $f_d$ parameters can be obtained (Eqn. 9 and 10). These is considerable intra-PFT variability of both $f_0$ and $f_d$ (Fig. B1). There are also intra-site variations associated with varying $H_c$. Given the large variability in both $f_0$ and $f_d$, the rule-of-thumb approach would incur large bias in estimated aerodynamic and surface resistances and subsequently the modelled $Q_E$. To reduce such bias, in the evaluation of the other sub-models and parameter determinations in this paper, we use the derive $z_{0m}$ and $z_d$ determined for each vegetation stage and site.

[Figure]

**Figure B1**. Relations between canopy height ($H_c$) and a) roughness length for momentum ($z_{0m}$, Eqn. B2) and b) displacement height ($z_d$, Eqn. B3) for different vegetation stages based on LAI (see Sect. 4.1 for classification details)

8. *Appendix C should be moved to the appropriate place in the main text. Perhaps adding boxplots for different temperature bins would help support your decision. I would suggest rephrasing as there does not seem to be a point where evaporation 'switches off' and three of the sites have very little data below the suggested cut-off. It should be made clear that this is observed data, not model output.*

9. *Appendix D: what do the authors want the reader to take away from these three tables? L564-565 makes no sense. Suggest deleting.*

10. *Appendix E: as explained before, this comparison does not make sense. The Ward et al. (2016) parameters were derived for bulk urban surfaces, and were not intended to be used for non-urban areas. Suggest deleting.*

11. *Appendix F: What does this plot add to what is already shown (and more) in Fig 10? Suggest deleting.*

New paper structure:

- Old Appendices C-F removed.
- New Appendix C added but with different material: now Matsumoto et al.'s (2008) upper-boundary-based method (Sect. 4.3) is adopted to determine the surface conductance related parameters; here we report detailed site-level values derived for SUEWS parameters.

12. *In addition, general readability could be improved by:*

a) *using fewer cross-references: the reader has to work very hard to follow the text when we are constantly directed to Equation/Table/Figure/Appendix X. Use cross-references where necessary and helpful, but try to ensure the reader knows what variable/site you're talking about.*

b) *avoiding vague language; instead specify what you mean (particularly with respect to this study versus previous studies and what is generally true/what is true in the model/what is done here).*

*c) using more words so that the text flows more naturally and is therefore more easily understandable to the reader. This is particularly true for the table and figure captions, many of which don't really make sense.*

*d) don't include key methodological information in captions instead of the text (e.g. L381-382).*

Done as suggested.

*13. A lot of space (Figures and Tables) is given to presenting MAE, MBE and nMAE for the different sites at different times and different states of vegetation. However, there is little insight gained and very little discussion in the text. I therefore suggest removing these figures and, if necessary, compiling the information into a single figure or table*

New Sect. 5.2–5.4 (which includes new Fig. 14–24) are undertaken to address this:

- Bias attribution: a new Sect. 5.2 is added in the revised manuscript to analytically attribute the bias in modelled $Q_E$ to different parameter contributors using a sensitivity analysis framework by McCuen (1974), the results of which indicate surface conductance $g_s$ is critical to the model performance in $Q_E$ prediction.

- Impact of $g_s$ parameters: given the importance of $g_s$ suggested by the above analysis (Sect. 5.2), we have compared the model performance by simulations with two sources of $g_s$ parameters – FLUXNET- and NOAH-based values – to examine their impacts (Sect. 5.3), which indicate the better model performance using FLUXNET-based values in particular at finer temporal scales (monthly and hourly) compared to the NOAH-based ones.

- Site-scale performance and key determinants: moreover, we have chosen sites of each PFT with best and poorest performance to understand the causes (Sect. 5.4) and found that correct prediction of LAI timing has a crucial influence on overall performance.

*14. The background of this work seems to be the SUEWS model – i.e. an urban land surface scheme that has been developed by 'urbanising' sub-models developed over non-urban environments. The latent heat flux calculation is based on the Penman-Monteith equation with the Jarvis formulation of the surface conductance. This work seems to 'start' from SUEWS and then 'un-urbanise' the equations again by setting the anthropogenic heat flux to zero and fitting parameters for non-urban sites. In many places, the manuscript needs adjusting to reflect that these non-urban forms exist – and in fact existed long before SUEWS!*

We did not intend to imply that non-urban parameters did not exist. Text has been changed (Sect. 1, L76–L86)

> Central to the SUEWS biophysics, is the Penman-Monteith approach (Penman 1948; Monteith 1965) with a Jarvis-type (Jarvis 1976) surface moisture conductance (Grimmond and Oke 1991). Despite various parameters having been derived to account for different urban areas (e.g. land cover differences) and regions (e.g. high/mid-latitude) to allow for changing phenology, conductance and storage heat flux related parameters (e.g. Järvi et al. 2011, 2014; Ward et al. 2016), urban parameter estimates are lacking partly because of limited observations and lack of a standard workflow for deriving parameters. Other land surface schemes have parameters for a wide range of plant functional types (PFT) (e.g. NOAH within WRF, Chen et al. 1996, Chen and Dudhia 2001) but are often derived from a small number of observational sites and their widespread applicability is unexamined. For example, NOAH largely adopted values from the HAPEX-MOBILHY observational program (Andre et al. 1986) following Noilhan and Planton (1989).

*15. Not only is this acknowledgement missing in the model description, but also in the Introduction, Results and Conclusion. In the Introduction, the motivation for this work needs to be set in the wider context of land surface modelling – i.e. at least a paragraph describing previous work that has been done on albedo, LAI and evaporation in forest, grassland, agricultural environments and for water*

*and bare soil surfaces too. In the Results section, the results obtained here should be compared to previous results obtained in some of these previous non-urban studies. In Table 6, how do the nonSUEWS specific parameters (albedo, LAI, roughness length and displacement height) obtained here compare to the body of literature over the last few decades and why should future users of SUEWS use the values presented here instead of those in the literature?*

Section 1 motivation:

- Necessity of using the same modelling framework for urban-rural comparison (L66–L70):

  As SUEWS v2020a (Tang et al. 2021) can diagnose near surface meteorology in the roughness sub-layer and canopy layer (e.g., air temperature and humidity at 2 m agl (above ground level), wind speed at 10 m agl), it is essential to ensure that any urban-rural comparison in these diagnostics has the proper rural skill and parameters (i.e. values used in parameterisations).

- Requirement by WRF-SUEWS coupling (L72–L75):

  With plans to couple SUEWS to a meso-scale model (e.g. Weather Research and Forecasting (WRF), Skamarock and Klemp (2008)), most regions have extensive areas that have completely pervious grid cells. As these need to be simulated using a consistent surface scheme, it is essential to have appropriate parameters for these areas.

- Necessity of examining the widely used surface conductance parameters using more recent observations: Text modification as indicated in previous response to R1C14.

Comparison in model parameters between this and previous studies are now added:

- Albedo (Sect. 4.1, L357–L358):

  see Cescatti et al. 2012 for a detailed analysis of albedo dynamics at FLUXNET sites.

- OHM coefficients (Sect. 4.2, L410–L412):

  In addition to the values derived here, we note that more detailed $\Delta Q_S$ observations are available for vegetated sites to derive such OHM coefficients (e.g. McCaughey (1985), Oliphant et al. (2004))

- Surface conductance related parameters (Sect. 4.3, L455–L458):

  The $g_{max}$ results are consistent with Hoshika et al. (2018) in terms of inter-PFT magnitude (Grass > EveTr and DecTr). The grass and crop values are comparable (Table C3) to Hoshika et al. (2018). However, our derived deciduous trees values are smaller (22 cf. 31 mm s$^{-1}$) and EveTr values larger (20 cf. 12 mm s$^{-1}$).

16. *Are the roughness length and displacement height values given in Table 6 really useful? Wouldn't it be more reasonable to use the rule of thumb relating these parameters to vegetation height at the site?*

We now use our observation derived roughness length and displacement height by phenology state (dormant, growing, peak and senescence as detailed in Sect. 4.1) – instead of the rule-of-thumb approach – to remove the additional source of uncertainty in the $Q_E$ simulations, because the relational ratios show large intra-PFT variability (even the same site with varying height; see Fig. B1, reproduced above under response to R1C7b) and may lead to considerable bias in modelled aerodynamic resistance if using the rule-of-thumb approach. This has been discussed in Appendix B (L798–L801):

  Given the large variability in both $f_0$ and $f_d$, the rule-of-thumb approach would incur large bias in estimated aerodynamic and surface resistances and subsequently the modelled $Q_E$. To reduce such bias, in the evaluation of the other sub-models and parameter determinations in this paper, we use the derive $z_{0m}$ and $z_d$ determined for each vegetation stage and site.

17. *For SUEWS applications in the urban environment, Järvi et al. (2011) and Ward et al. (2016) derived parameters for the surface conductance using datasets collected in urban areas with the aim of better capturing latent heat fluxes in urban environments. The comparison using the Ward et al. (2016) parameter values therefore does not make sense at these urban sites, as those parameter values were never intended to be used for non-urban surfaces! It would make more sense to compare the current results to those derived over non-urban surfaces, such as Ogink-Hendriks (1995) or Stewart (1988) over forest.*

Discussion added in Sect. 4.3 (L424–428):

> As the Jarvis-type formulation of stomatal/surface conductance is widely used for many land cover types, many parameter sets exist (e.g. Stewart 1988; Grimmond and Oke 1991; Ogink-Hendriks 1995; Wright et al. 1995; Bosveld and Bouten 2001; Järvi et al. 2011). Hoshika et al.'s (2018) comprehensive meta-analysis of published Jarvis-type stomatal conductance parameter values includes major woody and crop plants broadly similar to PFTs examined here.

Sect. 5.3: comparison changed to use the PFT-specific NOAH values (Appendix A) with the FLUXNET-based $g_s$ parameters derived in the paper. Other comparison removed.

18. *A comparison of SUEWS model performance over these non-urban surfaces with previously published results of SUEWS model performance over urban surfaces could also be useful.*

See response to R1C17.

19. *The analysis/interpretation is generally superficial and needs to be developed substantially.*

Additional analysis includes:

- Variability in the derived parameters (Sect. 4).
- Bias attribution of modelled $Q_E$ using an analytical framework (Sect. 5.2).
- Model performance in $Q_E$ prediction at both intra-annual and sub-daily scales (Sect. 5.3 and 5.4).

20. *For example, in L353-355 the timing of the decrease in LAI for wheat is much worse than for rice but this is not discussed.*

N/A - as sites changed to a consistent dataset and crop-specific work removed.

21. *For a more complete paper and, crucially, to avoid drawing misinformed conclusions, the observed data must also be analysed. What is the explanation for the observed variation in the albedo of grassland (are the data even reliable)?*

Changed to FLUXNET2015 dataset to ensure better consistency and QC of all data used. Text added to discuss aspects that have been analysed of different processes (Sect. 1, L90–L96):

> Extensive analysis of FLUXNET datasets for the variety of terrestrial PFTs have considered various surface atmosphere controls (e.g., albedo: Cescatti et al. 2012; latent heat flux: Ershadi et al. 2014; spatiotemporal representativeness: Chu et al. 2017, Villarreal and Vargas 2021; energy balance closure: Franssen et al. 2010; landscape heterogeneity: Göckede et al. 2008, Stoy et al. 2013) to enhance understanding of land-atmosphere interactions. As such, this is an ideal data source for deriving widely applicable parameters and assessing performance of SUEWS over different land covers.

22. *How is the variation in albedo at US-AR1 related to the variation in LAI – more explanation is required, i.e. what is the mechanism proposed behind the low rainfall in 2011 mentioned in L341-343?*

New Appendix D is added to demonstrate the rationale for hydrological control of LAI dynamics.

A different site, US-SRG, with has more pronounced relational pattern between LAI and precipitation is chosen to demonstrate this (L838–L845):

b) Rainfall and thermal controls (US-SRG; Fig. D2): at this grassland site in Arizona the intra-annual precipitation has clear dry and wet seasons. The monsoon wet season after the peak air temperature in July through September (Fig. D2a), which has warmest air temperatures, Unlike US-MMS (Fig. D2b), the peak air temperature is more distinct (for a shorter period). A clear relation between the onset of rainfall and LAI enhancement can be seen but the GDD and SDD relation differs from US-MMS and it not captured by the current models in SUEWS. The rainfall and enhanced LAI and $Q_E$ are associated with cooler daily air temperatures. Sites where the LAI dynamics are not captured are not explored further in this paper.

[Figure]

**Figure D2** As Fig. D1 but for US-SRG (GRA according to IGBP; time span: 2008–2015; DOI: 10.18140/FLX/1440114).

23. *Why are the results for water and bare soil not shown?*

N/A: work related to these land covers are removed from paper as all data now from FLUXNET2015 dataset

24. *What is the reason for the very different annual cycles for evergreen trees seen in Fig 10?*

N/A: these sites are not included in the revision as the main data source has been changed to FLUXNET2015.

*25. Why are the conductance parameters for the individual sites not shown (L438)?*

Site-specific values now given in Table C3 and intra-PFT variability in Table 7.

*26. Some physical interpretation of the G2-G6 values should be attempted (i.e. why did the fitting procedure result in these values and what does it tell us?).*

An upper-boundary-based approach is now used to derive these parameters (Sect. 4.3, L435–L438):

> However, as the optimisation may not return values because of the complexity in Eqn. 14 and the challenge of interpreting the derived parameter values, we adopt Matsumoto et al.'s (2008) approach to derive these parameters. Rather than using all the data combinations for $g_s$, the upper boundary of each forcing variable component (e.g. $g(K_\downarrow)$) is considered as the response for unconstrained conditions.

**Specific comments:**

*27. The Abstract is quite vague and suggests some topics will be analysed more deeply than they are. Some suggestions:*

 *a)  add 'Here' at the start of the sentence on L34 to make clear this is what you did in this study and is not a general feature of SUEWS*

 *b)  add 'from multiple sites' to the end of this sentence*

 *c)  meaning of 'guidance to apply SUEWS are provided' (L38) is unclear*

 *d)  L39: 'impacts' on what?*

 *e)  'The relation between LAI and albedo is explored' (L39-40) – I'm not sure this is really covered in the analysis*

 *f)  Add 'in the model' after 'captured' in L44*

 *g)  L44-45: the meaning is unclear and there is no discussion in the manuscript of how latent heat fluxes affects modelled canopy-layer air temperature.*

We have incorporated the suggestion and rewritten the abstract as follows:

> To compare urban and rural areas, the fully vegetated areas (e.g. deciduous trees, evergreen trees and grass) commonly found adjacent to cities need to be modelled. Here we provide a general workflow to derive parameters for SUEWS (Surface Urban Energy and Water Balance Scheme), including those associated with vegetation phenology (via leaf area index, LAI), heat storage and surface conductance. As expected, attribution analysis of bias in SUEWS modelled $Q_E$ finds the surface conductance ($g_s$) plays the dominant role, hence there is need for more estimates of surface conductance parameters. The workflow is applied at 38 FLUXNET sites. The derived parameters vary between sites with the same plant functional type (PFT), demonstrating the challenge of using a single set of parameters for a PFT. SUEWS skill at simulating monthly and hourly latent heat flux ($Q_E$) is examined using the site-specific derived parameters, with the default NOAH surface conductance parameters (Chen et al. 1996). Overall evaluation for two years has similar metrics for both configurations: median hit rate between 0.6 and 0.7, median mean absolute error less than 25 W m$^{-2}$, and median mean bias error ~5 W m$^{-2}$. Performance differences are more evident at monthly and hourly scales, with larger mean bias error (monthly: ~40 W m$^{-2}$; hourly ~30 W m$^{-2}$) results using the NOAH-surface conductance parameters, suggesting that they should be used with caution. Assessment of sites with contrasting $Q_E$ performance demonstrates how critical capturing the LAI dynamics is to the SUEWS prediction skills of $g_s$ and $Q_E$. Generally $g_s$ is poorest in cooler periods (more pronounced at night, when underestimated by ~3 mm s$^{-1}$). Given the global LAI data availability and the workflow provided in this study, any site to be simulated should benefit.

28. *The treatment of snow cover is rather strange. Judging from Fig 10, either the detection of snow cover is not appropriate or the modelled albedo does not capture the true seasonal variability in vegetation characteristics. As snow cover is not considered in this work, perhaps additional sites with much less snow during leaf-off periods would be more valuable.*

Sect. 2.2 (L168–L170): We clarify that

"our focus is on snow-free conditions"

and indicate:

"evaluating the snow module is a large task in its own right"

Sect. 4.1: We illustrate how the snow-affected albedo values are filtered out (L349–L355):

- $\alpha_{min}$ / $\alpha_{max}$: 10th/90th percentile of daily albedo values after the growth and before the senescence. A daily albedo is calculated from 30/60 min FLUXNET observations of incoming and outgoing shortwave radiation for the period 10:00 to 14:00 (local standard time). To remove outliers a clustering method is applied (`ClusterClassify` of *Mathematica* v12.3.1 Wolfram Research 2020). For example, at some high-latitude sites (e.g. CA-Oas) snow occurs, the winter values are based on data from shortly after senescence to shortly before growth (next spring) and the clustering approach removes the snow period albedo values.

As such, although we didn't explicit model snow-related physical processes in this work, we deem our treatment can effectively select albedo values under snow-free conditions for deriving the desired albedo-related parameters (i.e. $\alpha_{min}$ and $\alpha_{max}$).

29. *Why are albedo and LAI parameters derived for each site but surface conductance parameters derived for each land cover type? Analysis of the variation in conductance parameters between sites would be informative and may help to inform about applicability to other sites.*

The parameters are derived and reported for all 38 sites (Table C3). The text has been updated (Sect. 4.3, L452–L458):

The derived surface conductance parameters for the 38 FLUXNET sites (Table 7 and C3) have different intra-PFT variability based on the IQR (dotted lines, Fig. 10) and demonstrates the benefit of the observations and of deriving site-values when possible. It may help in selecting appropriate PFT from other sources (e.g. NOAH values in Appendix A). The $g_{max}$ results are consistent with Hoshika et al. (2018) in terms of inter-PFT ordering (Grass > EveTr and DecTr) and the grass and crop values are comparable (Table C3). However, our derived deciduous trees values are smaller (22 cf. 31 mm s$^{-1}$) and evergreen trees values larger (20 cf. 12 mm s$^{-1}$).

[Figure]

**Figure 10** Median (thick), interquartile range (dashed) and site (thin lines) derived surface conductance related parameters for three land cover types (colour).

**Table 7** As Table 5, but for surface conductance related parameters (Sect. 4.3). See Fig. 10 and Appendix C.

| | $g_{max}$ [mm s$^{-1}$] | $G_K$ [W m$^{-2}$] | $G_T$ [°C] | $T_L$ [°C] | $T_H$ [°C] | $G_{q,base}$ [-] | $G_{q,shape}$ [-] | $G_\theta$ [-] | $\Delta\theta_{WP}$ [mm] |
|---|---|---|---|---|---|---|---|---|---|
| EveTr | 20.5±1.7 | 62±5 | 10.3±1.8 | -13±4 | 41.4±2.0 | 0.391±0.033 | 0.9 | 0.033±0.009 | 511±75 |
| DecTr | 21.2±2.5 | 100±23 | 18.0±4.0 | -18±5 | 38.0±1.5 | 0.439±0.024 | 0.9 | 0.029±0.010 | 521±58 |
| Grass | 38.6±2.8 | 87±13 | 26.1±1.9 | -13±5 | 40.1±2.2 | 0.467±0.033 | 0.9 | 0.048±0.010 | 521±54 |

30. *The paper is missing a balanced consideration of shortcomings of this study. For example,*

   a) *Would these non-urban parameters be expected to be appropriate for e.g. deciduous trees in residential areas (L66) given the possibility of increased urban temperatures or advective effects?*

The starting premise is that the parameters are for large extensive vegetated areas (see Sect. 3.1 where fetch of the sites are discussed), these are unlikely to be found in many urban areas. However, the workflows provided can be used in urban areas when the required observations are available.

   b) *Considering the importance placed on seasonal variation the impact of assuming constant OHM coefficients should be addressed.*

The seasonally-varying OHM coefficients are given in Table C2 and discussed (Sect. 4.2, L398–L412):

> The derived OHM coefficients (Fig. 6) can be determined by season (Anandakumar 1999; Ward et al., 2016; Sun et al., 2017). We distinguish warm ("summer") and cold ("winter") seasons using months (summer: Northern Hemisphere JJA; Southern Hemisphere: DJF; winter: DJF (JJA), respectively). For simplicity, we omit periods when LAI may be changing rapidly. If the daily mean air temperature is warmer/cooler than the annual mean of daily median temperature, then summer/winter OHM coefficients are used in the simulations.

> The OHM coefficients derived for the 38 FLUXNET sites (Table 6, Fig. 7) vary between land cover types and seasons. For each land cover type, $a_1$ and $a_3$ are notably larger in winter than in summer while the seasonal difference in $a_2$ is relatively small. Thus the overall fraction of heat stored does not vary much but diurnal hysteresis effect is weaker in winter. These results are consistent with previous analytical results (Sun et al., 2017). Within each PFT, there is larger variability in $a_2$ and $a_3$ (cf. $a_1$), notably for evergreen and deciduous trees, suggesting using the most appropriate site values (e.g. medians) may improve predictions of the storage heat flux. In addition to the values derived here, we note that more detailed $\Delta Q_S$ observations are available for vegetated sites to derive such OHM coefficients (e.g. McCaughey (1985), Oliphant et al. (2004)).

   c) *By not including snow cover much of the leaf-off periods are not useable, and the significance of accurately capturing seasonal variation in LAI is reduced. If snow cover is not addressed here, these sites seem like a strange choice.*

See response to R1C28.

31. *Previous shortcomings of LSMs have highlighted that parameters derived for particular conditions can lead to bias in model results at other sites. Although this study aims to provide new, generalised parameter values the range of sites used for each land cover type is not very large and does not cover a large geographical area. The limitations of this should at least be mentioned.*

We have added the following recommendations in the concluding remarks (Sect. 6, L709–L713 and L744–L746):

> - Where observations are available, we recommend determining local parameters, as derived parameters vary within PFT (Appendix C). The tools provided here are designed to facilitate this (Sect. 4).

- Given the global availability of MODIS LAI and reanalysis-based air temperature datasets (e.g., ERA5), it is feasible to derive site by site LAI parameters for SUEWS (Sect. 4.1).
- A potential source of parameters values for PFT beyond those studied here (i.e. values provided Appendix C, Sun et al. 2021) could be NOAH-based parameters (Appendix A) but these should be used with caution, as demonstrated (Section 5).

See also response to R1C14.

32. *Keep in mind the appropriateness of the statistics used when comparing different sites and different seasons, where the variables may have different magnitudes and different amounts of data available. Depending on the analyses that end up in the revised paper, this may not be an issue, but it is important to consider. Was any consideration given to other statistical measures (e.g. the correlation coefficient would indicate how well the model reproduces the variability, even if the magnitude is wrong)?*

We have added the non-dimensional metric, hit rate (HR) to examine the frequency the model performance is within an acceptable threshold (Sect. 5.1, L507–L513):

1) hit rate (HR):

$$HR = \frac{\sum_{j=1}^{N} H(\delta_{Y,j} - |Y_{\text{mod},j} - Y_{\text{obs},j}|)}{N} \tag{31}$$

with Heaviside step function $H$ defined by

$$H(x) = \begin{cases} 0, & x < 0 \\ 1, & x \geq 0 \end{cases} \tag{32}$$

and the threshold $\delta_{Y,j}$ being a value dependent on evaluation variable $Y$.

In particular, $\delta_{Y,j}$ for $Q_E$ is determined as a function of net all-wave radiation $Q^*$ following Hollinger and Richardson (2005) to be $\delta_{Y,j} = 0.1Q_j^* + 10$ (in W m$^{-2}$) based on measurement uncertainties.

33. *No uncertainty estimates are provided making it very difficult to judge the robustness and accuracy of these results. This should be rectified in the revised manuscript.*
- See response to R1C32 on hit rate.
- Also added uncertainty estimates in standard deviations into Tables 5–7 following the suggestions.

**Minor comments**

34. *L56-58: Why only mention anthropogenic heat and water here? Urban LSMs have many more additions than this.*

N/A (original text has been removed).

35. *L62: make it clear that these land covers can occur in a single grid. Somewhere the intended grid size for SUEWS needs to be mentioned.*

Rephrased as follows (Sect. 1, L55–L57):

SUEWS characterises the heterogeneity of urban surfaces allowing an integrated mix of seven land covers within a grid cell (neighbourhood scale: $O(0.1–10$ km)) of impervious (buildings, paved) and pervious (evergreen trees/shrubs, deciduous trees/shrubs, grass, soil, water) types.

36. *L64, 67: what is meant by 'integrated'?*

N/A: original text removed.

37. *L70: Here would be a good place to bring in some of the non-urban literature.*

Non-urban related references on LAI, heat storage and surface conductance have been added Sect. 4.1, 4.2 and 4.3, respectively.

38. *L71 'bridge this gap' – you have not really talked about a gap so this doesn't really make sense.*

N/A: original text removed.

39. *L76-79: This overview does not make sense (perhaps reflecting the confused structure)!*

Rephrased based on the new structure (Sect. 1, L99–L105):

> We briefly review the key vegetation biophysics schemes in SUEWS (Sect. 2), describe the FLUXNET2015 (Pastorello et al. 2020) and auxiliary datasets used (Sect. 3), and outline the workflows for deriving parameters (Sect. 4). To assess the quality of the derived parameters the SUEWS modelled latent heat flux is evaluated (Sect. 5). Model parameters related to surface conductance are derived for NOAH at the PFT level (Appendix A) as well as those related to surface roughness based on FLUXNET2015 dataset at the site level (Appendix B). Other model parameters derived following workflows (Sect. 4) are also provided (Appendix C).

40. *L93: 'phenology changes key model parameters' – changes what to what?*

N/A: original text removed

41. *L93-102: these paragraphs do not really make sense. Suggest rephrasing and incorporating at the appropriate stage in the model description section*

Restructured the model description part (Sect. 2.2) as suggested.

42. *Table 1: I personally find the last 3 columns unhelpful. There are details missing from the Definition column (GDD abbreviation, 'coefficients' alone is not informative, 'shortwave radiation', OHM) but possibly this table could be deleted as everything should be defined and explained in the text anyway.*

We respectfully keep the last three columns of Table 1 as we consider they include essential information. We have added the definitions in the caption.

43. *L163: what is t?*

$t$ is time; this definition has been added in Sect. 2.2.2 (L180):

> … and $t$ time.

44. *L165-169: This methodology should be separated from the model description and more details should be added, including what timestep and what type of regression. What is the justification for ignoring variation in these OHM coefficients, particularly for this paper on seasonal variation – was any observed? At least a couple of sentences with references should be added here.*

Separated the description of model physics (Sect. 2) and parameter derivation (Sect. 3).

See response to R1C30b for variations in OHM coefficients.

45. *L181: measurement height for wind speed appears to be Hu in L271*

Measurement height now $z_m$ throughout the paper.

46. *L187: 'stability scale' → 'stability parameter'*

Corrected as suggested.

47. *L201: 'Phenological state is critical' – what is the justification for this statement? Presumably all functions are important if they are not correctly parameterised.*

N/A: original text removed.

48. *L120: It may be easier to follow if this new LAI equation was presented along with Fig 5 so the reader understands why a different parameterisation is needed. Some justification for this equation would be helpful.*

N/A: original text removed; see also response to R1C20.

49. *L135: In reality or in the model?*

N/A: original text removed.

50. *L141-142: Meaning unclear. Which 'model parameters'? How does this paragraph fit with L148-154?*

N/A: original text removed.

51. *L143: And also the longwave radiation components*

Added as suggested (Sect. 2.2.1, L171–L172):

> Within SUEWS the albedo is used with the observed incoming shortwave radiation and longwave radiation to obtain $Q^*$.

52. *Table 2: Without discussion in the text this table is not much use. The caption does not make sense. Why were OHM coefficients for some surfaces derived here and others from the literature?*

OHM coefficients are now given for each site (Table C2) and their features analysed in Sect. 4.2.

53. *L218: How was it ensured that the surface was dry?*

Only days with zero precipitation are used in this work as clarified in L433–L434:

> This requires the surface be dry (Section 2.2.4) which we define as being without recorded rainfall in 24 h.

54. *L253-263: Table 3 is vaguely referenced 4 times here. Why not provide the appropriate information in the text if necessary?*

N/A: original text removed.

55. *Table 3: Why are the DOIs given here? They should be included in the Reference list.*

As the websites linked by DOIs store all related information of these FLUXNET sites and related datasets, we provide them in a summary table to ease the access to these resources.

56. *Table 3: How were the study years chosen and then how was it decided which to use for calibration and which for testing? Without information, a cynical reader may suspect the combination was selected which gives the best results…*

We have added both data available and SUEWS simulation periods in Table 3. Also, the rationale for choice of these periods is clarified in the revised manuscript:

- Site selection (Sect. 3.1, L258–L259):

> 2)     data availability (56/206): require both MODIS LAI data (available from 2002, Sect. 3.2) and long-term continuity (defined here as $\geq$ 3 years for the multiple needs).

- Model configuration (Sect. 5.1, L496–L499):

> Simulations are conducted, with forcing data interpolated to a 5 min timestep (Ward et al. 2016), for three years (Table 3, Evaluation period) starting in mid winter. The first year is discarded to allow for model spin-up. The two subsequent years are evaluated when observed latent heat flux are available.

57. *Figure 2: This figure does not seem very useful. Suggest deleting unless it is better described in the text and provides more insight.*

New Fig. 1 gives overview with details presented in Figs. 3, 6 and 8.

58. *L334-334: Units of LAI missing*

Units added throughout revised manuscript.

59. *L356-359: Meaning unclear.*

N/A: original text removed.

60. *Figure 7, 12, and 14 take up a lot of space, are not very easy to read and are hardly discussed in the text. Perhaps a more useful way to summarise this information could be found, or only the most relevant results displayed. The same goes for Figure 9d and Fig 11d. Note it is frustrating for the reader to have to adjust to essentially reading the same information as Fig 7 and 12 but presented in a different way.*

N/A: original figures removed

61. *Panel labels are missing from Fig 9 and 11.*

N/A: original figures removed

62. *Would it make sense to merge Fig 6 and 9a-c and Fig 10 and 11a-c so that these similar results are presented in a more comparable way? I would suggest perhaps even making the x-axis all one year long so the difference for the crops can be seen more immediately. Please use smaller points.*

N/A: original figures removed

63. *Table 4: Tplant, GDDv and GDDLAImax would be better as separate columns.*

N/A: rice related work not in the revision

64. *Fig 12 and L403: the MAE and MBE seem small for the grassland sites considering the results shown in Fig 10. Please check.*

N/A: original figures removed

65. *L427-429: Where is the justification for this statement? Was the performance of other fluxes checked (e.g. net radiation, storage heat flux)? If these fluxes are not modelled adequately, wouldn't they result in inappropriate conductance parameters being derived (e.g. parameters which are tuned to give the right results for the wrong reasons)?*

N/A: work related to WAT and BSV is not used in the revision. See also Sect. 3.1 for site selection in this revision.

66. *L430: More explanation needed here. Also Q\* needs to be reasonably accurate. The assumptions in the resistance approach and the uncertainties in the roughness parameters should be discussed too. The assumption of homogeneous fetch requires more explanation if it is included here.*

A more detailed analysis of bias attribution has been added in the revised manuscript (Sect. 5.2).

67. *L433-436: See above for explanation of why this comparison does not make sense.*

N/A: removed

68. *Fig 13: Difficult to read (make full-page?). Use smaller points. The annual diurnal pattern is of limited use considering the huge seasonal variation which makes the large interquartile ranges. Consider using daily or (if data availability is an issue) monthly evaporation totals instead, which may allow insight into when the latent heat flux is modelled well and when not.*

N/A: removed

69. *L456-457: This cannot be concluded here as it is not demonstrated in the paper. This hypothesis is suggested as 'a possible explanation' in L341-344 but no other possibilities are discussed and there is no further analysis to substantiate or contradict this suggestion.*

This has been clarified in Sect. 6 (L738–L741):

> None of the simple LAI schemes in SUEWS account for hydrological impacts on LAI. Vegetation with shallow roots (e.g. US-SRG in Arizona, US, categorised as grassland, Fig. D2) are not well modelled when air temperature if the only phenology forcing variable. Hydrological feedback should be considered in future development of the LAI scheme in SUEWS.

70. *L459: It should be stated somewhere that (presumably) to obtain fitted parameters for a specific site observations must be available.*

See response to R1C31.

71. *L558: 'Given this' – given what?*

N/A: text removed

72. *There are also a few typos that would need to be corrected at a later stage.*

Corrected throughout the paper.

**Reviewer 2**

**General Comments:**

1. *The manuscript aims to extent the SUEWS model to non-urban surfaces, with the overall goal to estimate the energy-balance fluxes in such areas. Therefore, specific parameters used to estimate the surface heat fluxes are inferred from observational data sampled at energy balance stations, which includes different vegetation types and different climate zones. The modelled surface fluxes with SUEWS were compared against observational data to evaluate the performance of the SUEWS model over rural surfaces. The topic of the paper itself fits well into the journal and is of interest to the research community, especially since reliable input of phenological data and surface data play a key role for reliable estimation of the surface-energy balance components in models. However, after extensive review, I cannot recommend the manuscript for publication until major revisions have been done and extended analysis is presented. My major concerns are outlined in the following.*

**Major comments**

2. *You added new methods and tuned parameters to model impervious surfaces in rural areas. However, the description of newly developed parameter estimation such as for LAI or albedo is mixed with parts of model description, so that it is hardly possible to extract what is new and what has been there already before. I would recommend to first described the state-of-the-art model and describe newly developed approaches separately. Also, the manuscript provides no condense model description of SUEWS but refers to previous papers. The manuscript itself should be readable as a standalone paper. Hence, even though not all details need to be brought-up, the manuscript needs to provide a proper overview of the model at one place. Further, please give all information concerning model description in the text, not within the appendix.*

See response to R1C5.

3. *The manuscript is sometimes hard to follow due to missing logical order between sentences. In several sections, sentences appear to be disconnected from each other rather than indicating a logical order. As a consequence the text reads more like a collection of notes.*

See responses to R1C5–12 for notable structural changes in this revision.

4. *The discussion of the results is not sufficient and lacks important aspects. For example, why is the bias error positive for some sites but negative for others. The authors provide the errors for all sites, but do not try to put these within the context of site-specific information. Also, one of the main problems of eddy-covariance measurements is the non-closure of the energy balance. Especially for the comparison of surface latent heat fluxes this needs to be discussed. In this context, the manuscript needs to provide also more information about the specific EC sites. At EC stations located in heterogeneous landscapes the measured fluxes are a mixture of signals emerging at different land-surface types rather than only one type (as assumed in this study), i.e. the footprint of the stations covers several land surfaces with different properties (LAI, roughness). As a consequence the value f\_i (which is assumed to be 1 in this study) is not necessarily one. To be able to evaluate the validity of the inferred parameters in this study, site specific information should be provided, e.g. the degree of surface heterogeneity, which in turn need to be correlated to the overall error in the surface latent heat flux for the individual sites.*

- See response to R1C13 for our improved analysis of bias error.

- Site representativeness:
  First, we need to clarify a detailed observational analysis of flux measurements is out of scope of this work. Meanwhile, we fully agree with the reviewer that a better understanding of the measurement contexts may help interpret the results presented here. Given FLUXNET2015 has been extensively analysed in many studies, instead of repeating similar analysis with respect to surface heterogeneity, we provide related references and discussions as follows (Sect.3.1, L272–L277):

The landscape heterogeneity of many FLUXNET EC flux measurements sites have been systematically examined by Stoy et al. (2013) using satellite imagery. Of the sites they examined, they found them to be located within homogeneous parts of the targeted PFT, but the larger landscape (~20 km) may have considerable variability. As a FLUXNET site is typically assigned to one PFT for land surface model development/evaluation (e.g. Stöckli et al. 2008, Zhang et al. 2017), we configure each as a homogeneous grid cell and assume $f_i$ =1.

**Minor Comments**

5.  *L55-56: You mention that there is a number of LSM's, but you cite only one.*

N/A: original text has been removed.

6.  *L54-58: In my opinion this leads the reader on a wrong track, the manuscript focuses on nonurban rural sites.*

We have restructured the introduction part to make the storytelling more rapidly reaching the nonurban topic in the first paragraph of revised manuscript (Sect. 1, L58–L61):

Although SUEWS has been evaluated in cities around the globe (e.g. Karsisto et al., 2016, Ward et al., 2016, Ao et al., 2018, Kokkonen et al., 2018, Harshan et al., 2018) with varying mixes of integrated impervious-pervious land covers, its performance has not been comprehensively examined in fully vegetated areas that commonly occur adjacent to cities.

7.  *L63: I guess you mean "around the globe".*

Modified as suggested.

8.  *L66: The word parameters is unclear at that point and need to be specified. Do you mean certain (bio)physical quantities such as leaf-are densities, surface or material properties, or do you mean certain values used in parametrizations?*

Clarified in Sect. 1 (L68–L70):

it is essential to ensure that any urban-rural comparison in these diagnostics has the proper rural skill and parameters (i.e. values used in parameterisations).

9.  *L71: Which gap does the authors mean? Please be more specific.*

NA (original text has been removed).

10. *L96: It is unclear to what does "The former" refer to.*

Rephased as follows for clarification (Sect. 4.1, L324–L328):

LAI changes also modify both aerodynamic roughness parameters (roughness length $z_0$, zero plane displacement height $z_d$) (e.g. Kent et al. 2017) impacting aerodynamic resistance ($r_a$) and surface resistance ($r_s$). LAI directly moderates $Q_E$ and canopy interception capacity, which modifies when potential evaporation occurs and aspects of the water balance.

11. *L98-99: "Model parameters ...": As a stand alone sentence this makes sense, though it becomes not directly apparent to the reader what is exactly meant. However, from this there is not obvious connection to the following sentence. With changes of the key parameters you may describe any type of vegetation, but how is this related to the statement that parameters need to be consistent?*

NA (original text has been removed).

12. *107-108: How are GDD and SDD defined? Are these vegetation-type specific?*

We have added the definitions of GDD and SDD in Sect. 2.2.1 in the revised manuscript, which are vegetation-type specific and given along with related symbols as follows (Sect. 2.2.1, L152–L157):

In SUEWS, leaf growth is tiggered by reaching a critical growing degree days ($GDD$) threshold ($T_{base,GDD,i}$), and similarly for leaf fall by senescence degree days ($SDD$, $T_{base,SDD,i}$) using daily ($d$) mean air temperatures ($T_d$) based on the previous day ($d$ - 1) for each vegetation type $i$ (one of evergreen trees, deciduous trees and grass). For forests and grass we use (Järvi et al., 2011):

$$LAI_{d,i} = \begin{cases} min(LAI_{max,i}, LAI_{d-1,i}^{\omega_{1,GDD,i}} GDD_{d,i}\, \omega_{2,GDD,i} + LAI_{d-1,i}), & T_{base,SDD,i} < T_d < T_{base,GDD,i} \\ max(LAI_{min,i}, LAI_{d-1,i}\, \omega_{1,SDD,i}(1 - SDD_{d,i})\, \omega_{2,SDD,i} + LAI_{d-1,i}), & T_{BaseGDD,i} < T_d < T_{base,SDD,i} \end{cases} \quad (1)$$

with $GDD_{d,i} = GDD_{d-1,i} + (T_d - T_{base,GDD,i})$, $SDD_{d,i} = SDD_{d-1,i} + (T_d - T_{base,SDD,i})$, and $\omega_{1/2,GDD/SDD,i}$ curve factors needing to be derived.

13. *Eq. 3: Does the index i includes all vegetation types including or excluding crops?*

Work related to the crop-specific LAI model has been removed in this revision.

Also the meaning of *i* has been clarified (Sect. 2.2.1, L154–L155):

> … for each vegetation type *i* (one of evergreen trees, deciduous trees and grass).

14. *Eq. 3: Is LAI_max/min a function of the time of the year? If this is the case, please indicate this somehow within the equation or text.*

$LAI_{max/min}$ is *not* a function of time of year but an adjustable parameter.

In the revised manuscript, we have clarified its meaning as follows (Sect. 2.2.1, L160–L161):

> For each site and vegetation type *i*, the maximum and minimum *LAI* values ($LAI_{max,i}$, $LAI_{min,i}$) and $T_{base,GDD}$ and $T_{base,SDD}$ are determined for each site (Sect. 4.1).

15. *115-116: Where does these max/min values come from? Here, a reference is required in the text.*

Determination of these LAI related parameters have now been detailed in Sect. 4.1.

16. *121: The note within the parenthesis is unclear to me, how are shorter / longer LAI_max times are reflected in Eq. 3?*

N/A: original text removed; see also response to R1C20.

17. *175/189: I guess you mean water vapor.*

Yes; this has been clarified throughout the revised manuscript.

18. *199/200: The authors should elaborate why removing G1 from the first term is a valid approach. According to the text it sounds to be an arbitrary decision, though I assume there is a specific reason for this?*

The G1 was introduced in SUEWS as an adjusting parameter for grid with a mixture of different vegetated land covers (Jarvi et al. 2011, Ward et al. 2016) that rescales the contributions to total surface conductance from different vegetated land covers with respect to their LAI values.

In this work, given the focus on homogeneous land covers, we removed the adjusting parameter G1 for formulation simplicity (we also note mathematically G1 and $g_{max}$ are interchangeable in the formulation for a fully homogeneous land cover as in this work).

19. *200-201 and following: This is not really a sentence but more a note. Also, the following sentences sound more like a note.*

Reworded.

20. *204: To be specific, soil moisture deficit is not really a meteorological quantity.*

Corrected.

21. *Eq. 13,14,15: G_2, G_3, G_4, G_5 are not defined in the text.*

Defined now in the revised manuscript in Sect 2.2.4; please also note we modified the notation with more explicit names for their physical meanings:

G2 → $G_K$: solar radiation ($K$) related parameter.

G3 → $G_{q,base}$: specific humidity ($q$) related parameter for the "base" value.

G4 → $G_{q,shape}$: specific humidity ($q$) related parameter for the curve shape.

G5 → $G_\theta$: soil moisture ($\theta$) related parameter depending on soil type.

22. *L240: Parameters itself cannot have a performance. What you mean is the performance of SUEWS using parameters for non-urban surfaces.*

We meant the model performance when configured with a specific set of parameters. Related text has been clarified throughout the revised manuscript.

23. *L246: What do the authors mean with surface state?*

"surface state ($C_i$)" refers to the water content on canopy, which has been clarified as follows (Sect. 2.2.4, L227–L228):

> The amount of water on the canopy of each surface ($C_i$)

24. *L285: What do the authors mean with "are not completely independent": among each other?*

N/A: original text removed.

25. *L347-349: In Fig. 6 the authors show the LAI distribution over the year. It does not become clear how this indicates that a constant LAI would lead to poor radiation and surface fluxes. If the authors see a link between these two things it should be given there.*

N/A: original text removed.

26. *Fig. 6: The LAI variation for the evergreen-tree sites is surprisingly high. The minimum LAI values for the respective Canadian sites are similar compared to the deciduous-tree sites. For evergreen trees I would expect a rather time-constant value, while here also the MODIS values indicate almost zero LAI. Could the maybe connected to snow cover on trees?*

We thank the reviewer for bringing up this concern, which led us to a more thorough investigation of intra-annual LAI dynamics of various land covers, including evergreen trees.

By looking into intra-annual LAI dynamics of evergreen trees using a long-term (1981–2015) MODIS LAI climatology dataset (Mao and Yan 2019), we find (Fig. R1):

- evergreen green broadleaf forest (EBF) keeps quasi-constant LAI values throughout a year (Fig. R1a).
- evergreen needleleaf forests (ENF) does show apparent intra-annual variability (Fig. R1b), which is consistent with Fig. 6 in our last submission.

[Figure]

Figure R1 Ensemble intra-annual LAI dynamics of a) evergreen green broadleaf forest (EBF) and b) evergreen green needleleaf forest (ENF). Medians are in bold lines while shadings for inter-quartile ranges. "n" denotes number of FLUXNET sites used in plots.

As for the low LAI values found at ENF sites in winter, it is a known issue in MODIS LAI product that seasonal variation can be exaggerated by unrealistically low LAI retrievals over high latitude ENF in winter (Garrigues et al. 2008, Heiskanen et al. 2012). Related discussions have been added in Sect. 4.1 (L365–L368):

> For EveTr sites, the large contrast between $LAI_{max}$ and $LAI_{min}$ in the ENF sites analysed here is consistent with MODIS derived LAI for ENF having larger seasonal variability than EBF (Heiskanen et al. 2012), but some of this is caused by a known issue of particularly low winter values (Garrigues et al. 2008).

27. *Fig 7b: MBE indicates that the modelled LAI values are biased towards smaller values (not for all sites, but for many), especially during the leaf-on period. However, I miss some discussion about this in the text (line 338-345).*

Please see response to R1C13.

28. *Fig 12: Please provide a full description what is shown in the figure. To switch between the figures to find out what is shown makes the figure hardly readable.*

N/A: original figure removed.

29. *Most of the equations: Punctuation is missing.*

Punctuation is added wherever appropriate.

**References**

Cescatti, A., Marcolla, B., Vannan, S. K. S., Pan, J. Y., Román, M. O., Yang, X., Ciais, P., Cook, R. B., Law, B. E., Matteucci, G., Migliavacca, M., Moors, E., Richardson, A. D., Seufert, G. and Schaaf, C. B.: Intercomparison of MODIS albedo retrievals and in situ measurements across the global FLUXNET network, Remote Sens Environ, 121, 323–334, doi:10.1016/j.rse.2012.02.019, 2012.

Chu, H., Baldocchi, D. D., John, R., Wolf, S. and Reichstein, M.: Fluxes all of the time? A primer on the temporal representativeness of FLUXNET, J Geophys Res Biogeosciences, 122(2), 289 307, doi:10.1002/2016jg003576, 2017.

Ershadi, A., McCabe, M. F., Evans, J. P., Chaney, N. W. and Wood, E. F.: Multi-site evaluation of terrestrial evaporation models using FLUXNET data, Agr Forest Meteorol, 187, 46–61, doi:10.1016/j.agrformet.2013.11.008, 2014.

Franssen, H. J. H., Stöckli, R., Lehner, I., Rotenberg, E. and Seneviratne, S. I.: Energy balance closure of eddy-covariance data: A multisite analysis for European FLUXNET stations, Agr Forest Meteorol, 150(12), 1553–1567, doi:10.1016/j.agrformet.2010.08.005, 2010.

Heiskanen, J., Rautiainen, M., Stenberg, P., Mõttus, M., Vesanto, V.-H., Korhonen, L. and Majasalmi, T.: Seasonal variation in MODIS LAI for a boreal forest area in Finland, Remote Sens Environ, 126, 104–115, doi:10.1016/j.rse.2012.08.001, 2012.

Mao, J., and B. Yan: Global Monthly Mean Leaf Area Index Climatology, 1981-2015. ORNL DAAC, Oak Ridge, Tennessee, USA, doi:10.3334/ORNLDAAC/1653, 2019.

Stöckli, R., Lawrence, D. M., Niu, G.-Y., Oleson, K. W., Thornton, P. E., Yang, Z.-L., Bonan, G. B., Denning, A. S., and Running, S. W.: Use of FLUXNET in the Community Land Model development, 113, doi:10.1029/2007JG000562, 2008.

Villarreal, S. and Vargas, R.: Representativeness of FLUXNET Sites Across Latin America, J Geophys Res Biogeosciences, 126(3), doi:10.1029/2020jg006090, 2021.

Wang, Y., Broxton, P., Fang, Y., Behrangi, A., Barlage, M., Zeng, X., and Niu, G.: A Wet-Bulb Temperature-Based Rain-Snow Partitioning Scheme Improves Snowpack Prediction Over the Drier Western United States, Geophys Res Lett, 46, 13825–13835, https://doi.org/10.1029/2019gl085722, 2019.

Garrigues, S., Lacaze, R., Baret, F., Morisette, J. T., Weiss, M., Nickeson, J. E., Fernandes, R., Plummer, S., Shabanov, N. V., Myneni, R. B., Knyazikhin, Y. and Yang, W.: Validation and intercomparison of global Leaf Area Index products derived from remote sensing data, J Geophys Res Biogeosciences 2005 2012, 113(G2), doi:10.1029/2007jg000635, 2008.

Zhang, X., Dai, Y., Cui, H., Dickinson, R. E., Zhu, S., Wei, N., Yan, B., Yuan, H., Shangguan, W., Wang, L. and Fu, W.: Evaluating common land model energy fluxes using FLUXNET data, Adv Atmos Sci, 34(9), 1035–1046, doi:10.1007/s00376-017-6251-y, 2017.

---

## Author Comment (AC2) · 4 Nov 2021

**Surface Urban Energy and Water Balance Scheme (v2020a) in vegetated areas:**

**parameter derivation and performance evaluation using FLUXNET2015 dataset**

Hamidreza Omidvar[1], Ting Sun[1,✉], Sue Grimmond[1,✉], Dave Bilesbach[2], Andrew Black[3], Jiquan Chen[4],
Zexia Duan[5], Zhiqiu Gao[5,6], Hiroki Iwata[7], Joseph P. McFadden[8]

[1] Department of Meteorology, University of Reading, Reading, RG6 6ET, UK
[2] Biological Systems Engineering Department, University of Nebraska, Lincoln, NE, 68588, USA
[3] Faculty of Land and Food System, University of British Columbia, Vancouver, BC, V6T 1Z4, CA
[4] Center for Global Change and Earth Observation, Department of Geography, Michigan State University, East Lansing, MI,
48824, USA
[5] Collaborative Innovation Centre on Forecast and Evaluation of Meteorological Disasters, School of Atmospheric Physics,
Nanjing University of Information Science and Technology, Nanjing, 210044, China
[6] State Key Laboratory of Atmospheric Boundary Layer Physics and Atmospheric Chemistry, Institute of Atmospheric Physics,
Chinese Academy of Sciences, Beijing, 100029, China
[7] Department of Environmental Science, Faculty of Science, Shinshu University, Nagano 390-8621, Japan
[8] Department of Geography, University of California, Santa Barbara, CA, 93106 USA

✉ *Correspondence to*: *ting.sun@reading.ac.uk; c.s.grimmond@reading.ac.uk*

**ORCID:**

Hamidreza Omidvar: *https://orcid.org/0000-0001-8124-7264*
Ting Sun: *https://orcid.org/0000-0002-2486-6146*
Sue Grimmond: *https://orcid.org/0000-0002-3166-9415*
Dave Bilesbach: *https://orcid.org/0000-0001-8661-9178*
Andrew Black: *https://orcid.org/0000-0001-9292-1146*
Jiquan Chen: *https://orcid.org/0000-0003-0761-9458*
Zexia Duan: *https://orcid.org/0000-0003-2822-7066*
Zhiqiu Gao: *https://orcid.org/0000-0001-8256-005X*
Hiroki Iwata: *https://orcid.org/0000-0002-8962-8982*
Joseph P. McFadden: *https://orcid.org/0000-0002-5869-7774*

**Abstract**

To compare urban and rural areas, the fully vegetated areas (e.g. deciduous trees, evergreen trees and grass) commonly found adjacent to cities need to be modelled. Here we provide a general workflow to derive parameters for SUEWS (Surface Urban Energy and Water Balance Scheme), including those associated with vegetation phenology (via leaf area index, LAI), heat storage and surface conductance. As expected, attribution analysis of bias in SUEWS modelled $Q_E$ finds the surface conductance ($g_s$) plays the dominant role, hence there is need for more estimates of surface conductance parameters. The workflow is applied at 38 FLUXNET sites. The derived parameters vary between sites with the same plant functional type (PFT), demonstrating the challenge of using a single set of parameters for a PFT. SUEWS

skill at simulating monthly and hourly latent heat flux ($Q_E$) is examined using the site-specific derived parameters, with the default NOAH surface conductance parameters (Chen et al. 1996). Overall evaluation for two years has similar metrics for both configurations: median hit rate between 0.6 and 0.7, median mean absolute error less than 25 W m$^{-2}$, and median mean bias error ~5 W m$^{-2}$. Performance differences are more evident at monthly and hourly scales, with larger mean bias error (monthly: ~40 W

m$^{-2}$; hourly ~30 W m$^{-2}$) results using the NOAH-surface conductance parameters, suggesting that they should be used with caution. Assessment of sites with contrasting $Q_E$ performance demonstrates how critical capturing the LAI dynamics is to the SUEWS prediction skills of $g_s$ and $Q_E$. Generally $g_s$ is poorest in cooler periods (more pronounced at night, when underestimated by ~3 mm s$^{-1}$). Given the global LAI data availability and the workflow provided in this study, any site to be simulated should benefit.

**Keywords**: SUEWS, FLUXNET, NOAH, LAI, surface conductance, Penman-Monteith equation

**1 Introduction**

The Surface Urban Energy and Water Balance Scheme (SUEWS, Grimmond et al., 1986, 1991,

Grimmond & Oke 1991, Järvi et al., 2011) is widely used to simulate urban surface energy and hydrological fluxes, with heat and water released by anthropogenic activities accounted for (Grimmond et al., 1986; Grimmond, 1992). SUEWS characterises the heterogeneity of urban surfaces allowing an integrated mix of seven land covers within a grid cell (neighbourhood scale: $O$(0.1–10 km)) of impervious (buildings, paved) and pervious (evergreen trees/shrubs, deciduous trees/shrubs, grass, soil, water) types.

Although SUEWS has been evaluated in cities around the globe (e.g. Karsisto et al., 2016, Ward et al.,

2016, Ao et al., 2018, Kokkonen et al., 2018, Harshan et al., 2018) with varying mixes of integrated impervious-pervious land covers, its performance has not been comprehensively examined in fully vegetated areas that commonly occur adjacent to cities.

One common and demanding application of urban climate models, including SUEWS, is to examine the very well-known canopy layer urban heat island effects – parts of cities are often warmer than their surroundings at night – and to understand the causes (Oke 1973, 1982). This requires both the "rural"

context – usually characterised by pervious land cover – to be simulated appropriately ideally using the same modelling framework. As SUEWS v2020a (Tang et al. 2021) can diagnose near surface meteorology in the roughness sub-layer and canopy layer (e.g., air temperature and humidity at 2 m agl (above ground level), wind speed at 10 m agl), it is essential to ensure that any urban-rural comparison in these diagnostics has the proper rural skill and parameters (i.e. coefficient values used in parameterisations).

For meso-scale numerical weather prediction (NWP) of an urban region, both rural and urban areas need to be simulated. With plans to couple SUEWS to a meso-scale model (e.g. Weather Research and

Forecasting (WRF), Skamarock and Klemp (2008)), most regions have extensive areas that have completely pervious grid cells. As these need to be simulated using a consistent surface scheme, it is essential to have appropriate parameters for these areas.

Central to the SUEWS biophysics, is the Penman-Monteith approach (Penman 1948; Monteith 1965) with a Jarvis-type (Jarvis 1976) surface moisture conductance (Grimmond and Oke 1991). Despite various parameters having been derived to account for different urban areas (e.g. land cover differences) and regions (e.g. high/mid-latitude) to allow for changing phenology, conductance and storage heat flux related parameters (e.g. Järvi et al. 2011, 2014; Ward et al. 2016), urban parameter estimates are lacking partly because of limited observations and lack of a standard workflow for deriving parameters. Other land surface schemes have parameters for a wide range of plant functional types (PFT) (e.g. NOAH

within WRF, Chen et al. 1996, Chen and Dudhia 2001) but are often derived from a small number of observational sites and their widespread applicability is unexamined. For example, NOAH largely adopted values from the HAPEX-MOBILHY observational program (Andre et al. 1986) following

Noilhan and Planton (1989).

FLUXNET (Baldocchi et al., 2001), a global network of research sites that monitor surface-atmosphere exchanges of carbon, water, and energy using the eddy covariance technique, provide unprecedented possibilities to advance the process-based land surface modelling with a comprehensive collection of datasets for either development (e.g. Stöckli et al. 2008) or evaluation (e.g. Zhang et al. 2017). Extensive analysis of FLUXNET datasets for the variety of terrestrial PFTs have considered various surface atmosphere controls (e.g., albedo: Cescatti et al. 2012; latent heat flux: Ershadi et al. 2014; spatiotemporal representativeness: Chu et al. 2017, Villarreal and Vargas 2021; energy balance closure: Franssen et al.

2010; landscape heterogeneity: Göckede et al. 2008, Stoy et al. 2013) to enhance understanding of land- atmosphere interactions. As such, this is an ideal data source for deriving widely applicable parameters and assessing performance of SUEWS over different land cover types.

In this work, we develop general workflows (Fig. 1) to derive vegetation related parameters associated with phenology, the storage heat flux and surface moisture conductance; and comprehensively examine model skill in modelling latent heat fluxes. We briefly review the key vegetation biophysics schemes in

SUEWS (Sect. 2), describe the FLUXNET2015 (Pastorello et al. 2020) and auxiliary datasets used (Sect.

3), and outline the workflows for deriving parameters (Sect. 4). To assess the quality of the derived parameters the SUEWS modelled latent heat flux is evaluated (Sect. 5). Model parameters related to surface conductance are derived for NOAH at the PFT level (Appendix A) as well as those related to surface roughness based on FLUXNET2015 dataset at the site level (Appendix B). Other model parameters derived following workflows (Sect. 4) are also provided (Appendix C). The source code, input data and model simulations analysed are provided in Sun et al. (2021).

[Figure]

**Figure 1** *Overview of workflows to derive parameters, to undertake and to evaluate simulations.*
*Acronyms are defined in Sect. 2 and 3. More details are provided in Figures 3, 5, and 6.*

**2   SUEWS model**

**2.1   Overview of SUEWS physics for vegetated areas**

Surface Urban Energy and Water Balance Scheme (SUEWS) is a local-scale land surface model for simulating the surface energy and hydrological fluxes (Grimmond and Oke 1986, 1991; Järvi et al. 2011,

2014; Offerle et al. 2003; Ward et al. 2016) without requiring specialised computing facilities. It has been extensively evaluated and applied in many cities (Lindberg et al.'s (2018) Table 3, Sun and Grimmond's (2019) Table 1). Other details of how SUEWS computes the surface energy, water and carbon fluxes are given in recent model description papers (Järvi et al., 2011; Ward et al., 2016; Järvi et al., 2019).

The surface energy and water balances are directly linked by the turbulent latent heat flux ($Q_E$) or its mass equivalent evaporation ($E$) (Grimmond and Oke 1986, 1991):

$$Q^* + Q_F = Q_H + Q_E + \Delta Q_S \tag{1}$$

$$P + I_e = E + R + \Delta S \tag{2}$$

where $Q^*$ is the net all-wave radiation flux, $Q_F$ is the anthropogenic heat flux, $Q_H$ is the turbulent sensible heat flux, $\Delta Q_S$ the net storage heat flux, and $P$, $I_e$, $\Delta S$ and $R$ are precipitation, external water use, net change in the water storage (e.g. canopy, soil moisture, water bodies) and runoff, respectively. The sites selected in this work are assumed to have no irrigation, so $I_e$ is assumed to be 0 mm s$^{-1}$.

In SUEWS, a modified Penman-Monteith equation (Penman, 1948; Monteith, 1965) is used to compute

$Q_E$ with an expectation in cities that the anthropogenic heat flux ($Q_F$) is greater than zero (Grimmond and

Oke, 1991):

$$Q_E = \frac{s(Q^* + Q_F - \Delta Q_S) + \frac{\rho c_p VPD}{r_a}}{s + \gamma\left(1 + \frac{r_s}{r_a}\right)} \tag{3}$$

However, with our current focus on extensive (non-urban) pervious areas $Q_F$ is assumed to be 0 W m$^{-2}$.

The atmospheric state is obtained from the slope of saturation water vapour pressure curve with respect to air temperature ($s$, units: Pa K$^{-1}$), density of air ($\rho$, kg m$^{-3}$), specific heat of air at constant pressure ($c_p$, J

K$^{-1}$ kg$^{-1}$), vapour pressure deficit (*VPD,* Pa), psychrometric 'constant' ($\gamma$,: Pa K$^{-1}$), and the aerodynamic resistance for water vapour ($r_a$, units: s m$^{-1}$).

Under given ambient meteorological conditions (e.g., incoming solar radiation $K_\downarrow$, air temperature $T_a$, humidity) at an extensive vegetated site, $Q_E$ from this method is sensitive to the estimation of available energy (i.e. $Q^* - \Delta Q_S$), aerodynamic resistance $r_a$ and surface resistance $r_s$. Hence, the critical vegetation related parameters (Table 1) are addressed with these caveats and/or assumptions:

- 140 surface emissivity $\varepsilon_0$ and canopy water storage capacity $S_i$ are assumed to the same as reported in
Ward et al. (2016)

- 142 aerodynamic resistance $r_a$ is highly dependent on aerodynamic parameters that vary with canopy
height ($H_c$) and leaf area index (LAI) (Kent et al., 2017; Appendix B). The temporal varying $H_c$ is
obtained from FLUXNET2015 (Sect. 2.2.3). LAI varies with phenology (Sect. 2.2.1)

**Table 1** *Parameters and the first process they are used in by SUEWS (i.e. most impact multiple*
*variables). Sources (S) include this study (\*), Ward et al. (2016) (W16) and FLUXNET2015 (F15,*
*Pastorello et al. 2020) where values are given (Table: T#, Sect.: S#) or used in individual*
*equations (E). Two key phenology periods are related to growing and senescence degree days*
*(GDD, SDD).*

| Category | Symbol | Definition | Value | | | E | S |
|---|---|---|---|---|---|---|---|
| **Leaf Area Index** | $LAI_{min}$ | Minimum LAI | TC1 | | | 3 | * |
| **(LAI)** | $LAI_{max}$ | Maximum LAI | TC1 | | | 3 | * |
| | $T_{base,GDD}$ | Base temperature for SDD | TC1 | | | 4 | * |
| | $T_{base,SDD}$ | Base temperature for GDD | TC1 | | | 4 | * |
| | $GDD_{full}$ | Ending GDD at $LAI_{max}$ | TC1 | | | 5 | * |
| | $SDD_{full}$ | Ending SDD at $LAI_{min}$ | TC1 | | | 5 | * |
| | $\omega_{1(2),GDD}$ $_{(SDD)}$ | Curve factors used in the LAI model dependent on GDD (SDD) | TC1 | | | 3 | * |
| **Radiation** | $\alpha_{LAI_{min}}$ | Albedo at $LAI_{min}$ | TC1 | | | 6 | * |
| | $\alpha_{LAI_{max}}$ | Albedo at $LAI_{max}$ | TC1 | | | 6 | * |
| | $\varepsilon_0$ | Emissivity | EveTr | DecTr | Grass | | W16 |
| | | | 0.98 | 0.98 | 0.93 | | |
| **Storage heat flux** | $a_1, a_2, a_3$ | Objective Hysteresis Model (OHM) coefficients | TC2 | | | 7 | * |
| **Aerodynamic** | $H_c$ | Vegetation height | F15 varies | | | | F15 |
| **resistance** | $z_{0m}$ | Roughness length for momentum | S2.2.3/B | | | 9/B1 | * |
| | $z_d$ | Zero plane displacement | S2.2.3/B | | | 9/B1 | * |
| **Surface** | $g_{max}$ | Maximum surface conductance | TC3 | | | 14 | * |
| **resistance** | $G_K$ | Solar radiation related parameter | TC3 | | | 15 | * |
| | $G_{q,base},$ $G_{q,shape}$ | Specific humidity related parameters for base value and curve shape | TC3 | | | 16 | * |
| | $G_T$ | Air temperature related parameter | TC3 | | | 17 | * |
| | $T_H, T_L$ | Temperature limits for switching off evaporation | TC3 | | | 17 | * |
| | $G_\theta$ | Soil moisture related parameter | TC3 | | | 18 | * |
| | $\Delta\theta_{WP}$ | Soil moisture deficit at wilting point | TC3 | | | 18 | * |
| **Water storage** | $S_i$ | Canopy water storage capacity (mm) | EveTr | DecTr | Grass | 21 | W16 |
| | | | 0.8 | 1.3 | 1.9 | | |

**150 2.2 $Q_E$ related sub-schemes in SUEWS**

**151 2.2.1 Leaf Area Index (LAI) and Radiation**

In SUEWS, leaf growth is tiggered by reaching a critical growing degree days ($GDD$) threshold ($T_{base,GDD,i}$), and similarly for leaf fall by senescence degree days ($SDD, T_{base,SDD,i}$) using daily ($d$) mean air temperatures ($T_d$) based on the previous day ($d$ - 1) for each vegetation type $i$ (one of evergreen trees, deciduous trees and grass). For forests and grass we use (Järvi et al., 2011):

$$LAI_{d,i} = \begin{cases} min(LAI_{max,i}, LAI_{d-1,i}^{\omega_{1,GDD,i}} GDD_{d,i}\, \omega_{2,GDD,i} + LAI_{d-1,i}), & T_{base,SDD,i} < T_{d-1} < T_{base,GDD,i} \\ max\big(LAI_{min,i}, LAI_{d-1,i}\, \omega_{1,SDD,i}(1 - SDD_{d,i})\, \omega_{2,SDD,i} + LAI_{d-1,i}\big), & T_{base,GDD,i} < T_{d-1} < T_{base,SDD,i} \end{cases} \quad (4)$$

with $\omega_{1/2,GDD/SDD,i}$ curve factors needing to be derived. $T_{d-1}$ is derived from the daily maximum ($T_a^{max}$) and minimum ($T_a^{min}$) air temperature of the previous day:

$$T_{d-1} = \frac{T_a^{max} + T_a^{min}}{2} \tag{5}$$

For each site and vegetation type $i$, the maximum and minimum $LAI$ values ($LAI_{max,i}$, $LAI_{min,i}$) and

$T_{base,GDD}$ and $T_{base,SDD}$ are determined for each site (Sect. 4.1). For sites at higher latitude (e.g. > 60 °), other characteristics – such as day-length and photo period – are helpful to account for corresponding LAI

controls (Bauerle et al., 2012; Gill et al., 2015).

LAI influences several processes in SUEWS – such as dynamics of surface conductance (later in Sect.

2.2.4) and albedo – the latter varies with daily $LAI$ between a minimum ($\alpha_{LAI_{min}}$) and maximum ($\alpha_{LAI_{max}}$) by vegetation type:

$$\alpha_{d,i} = \alpha_{d-1,i} + \left(\alpha_{LAI,max,i} - \alpha_{LAI,min,i}\right)\frac{LAI_{d,i} - LAI_{d-1,i}}{LAI_{max,i} - LAI_{min,i}} \tag{6}$$

We note the SUEWS urban snow module (Järvi et al., 2014) is not used in this work, so focus on snow- free conditions. This may bias some modelled $\alpha$ and subsequent fluxes, but evaluating the snow module is a large task in its own right.

Within SUEWS the albedo is used with the observed incoming shortwave radiation and longwave radiation to obtain $Q^*$. In the current analyses, the observed incoming longwave ($L_\downarrow$) and modelled outgoing longwave radiation ($L_\uparrow = (1 - \varepsilon_0)L_\downarrow + \varepsilon_0\sigma T_s^4$ where $\varepsilon_0$ is the surface emissivity, $\sigma$ is the

Stefan Boltzmann constant (W m$^{-2}$ K$^{-4}$), and $T_s$ is the surface temperature (K)) are used. Table 1 gives the emissivity values used.

**2.2.2   Storage heat flux**

Storage heat flux $\Delta Q_S$ is simulated with the objective hysteresis model (OHM (Grimmond et al., 1991)):

$$\Delta Q_S = \sum_i f_i \left[a_{1,i}Q^* + a_{2,i}\frac{\partial Q^*}{\partial t} + a_{3,i}\right] \tag{7}$$

where $f_i$ is the plan area (or 3-dimensional fraction area, Grimmond et al. 1991, Grimmond and Oke 1999)

fraction of surface $i$, $a_{1-3}$ the OHM coefficients (Sect. 4.2), and $t$ time.

**2.2.3   Aerodynamic resistance**

Aerodynamic resistance $r_a$ is obtained from (Järvi et al., 2011; van Ulden and Holtslag, 1985):

$$r_a = \frac{\left[\ln\left(\frac{z_m - z_d}{z_{0m}} - \psi_m(\zeta)\right)\right]\left[\ln\left(\frac{z_m - z_d}{z_{0v}} - \psi_v(\zeta)\right)\right]}{\kappa^2 u}, \tag{8}$$

where $z_m$ is the measurement height for mean wind speed ($u$) and $\kappa$ the von Kármán constant (0.4

assumed); the aerodynamic parameters $z_d$ (zero plane displacement height) and $z_{0m}$ (roughness length for the momentum) are estimated as a function of canopy height $H_c$ (Garratt 1990; Grimmond and Oke,

1999):

$$z_{0m} = f_0\, H_c \tag{9}$$

$$z_d = f_d\, H_c \tag{10}$$

with $f_0$ and $f_d$ being vegetation-based coefficients (see Appendix B for derivation details). The stability parameter $\zeta$ ($= (z_m - z_d)/L$) depends on the Obukhov length $L$. The atmospheric stability functions of momentum ($\psi_m$) and water vapour ($\psi_v$) for unstable condition are (Campbell and Norman, 1998):

$$\psi_v = 2 \ln\left[\frac{1 + (1 - 16\zeta)^{1/2}}{2}\right], \qquad \psi_m = 0.6\psi_v \tag{11}$$

and for stable condition (Campbell and Norman, 1998; Högström, 1988):

$$\begin{aligned} \psi_v &= -4.5 \ln(1 + \zeta) \\ \psi_m &= -6 \ln(1 + \zeta) \end{aligned} \tag{12}$$

**2.2.4   Surface resistance ($r_s$) or conductance ($g_s$)**

For completely wet surfaces, the surface resistance $r_s$ is assumed to be 0 s m$^{-1}$ (i.e. potential evaporation is calculated from Eq. 3). Otherwise $r_s$, or its inverse, surface conductance $g_s$, is parameterised with a Jarvis- type formulation (Jarvis 1976) in SUEWS (Grimmond and Oke 1991; Järvi et al. 2011; Ward et al. 2016):

$$r_s^{-1} = g_s = \sum_i \left( g_{\mathrm{max},i} \cdot f_i \cdot g(LAI_i) \right) g(K_\downarrow)g(\Delta q)g(T_a)g(\Delta\theta_{soil}). \tag{13}$$

where $g_s$ is determined from the $i$-th land cover areally weighted maximum surface conductance ($g_{\mathrm{max},i}$)

(with $f_i = 1$ for a "homogeneous" site) and  environmental ($x$) rescaling functions ($g(x)$) ranging between

[0, 1], including:

• Leaf area index (LAI) (Ward et al. 2016):

$$g(LAI_i) = \frac{LAI_i}{LAI_{\mathrm{max},i}} \tag{14}$$

is relative to the maximum LAI of land cover $i$ ($LAI_{max,i}$). For bare soil surfaces (i.e. no vegetation), when LAI is irrelevant $g(LAI_i) = 1$.

• incoming shortwave radiation ($K_\downarrow$) (Stewart 1988):

$$g(K_\downarrow) = \frac{\dfrac{K_\downarrow}{G_K + K_\downarrow}}{\dfrac{K_{\downarrow,\mathrm{max}}}{G_K + K_{\downarrow,\mathrm{max}}}} \tag{15}$$

where the $G_K$ parameter modifies the $K_\downarrow$ response, relative to $K_{\downarrow,\max}$ the maximum observed incoming shortwave radiation (= 1200 W m$^{-2}$): when $K_\downarrow$ approaches $G_K$, $g(K_\downarrow)$ reaches 50% of

$g_{s,\max}\left(\frac{K_{\downarrow,\max}}{G_K+K_{\downarrow,\max}}\right)^{-1}$ (i.e., $g_{s,\max}$ normalised by $\frac{K_{\downarrow,\max}}{G_K+K_{\downarrow,\max}}$). At night $g(K_\downarrow)$ goes to 1.

• specific humidity deficit ($\Delta q$) (Ogink-Hendriks 1995):

$$g(\Delta q) = G_{q,base} + \left(1 - G_{q,base}\right)G_{q,shape}^{\Delta q} \tag{16}$$

where the specific humidity related parameters are for the 'base' $G_{q,base}$ and curve shape $G_{q,shape}$: the former indicates the limit of $g(\Delta q)$ when $\Delta q$ approaches extremely large values, while the latter determines the curvature of the $g(\Delta q)$ (e.g., Fig. 9c)

• air temperature ($T_a$) (Stewart 1988):

$$g(T_a) = \frac{(T_a - T_L)(T_H - T_a)^{T_c}}{(G_T - T_L)(T_H - G_T)^{T_c}} \tag{17}$$

where $T_c = \frac{T_H - G_T}{G_T - T_L}$ is a function of the lower ($T_L$) and upper ($T_H$) limits when the evaporation occurs, and

$G_T$ the optimal temperature for evaporation to reach its potential maximum.

• soil moisture deficit ($\Delta\theta_{soil}$, difference between soil water capacity and soil moisture content) (Ward et al. 2016):

$$g(\Delta\theta_{soil}) = \frac{1 - \exp\left(G_\theta(\Delta\theta_{soil} - \Delta\theta_{WP})\right)}{1 - \exp(-G_\theta\Delta\theta_{WP})} \tag{18}$$

both the wilting point ($\Delta\theta_{WP}$) and a soil type dependent parameter ($G_\theta$) vary with soil and plant type.

Appendix A gives the equivalent form used in the NOAH model for Eqn. 13.

SUEWS has a running water balance that accounts for the multiple surface types. The amount of water on the canopy of each surface ($C_i$) (Grimmond & Oke 1991) is used to vary the surface resistance between dry and wet ($r_s = 0$ s m$^{-1}$) by replacing $r_s$ with $r_{ss}$ (Shuttleworth, 1978):

$$r_{ss} = \left[\frac{W}{r_b(s/\gamma + 1)} + \frac{(1 - W)}{r_s + r_b(s/\gamma + 1)}\right]^{-1} - r_b(s/\gamma + 1), \tag{19}$$

where $W$ is a function of the relative amount of water present on each surface to its water storage capacity ($S_i$, Table 1):

$$\begin{aligned} W = 1 \qquad & C_i \geq S_i \\ W = \frac{K - 1}{K - S_i/C_i} \quad & C_i < S_i \end{aligned} \tag{20}$$

$K$ depends on the aerodynamic and surface resistances:

$$K = \frac{(r_s/r_a)/(r_a - r_b)}{r_s + r_b(s/\gamma + 1)}, \tag{21}$$

where $r_b$, the boundary layer resistance, is a function of friction velocity $u_*$ (Shuttleworth, 1983):

$$r_b = 1.1u_*^{-1} + 5.6u_*^{\frac{1}{3}}.\tag{22}$$

Eqn. 19-22 ensure that the surface resistance $r_{ss}$ has a smooth transition from 0 (a completely wet surface)

to $r_s$ (a dry surface).

**3   Global observational datasets used**

We use three global datasets FLUXNET2015 (Pastorello et al. 2020), MODIS (Myneni et al., 2015) and

SoilGrids (Hengl et al. 2014) to derive the parameters. The FLUXNET2015 surface energy fluxes and other meteorology observations are used for three purposes: to derive parameter values, force simulations and evaluate simulations. The remotely sensed (MODIS) derived LAI products are used for the LAI

related parameters. To derive soil moisture related parameters the SoilGrids data are used. Unlike the

FLUXNET2015 data, the latter two datasets are spatially continuous.

**3.1   FLUXNET2015**

The FLUXNET2015 dataset (Pastorello et al. 2020) is the newest version of the FLUXNET data products. The gap-filled dataset includes 212 flux sites from around the world. Although the FLUXNET

focus is on local scale ecosystem $CO_2$ eddy covariance (EC) fluxes, it also includes water and energy EC

fluxes plus other meteorological and biological data. The biosphere-atmosphere exchange dataset contains more than 1500 site-years for the period to the end of 2014. The open source package ONEFlux (Open

Network-Enabled Flux processing pipeline; *https://github.com/fluxnet/ONEFlux*) is used to produce

FLUXNET2015 (Pastorello et al. 2020).

Half-hourly observations (Table 2) are used from 38 sites (Table 3) in three regions (Fig. 2). These sites are selected to meet the following criteria (number of remaining sites that met the criteria):

1)   sites with CC-BY 4.0 license (206/212).

2)   data availability (56/206): require both MODIS LAI data (available from 2002, Sect. 3.2) and long- term continuity (defined here as ≥ 3 years for the multiple needs).

3)   model capacity (38/56): the SUEWS v2020a LAI scheme used is driven by air temperature but other meteorological variables (e.g., rainfall) may strongly influence some sites phenology (Appendix D).

Unfortunately, no datasets are left after selection based on the above criteria for some regions (Fig. 2), including Africa, Asia and South America.

As SUEWS allows any grid cell to have three vegetation or plant functional types (PFT), with the sub- type or properties varying between grids, we subdivide the 38 sites into three classes using IGBP (Table

3) (code, number of sites):

a)  Evergreen trees/shrubs (EveTr, 13): Evergreen Needleleaf Forests (ENF, 12), Evergreen Broadleaf
Forests (EBF, 1)
b)  Deciduous trees/shrubs (DecTr, 11): Mixed Forests (MF, 2), Deciduous Broadleaf Forests (DBF, 5),
Open Shrublands (OSH, 1), Woody Savannas (WSA, 1), Savannas (SAV, 2).
c)  Grass (14): Grasslands (GRA, 8), Croplands (CRO, 6).

The landscape heterogeneity of many FLUXNET EC flux measurements sites have been systematically
examined by Stoy et al. (2013) using satellite imagery. Of the sites they examined, they found them to be
located within homogeneous parts of the targeted PFT, but the larger landscape (~20 km) may have
considerable variability. As a FLUXNET site is typically assigned to one PFT for land surface model
development/evaluation (e.g. Stöckli et al. 2008, Zhang et al. 2017, Chu et al. 2021), we configure each as
a homogeneous grid cell and assume $f_i$ =1.

**Table 2** FLUXNET2015 (Pastorello et al. 2020) variables used in this work to derive parameters (P),
to force (F) model simulations and to evaluate (E) models.

| Variable | Unit | Description | Usage |
|---|---|---|---|
| $H_c$ | m (agl) | Canopy height | P |
| $K_\uparrow$ | W m$^{-2}$ | Outgoing solar radiation | P |
| $K_\downarrow$ | W m$^{-2}$ | Incoming solar radiation | P, F |
| $L_\uparrow$ | W m$^{-2}$ | Outgoing longwave radiation | P |
| $L_\downarrow$ | W m$^{-2}$ | Incoming longwave radiation | F |
| $P$ | mm h$^{-1}$ | Precipitation rate | P, F |
| PA | Pa | Station atmospheric pressure | P, F |
| $Q^*$ | W m$^{-2}$ | Net all-wave radiation | P |
| $Q_E$ | W m$^{-2}$ | Latent heat flux | P, E |
| $Q_H$ | W m$^{-2}$ | Sensible heat flux | P |
| RH | % | Relative humidity | P, F |
| $T_a$ | °C | Air temperature | P, F |
| $u$ | m s$^{-1}$ | Wind speed | P, F |
| $u_*$ | m s$^{-1}$ | Friction velocity | P |
| VPD | Pa | Vapour pressure deficit | P |
| $\theta$ | m$^3$ m$^{-3}$ | Volumetric soil water content | P |

**Table 3** *Key information about the FLUXNET2015 sites (Pastorello et al. 2020), and their DOI reference) used in this work: site name (country-name) with altitude above sea level (asl)) or anemometer sensor height above ground level (agl), More details about the simulation and analyses given in Sect. 5.1. The land cover type as defined based on IGBP (International Geosphere–Biosphere Programme) by the FLUXNET curators (https://fluxnet.org/data/badm-data-templates/igbp-classification/) with the crops (CRO) being: 1 - Rotation: cereal, potato, sugar beet (Moureaux et al., 2006), 2 - Rotation: winter barley, rapeseed, winter wheat, maize and spring barley at DE-Kli (Prescher et al., 2010), 3 - Continuous maize (https://doi.org/10.18140/FLX/1440084); 4 - Rotation: maize and soybean (https://doi.org/10.18140/FLX/1440085), 5- Rotation: maize and soybean (https://doi.org/10.18140/FLX/1440086). The SUEWS recommended vegetation/PFT class (informed by IGBP) data as used in this paper are given in Sun et al. (2021).*

| Site | Latitude (°) | Longitude (°) | Altitude (m asl) | Measurement height $z_m$ (m agl) | Parameter derivation period | Evaluation period | Temporal resolution (min) | No. of valid entries | IGBP | SUEWS | DOI |
|---|---|---|---|---|---|---|---|---|---|---|---|
| AT-Neu | 47.1167 | 11.3175 | 970 | 3.0 | 2005 – 2012 | 2002 – 2004 | 30 | 33582 | GRA | Grass | 10.18140/FLX/1440121 |
| AU-ASM | -22.283 | 133.249 | 607 | 11.7 | 2013 – 2014 | 2010 – 2012 | 30 | 33710 | SAV | DecTr | 10.18140/FLX/1440194 |
| AU-DaS | -14.1593 | 131.388 | 73 | 21.0 | 2011 – 2014 | 2008 – 2010 | 30 | 34022 | SAV | DecTr | 10.18140/FLX/1440122 |
| AU-Gin | -31.3764 | 115.714 | 51 | 15.0 | 2011 – 2011 | 2011 – 2013 | 30 | 28766 | WSA | DecTr | 10.18140/FLX/1440199 |
| AU-Wom | -37.4222 | 144.094 | 705 | 30.0 | 2013 – 2014 | 2010 – 2012 | 30 | 34434 | EBF | EveTr | 10.18140/FLX/1440207 |
| BE-Lon | 50.5516 | 4.74623 | 167 | 2.7 | 2007 – 2014 | 2004 – 2006 | 30 | 34486 | CRO[1] | Grass | 10.18140/FLX/1440129 |
| CA-Gro | 48.2167 | -82.1556 | 340 | 43.3 | 2006 – 2014 | 2003 – 2005 | 30 | 32669 | MF | DecTr | 10.18140/FLX/1440034 |
| CA-Oas | 53.6289 | -106.198 | 530 | 39.0 | 1995 – 2010 | 2002 – 2004 | 30 | 34885 | DBF | DecTr | 10.18140/FLX/1440043 |
| CA-Qfo | 49.6925 | -74.3421 | 382 | 24.0 | 2006 – 2010 | 2003 – 2005 | 30 | 32980 | ENF | EveTr | 10.18140/FLX/1440045 |
| CA-SF2 | 54.2539 | -105.878 | 520 | 9.1 | 2005 – 2005 | 2002 – 2004 | 30 | 31701 | ENF | EveTr | 10.18140/FLX/1440047 |
| CA-SF3 | 54.0916 | -106.005 | 540 | 20.0 | 2005 – 2007 | 2002 – 2004 | 30 | 34618 | OSH | DecTr | 10.18140/FLX/1440048 |
| CA-TP4 | 42.7102 | -80.3574 | 184 | 28.0 | 2005 – 2014 | 2002 – 2004 | 30 | 34533 | ENF | EveTr | 10.18140/FLX/1440053 |
| CH-Cha | 47.2102 | 8.41044 | 393 | 2.4 | 2006 – 2014 | 2005 – 2007 | 30 | 34480 | GRA | Grass | 10.18140/FLX/1440131 |
| CH-Dav | 46.8153 | 9.85591 | 1639 | 35.0 | 2005 – 2014 | 2002 – 2004 | 30 | 24456 | ENF | EveTr | 10.18140/FLX/1440132 |
| CH-Oe1 | 47.2858 | 7.73194 | 450 | 1.2 | 2005 – 2008 | 2002 – 2004 | 30 | 34768 | GRA | Grass | 10.18140/FLX/1440135 |
| DE-Gri | 50.95 | 13.5126 | 385 | 3.0 | 2007 – 2014 | 2004 – 2006 | 30 | 34659 | GRA | Grass | 10.18140/FLX/1440147 |
| DE-Hai | 51.0792 | 10.4522 | 430 | 42.0 | 2005 – 2012 | 2002 – 2004 | 30 | 35028 | DBF | DecTr | 10.18140/FLX/1440148 |
| DE-Kli | 50.8931 | 13.5224 | 478 | 3.5 | 2007 – 2014 | 2004 – 2006 | 30 | 33933 | CRO[2] | Grass | 10.18140/FLX/1440149 |
| DE-Lkb | 49.0996 | 13.3047 | 1308 | 9.0 | 2012 – 2014 | 2009 – 2011 | 30 | 29726 | ENF | EveTr | 10.18140/FLX/1440214 |
| DE-Obe | 50.7867 | 13.7213 | 734 | 30.0 | 2011 – 2014 | 2008 – 2010 | 30 | 33872 | ENF | EveTr | 10.18140/FLX/1440151 |
| FI-Hyy | 61.8474 | 24.2948 | 181 | 24.0 | 2005 – 2014 | 2002 – 2004 | 30 | 30979 | ENF | EveTr | 10.18140/FLX/1440158 |
| FR-LBr | 44.7171 | -0.7693 | 61 | 41.5 | 2005– 2008 | 2002 – 2004 | 30 | 34364 | ENF | EveTr | 10.18140/FLX/1440163 |
| IT-Col | 41.8494 | 13.5881 | 1560 | 32.0 | 2005 – 2014 | 2002 – 2004 | 30 | 22918 | DBF | DecTr | 10.18140/FLX/1440167 |
| IT-Sro | 43.7279 | 10.2844 | 6 | 22.5 | 2005– 2012 | 2002 – 2004 | 30 | 26961 | ENF | EveTr | 10.18140/FLX/1440176 |
| IT-Tor | 45.8444 | 7.57806 | 2160 | 2.7 | 2008 – 2014 | 2008 – 2010 | 30 | 33126 | GRA | Grass | 10.18140/FLX/1440237 |
| NL-Loo | 52.1666 | 5.74356 | 25 | 26.0 | 2005– 2014 | 2002 – 2004 | 30 | 34098 | ENF | EveTr | 10.18140/FLX/1440178 |
| US-AR1 | 36.4267 | -99.42 | 611 | 2.8 | 2012 – 2012 | 2009 – 2011 | 30 | 35024 | GRA | Grass | 10.18140/FLX/1440103 |
| US-CRT | 41.6285 | -83.3471 | 180 | 2.0 | 2014 – 2014 | 2011 – 2013 | 30 | 34895 | CRO[3] | Grass | 10.18140/FLX/1440117 |
| US-Goo | 34.2547 | -89.8735 | 87 | 4.0 | 2005 – 2007 | 2002 – 2004 | 30 | 30848 | GRA | Grass | 10.18140/FLX/1440070 |

| | | | | | | | | | | |
|---|---|---|---|---|---|---|---|---|---|---|
| US-IB2 | 41.8406 | -88.241 | 226.5 | 3.8 | 2005 – 2011 | 2004 – 2006 | 30 | 34339 | GRA | Grass | 10.18140/FLX/1440072 |
| US-Me6 | 44.3233 | -121.608 | 998 | 12.0 | 2013 – 2015 | 2010 – 2012 | 30 | 32141 | ENF | EveTr | 10.18140/FLX/1440099 |
| US-MMS | 39.3232 | -86.4131 | 275 | 46.0 | 2005 – 2014 | 2002 – 2004 | 60 | 17508 | DBF | DecTr | 10.18140/FLX/1440083 |
| US-Ne1 | 41.1651 | -96.4766 | 361 | 3.0 | 2005 – 2013 | 2002 – 2004 | 60 | 17450 | CRO[3] | Grass | 10.18140/FLX/1440084 |
| US-Ne2 | 41.1649 | -96.4701 | 362 | 3.0 | 2005 – 2013 | 2002 – 2004 | 60 | 17407 | CRO[4] | Grass | 10.18140/FLX/1440085 |
| US-Ne3 | 41.1797 | -96.4397 | 363 | 3.0 | 2005 – 2013 | 2002 – 2004 | 60 | 17351 | CRO[5] | Grass | 10.18140/FLX/1440086 |
| US-NR1 | 40.0329 | -105.546 | 3050 | 21.5 | 2005 – 2014 | 2002 – 2004 | 30 | 35023 | ENF | EveTr | 10.18140/FLX/1440087 |
| US-Oho | 41.5545 | -83.8438 | 230 | 32.0 | 2007 – 2013 | 2004 – 2006 | 30 | 31180 | DBF | DecTr | 10.18140/FLX/1440088 |
| US-Syv | 46.242 | -89.3477 | 540 | 36.0 | 2005– 2014 | 2002 – 2004 | 30 | 27276 | MF | DecTr | 10.18140/FLX/1440091 |

[Figure]

***Figure 2*** *Location of FLUXNET sites (Table 3) coded by into three land cover types: deciduous trees*
*(DecTr), evergreen trees (EveTr), and grass (Grass). Source of base map: OpenStreetMap*
*(2017).*

**3.2 MODIS LAI**

The NASA Moderate Resolution Imaging Spectroradiometer (MODIS, (Nishihama et al., 1997)) four-day composite product MCD15A3H (Myneni et al., 2015) with 500 m resolution is treated as 'observed' LAI.

Product data are available from 2002. We use the Fixed Sites Subsetting and Visualization Tool (ORNL

DAAC 2018) for FLUXNET dataset to extract the MCD15A3H time series. To obtain a daily timeseries we linearly interpolate between values, for parameter derivation (Sect. 4.1).

**3.3 Soil information**

The SoilGrids (Hengl et al. 2014) database provides soil properties (i.e., organic carbon, bulk density, Cation Exchange Capacity (CEC), pH, soil texture fractions and coarse fragments) at seven depths (0, 0.05, 0.15, 0.30, 0.60, 1.00 and 2.00 m), as well as a bedrock depth prediction, World Reference Base (WRB) and USDA soil classes. We use the SoilGrids250m (Hengl et al. 2017) version, with its ~280 raster layers, to obtain the parameters (Table 4) to derive soil moisture deficit at wilting point ($\Delta\theta_{wp}$ in mm) for soil moisture related calculations (e.g. Eqn. 18).

The difference in soil moisture between field capacity ($\theta_{FC}$ in mm) and wilting point ($\theta_{WP}$ in mm) are used with parameters defined as (Saxon and Rawls 2006):

$$\Delta\theta_{wp} = \theta_{FC} - \theta_{WP} = (W_{FC} - W_{WP})(1 - f_{CF})\, d_r \tag{23}$$

with

$$W_{FC} = k_{FC} + \left(1.283 \cdot k_{FC}^2 - 0.374 \cdot k_{FC} - 0.015\right) \tag{24}$$

using the weight fractions of sand $f_{sand}$, clay $f_{clay}$ and $f_{OM}$ organic matter

$$
\begin{aligned}
k_{FC} = {}&-0.251 \cdot f_{sand} + 0.195 \cdot f_{clay} + 0.011 \cdot f_{OM} \\
&+0.006 \cdot \left(f_{sand}\, f_{OM}\right) \\
&-0.027 \cdot \left(f_{clay}\, f_{OM}\right) \\
&+0.452 \cdot \left(f_{sand}\, f_{clay}\right) + 0.299
\end{aligned}
\tag{25}
$$

and

$$W_{WP} = k_{WP} + (0.14 \cdot k_{WP} - 0.02) \tag{26}$$

where

$$
\begin{aligned}
k_{WP} = {}&-0.024 \cdot f_{sand} + 0.487 \cdot f_{clay} + 0.006 \cdot f_{OM} \\
&+0.005 \cdot \left(f_{sand}\, f_{OM}\right) \\
&-0.013 \cdot \left(f_{clay}\, f_{OM}\right) \\
&+0.068 \cdot \left(f_{sand}\, f_{clay}\right) + 0.031
\end{aligned}
\tag{27}
$$

**Table 4** *Soil related parameters obtained from the SoilGrids (Hengl et al. 2014) database for each site (Table 3) at 250 m resolution for seven depths (Sect. 3.3). Values for each site (Table 3) are given in Sun et al. (2021).*

| Parameter | Unit | Description |
|-----------|------|-------------|
| $d_r$ | mm | Soil depth |
| $f_{CF}$ | m m$^{-3}$ | coarse fragment fraction |
| $f_{clay}$ | kg kg$^{-1}$ | clay fraction |
| $f_{sand}$ | kg kg$^{-1}$ | sand fraction |
| $f_{silt}$ | kg kg$^{-1}$ | silt fraction |
| $f_{OM}$ | kg kg$^{-1}$ | organic carbon fraction |
| $\rho_{bulk}$ | kg m$^{-3}$ | Bulk density |

**4    Parameter derivation for vegetated land covers: workflows and results**

**4.1    Leaf area index (LAI) and albedo related parameters**

LAI is a key phenology parameter in SUEWS, it moderates albedo ($\alpha$) and therefore surface radiative exchanges. LAI changes also modify both aerodynamic roughness parameters (roughness length $z_0$, zero plane displacement height $z_d$) (e.g. Kent et al. 2017) impacting aerodynamic resistance ($r_a$) and surface resistance ($r_s$). LAI directly moderates $Q_E$ and canopy interception capacity, which modifies when potential evaporation occurs and aspects of the water balance.

As the SUEWS LAI equation (Eqn. 4) makes global optimisation techniques numerically challenging to derive all the required parameters, we take a two-step approach (Fig. 3):

*1)   Approximating growing stages using an asymmetric Tukey window function:*

Tukey or cosine-tapered window (TW) is used in signal processing applications when data needs to be processed in short segments. It is defined (Bloomfield 2000):

$$TW(x,a) = \begin{cases} 1 & (0 < a < 1 \wedge a - 2x - 1 \leq 0 \wedge a + 2x - 1 \leq 0) \vee (a = 1 \wedge x = 0) \vee \left(a \leq 0 \wedge -\frac{1}{2} \leq x \leq \frac{1}{2}\right) \\ \frac{1}{2}(\cos(2\pi x) + 1) & a > 1 \wedge -\frac{1}{2} \leq x \leq \frac{1}{2} \\ \frac{1}{2}\left(\cos\left(\frac{2\pi\left(-\frac{a}{2} + x + \frac{1}{2}\right)}{a}\right) + 1\right) & 0 < a \leq 1 \wedge x \geq -\frac{1}{2} \wedge a - 2x - 1 > 0 \\ \frac{1}{2}\left(\cos\left(\frac{2\pi\left(\frac{a}{2} + x - \frac{1}{2}\right)}{a}\right) + 1\right) & 0 < a \leq 1 \wedge a + 2x - 1 > 0 \wedge x \leq \frac{1}{2} \\ 0 & |x| > \frac{1}{2} \end{cases} \tag{28}$$

where $x$ is the independent variable and $a$ the curve shape factor. We propose an asymmetric form of Tukey window (*aTW*) to approximate the intra-annual LAI dynamics:

$$aTW(x,a,b,x_0,l) = \begin{cases} \text{TW}\left(\frac{x - x_0}{l}, a\right) & x < x_0 \\ \text{TW}\left(\frac{x - x_0}{l}, b\right) & x \geq x_0 \end{cases} \tag{28}$$

where $b$ is a curve shape factor for different segments, $x_0$ the segment parameter, and $l$ the rescaling factor.

To demonstrate this we use the US-MMS site (Table 3), to fit the intra-annual LAI dynamics using an *aTW* curve (blue line, Fig. 4) to determine different phenological stages (shading, Fig. 4) and subsequently derive several related parameters:

- $LAI_{min}$: 5th percentile of LAI values before the growth and after the senescence
- $LAI_{max}$: 75th percentile of LAI values after the growth and before the senescence

▪    $T_{\text{base,GDD}}$: 99[th] percentile of air temperatures before the growth

▪    $T_{\text{base,SDD}}$: 10[th] percentile of air temperature after the growth and before the senescence.

▪    $GDD_{\text{full}}$: GDD at the end of growth based on $T_{\text{base,GDD}}$

▪    $SDD_{\text{full}}$: SDD at the end of senescence based on $T_{\text{base,SDD}}$

▪    $\alpha_{\text{min}}$ / $\alpha_{\text{max}}$: 10[th]/90[th] percentile of daily albedo values after the growth and before the senescence. A

daily albedo is calculated from 30/60 min FLUXNET observations of incoming and outgoing shortwave radiation for the period 10:00 to 14:00 (local standard time). To remove outliers a clustering method is applied (`ClusterClassify` of *Mathematica* v12.3.1 Wolfram Research 2020).

For example, at some high-latitude sites (e.g. CA-Oas) snow occurs, the winter values are based on data from shortly after senescence to shortly before growth (next spring) and the clustering approach removes the snow period albedo values.

For evergreen and deciduous trees, $\alpha_{LAI,\text{min}}$ ($\alpha_{LAI,\text{max}}$) in Eqn. 6 typically corresponds to $\alpha_{\text{min}}$ ($\alpha_{\text{max}}$), while for grassland a reverse relation holds (i.e. $\alpha_{LAI,\text{min}}$ corresponds to $\alpha_{\text{max}}$ and vice versa; see Cescatti et al.

2012 for a detailed analysis of albedo dynamics at FLUXNET sites).

*2) Deriving curve factors used in SUEWS LAI scheme:*

With the parameters derived in step 1, we can determine the curve factors $\omega_{1/2,GDD/SDD}$ by minimising the bias between MODIS observed (open triangle, Fig. 4) and SUEWS simulated (red line, Fig. 4) LAI

values.

The derived LAI related parameters for the 38 FLUXNET sites vary between different land cover groups (Table 5, Fig. 5). The derived $LAI_{\text{max/min}}$ results are consistent with those reported in the literature (Asaadi et al. 2018). For EveTr sites, the large contrast between $LAI_{\text{max}}$ and $LAI_{\text{min}}$ in the ENF sites analysed here is consistent with MODIS derived LAI for ENF having larger seasonal variability than EBF (Heiskanen et al. 2012), but some of this is caused by a known issue of particularly low winter values (Garrigues et al.

2008).

Given the global availability of MODIS LAI and reanalysis-based air temperature datasets, we suggest the LAI related parameters be derived following this workflow (Fig. 3) to set parameters for SUEWS

simulations. This can be done independent of the availability of flux tower observations.

[Figure]

**Figure 3** *Workflow 1 (Fig. 1) for deriving LAI and albedo related parameters. Related Jupyter*
*notebooks are provided in Sun et al. (2021).*

[Figure]

**Figure 4** *Intra-annual LAI dynamics at US-MMS multi-year (2002–2014) ensemble median derived*
*from MODIS observations (open triangle, Sect. 3.2) and simulated by SUEWS temperature-*
*based LAI scheme (Eqn. 4) (orange line) with Tukey window fit (blue line, Sect. 4.1) using to*
*derive the leaf-on or growth period (green shading) and senescence (yellow shading) periods.*

**Table 5** *Inter-site variability within the three vegetation classes of LAI and albedo related parameters*
*(Eqn. 4, Sect. 4.1) shown by mean and standard deviation. For individual site and by PFT type*
*parameters see Appendix C (for digital version see Sun et al. 2021). Median and interquartile*
*range (IQR) see Fig. 5.*

| | $\alpha_{min}$ [-] | $\alpha_{max}$ [-] | $LAI_{max}$ [m² m⁻²] | $LAI_{min}$ [m² m⁻²] | $GDD_{full}$ [°C day] | $SDD_{full}$ [°C day] |
|---|---|---|---|---|---|---|
| EveTr | 0.093±0.007 | 0.113±0.010 | 2.46±0.20 | 0.55±0.07 | 625±83 | -501±87. |
| DecTr | 0.102±0.004 | 0.125±0.006 | 2.9±0.4 | 0.66±0.07 | 475±137 | -273±89. |
| Grass | 0.156±0.007 | 0.185±0.009 | 2.15±0.13 | 0.35±0.06 | 484±102 | -482±74. |

| | $T_{base,GDD}$ [°C] | $T_{base,SDD}$ [°C] | $\omega_{1,GDD}$ [-] | $\omega_{1,SDD}$ [-] | $\omega_{2,GDD}$ [-] | $\omega_{2,SDD}$ [-] |
|---|---|---|---|---|---|---|
| EveTr | 2.8±1.2 | 12.5±1.2 | -1.89±0.07 | -0.0031±0.0006 | 0.00067±0.00023 | 0.96±0.31 |

| | | | | | | |
|---|---|---|---|---|---|---|
| DecTr | 5.9±1.7 | 14.7±1.1 | -1.54±0.20 | -0.0043±0.0009 | 0.0018±0.0009 | 1.55±0.35 |
| Grass | 9.9±2.1 | 16.9±1.3 | -1.93±0.04 | -0.0031±0.0007 | 0.0012±0.0006 | 1.05±0.26 |

**Figure 5** *Variation in LAI related parameters (Sect. 2.2.1) within three land cover classes (colour)*
*showing median (thick line), interquartile range (IQR, 25th and 75th percentiles, dashed lines),*
*site-specific values (thin lines).*

**4.2 Storage heat flux coefficients**

To calculate the storage heat flux $\Delta Q_S$, the required OHM coefficients (Eqn. 7), can be determined from observed net all-wave radiation and observed storage heat flux, using ordinary linear regression. As the FLUXNET sites chosen are considered to be homogeneous, we derive coefficients for each site.

Ideally the storage heat flux measurements would include each of the components than are heating and cooling down on a daily basis, such as the soil, trunk, branches, leaves and air volume in a forest (e.g. McCaughey et al. 1985, Oliphant et al. 2004). However, these measurements are unavailable in the FLUXNET2015 dataset. Hence, we calculate a residual flux $\Delta Q_{S,res} = Q^* - (Q_H + Q_E)$ by assuming energy balance closure. This has the problem of accumulating the net measurement errors in this term (Grimmond et al. 1991, Grimmond and Oke 1999).

The derived OHM coefficients (Fig. 6) can be determined by season (Anandakumar 1999; Ward et al., 2016; Sun et al., 2017). We distinguish warm ("summer") and cold ("winter") seasons using months (summer: Northern Hemisphere JJA; Southern Hemisphere: DJF; winter: DJF (JJA), respectively). For simplicity, we omit periods when LAI may be changing rapidly. If the daily mean air temperature is warmer/cooler than the annual mean of daily median temperature, then summer/winter OHM coefficients are used in the simulations.

The OHM coefficients derived for the 38 FLUXNET sites (Table 6, Fig. 7) vary between land cover types and seasons. For each land cover type, $a_1$ and $a_3$ are notably larger in winter than in summer while the seasonal difference in $a_2$ is relatively small. Thus the overall fraction of heat stored does not vary much but diurnal hysteresis effect is weaker in winter. These results are consistent with previous analytical results (Sun et al., 2017). Within each PFT, there is larger variability in $a_2$ and $a_3$ (cf. $a_1$), notably for evergreen and deciduous trees, suggesting using the most appropriate site values (e.g. medians) may improve predictions of the storage heat flux. In addition to the values derived here, we note that more detailed $\Delta Q_S$ observations are available for vegetated sites to derive such OHM coefficients (e.g.

McCaughey (1985), Oliphant et al. (2004)).

[Figure]

**Figure 6** *Workflow 2 (Fig. 1) to derive OHM coefficients. Related notebooks are provided in Sun et*
*al. (2021).*

**Table 6** *As Table 5, but for OHM related parameters (Sect. 4.2). See Fig .7 for comparison and*
*Table C2 for site specific values.*

| | $a_1$ [-] | | $a_2$ [h] | | $a_3$ [W m$^{-2}$] | |
|---|---|---|---|---|---|---|
| | summer | winter | summer | winter | summer | winter |
| EveTr | 0.294±0.027 | 0.49±0.04 | 0.140±0.023 | 0.110±0.021 | -12.7±3.0 | -4.8±2.8 |
| DecTr | 0.312±0.021 | 0.396±0.035 | 0.122±0.019 | 0.166±0.019 | -12.9±2.8 | -11.0±4.0 |
| Grass | 0.318±0.020 | 0.62±0.06 | 0.079±0.013 | 0.046±0.011 | -14.9±2.0 | -4.1±2.5 |

[Figure]

**Figure 7** *Boxplots showing variability of OHM coefficients between land covers (EveTr, DecTr and*
*Grass) and seasons (summer and winter): a) $a_1$ b) $a_2$ and c) $a_3$. Boxes (25$^{th}$ and 75$^{th}$ percentiles),*
*whiskers (5$^{th}$ and 95$^{th}$ percentiles), with median (red line) and mean (middle grey diamond, with*
*95% confidence level (top and bottom) values, and outliers (dots).*

**4.3 Surface conductance related parameters**

As the Jarvis-type formulation of stomatal/surface conductance is widely used for many land cover types, many parameter sets exist (e.g. Stewart 1988; Grimmond and Oke 1991; Ogink-Hendriks 1995; Wright et al. 1995; Bosveld and Bouten 2001; Järvi et al. 2011). Hoshika et al.'s (2018) comprehensive meta- analysis of published Jarvis-type stomatal conductance parameter values includes major woody and crop plants broadly similar to PFTs examined here.

Conventionally, the surface conductance parameters are derived by minimising the bias between the parameterised (Eqn. 14) and so-called "observed" values derived from an inverse form of Penman-

Monteith equation (Eqn. 3):

$$\frac{1}{g_s} = r_s = \left[\frac{s}{\gamma}\frac{Q_H}{Q_E} - 1\right]r_a + \frac{\rho c_p VPD}{\gamma Q_E}. \tag{29}$$

This requires the surface be dry (Section 2.2.4) which we define as being without recorded rainfall in 24

h.

However, as the optimisation may not return values because of the complexity in Eqn. 14 and the challenge of interpreting the derived parameter values, we adopt Matsumoto et al.'s (2008) approach to derive these parameters. Rather than using all the data combinations for $g_s$, the upper boundary of each forcing variable component (e.g. $g(K_\downarrow)$) is considered as the response for unconstrained conditions.

Specifically, the workflow is (Fig. 8):

1)   Calculate aerodynamic resistance $r_a$ (Eqn. 8) with roughness length $z_{0m}$ and displacement height $z_d$

derived from observed wind speed under neutral conditions (Appendix B).

2)   Calculate "observed" surface conductance $g_{s,obs}$ (Eqn. 30).

3)   Remove outliers from $g_{s,obs}$ data (step 2) iteratively (i.e. values larger than the 98[th] percentile until difference between 98[th] and 99[th] percentiles is < 1 mm s[-1]). The remainder are used for deriving parameters.

4)   Determine the upper boundaries of $g_s$ curves with each component variable. To demonstrate this we use the US-MMS site (Fig. 9). First, original data are binned (sizes: 50 W m[-2] for $K_\downarrow$, 2 °C for $T_a$, 2 kg kg[-1] for $\Delta q$ and 10 mm for $\Delta\theta$), the 95[th] percentiles of these bins are sampled 100 times (bootstrapped)

to determine anchor points (red dots, Fig. 9). Second, the parameters are fit to the $g_s$ related curves (Eqn. 14–17) using the anchor points using `NonlinearModelFit` of *Mathematica* v12.3.1 (Wolfram

Research 2008).

The derived surface conductance parameters for the 38 FLUXNET sites (Table 7 and C3) have different intra-PFT variability based on the IQR (dotted lines, Fig. 10) and demonstrates the benefit of the observations and of deriving site-values when possible. It may help in selecting appropriate PFT from other sources (e.g. NOAH values in Appendix A). The $g_{max}$ results are consistent with Hoshika et al.

(2018) in terms of inter-PFT ordering (Grass > EveTr and DecTr). The grass and crop values are comparable (Table C3) to Hoshika et al. (2018); however, our derived deciduous trees values are smaller (22 cf. 31 mm s$^{-1}$) and evergreen trees values larger (20 cf. 12 mm s$^{-1}$).

A consistent $G_{q,shape}$ (eqn 16) value (0.9 units) is obtained for all sites, implying potential for improvements to the $g(\Delta q)$ relation between $g_s$ and $\Delta q$ (e.g. other formulations $g_s(\Delta q)$ in Matsumoto et al.

(2008)) in future SUEWS development. This would be beneficial as there is a clear "plateau" observed for low $\Delta q$ (Fig. 9c). Similar issues are found in $g(\Delta\theta)$ for soil moisture related parameters. Although the parameters derived here are the 'best' fit to the $g_s$ forms in SUEWS v2020a, for each component variable multiple $g_s$ formulations exist with a range of variable fitting performance (e.g. Fig A1 in Ward et al.

2016). Here, we focus on *deriving the parameters* rather than *proposing new or more appropriate*

*formulations.*

Solar radiation related $G_K$ parameter is linked to the level of incoming solar radiation needed for evapotranspiration to occur. Given incoming solar radiation intensity varies with latitude, we see $G_K$

generally decreases polewards (Fig. 11a), suggesting geographical location could be used as a proxy for deriving $G_K$.

The air temperature related $G_T$ parameter indicates the optimal temperature for evapotranspiration to reach its probable maxima. $G_T$ appears to have a negative relation with latitude, but the two other temperature parameters ($T_L$ and $T_H$) have a very weak (none) with latitude (Fig. 11b). This suggests a universal temperature range between $T_L$ and $T_H$ might be applicable across different sites while $G_T$ should be determined on a site-by-site basis.

[Figure]

**Figure 8** *Workflow 3 (Fig. 1) for deriving surface conductance related parameters. Related notebooks are provided in Sun et al. (2021).*

[Figure]

**Figure 9** *Derived relations (blue lines) between normalised surface conductance $\widetilde{g_s}$ and (a) incoming solar radiation $K_\downarrow$,(b) air temperature $T_a$, (c) specific humidity deficit $\Delta q$, and (d) soil moisture deficit $\Delta\theta$ based on anchor data points (red dots) after bootstrapping of observations (blue dots) for an example site (US-MMS).*

**Table 7** *As Table 5, but for surface conductance related parameters (Sect. 4.3). See Fig. 10 and Appendix C.*

| | $g_{max}$ [mm s$^{-1}$] | $G_K$ [W m$^{-2}$] | $G_T$ [°C] | $T_L$ [°C] | $T_H$ [°C] | $G_{q,base}$ [-] | $G_{q,shape}$ [-] | $G_\theta$ [-] | $\Delta\theta_{WP}$ [mm] |
|---|---|---|---|---|---|---|---|---|---|
| EveTr | 20.5±1.7 | 62±5 | 10.3±1.8 | -13±4 | 41.4±2.0 | 0.391±0.033 | 0.9 | 0.033±0.009 | 511±75 |
| DecTr | 21.2±2.5 | 100±23 | 18.0±4.0 | -18±5 | 38.0±1.5 | 0.439±0.024 | 0.9 | 0.029±0.010 | 521±58 |
| Grass | 38.6±2.8 | 87±13 | 26.1±1.9 | -13±5 | 40.1±2.2 | 0.467±0.033 | 0.9 | 0.048±0.010 | 521±54 |

[Figure]

**Figure 10** *As Fig.5, but for surface conductance related parameters.*

[Figure]

**Figure 11** *Relations between absolute latitude and derived parameters: (a) solar radiation related $G_K$ by PFT (symbol) and (b) temperature related $G_T$, $T_L$ and $T_H$. Lines are derived by ordinary linear regression. See text for notation definitions.*

**5 SUEWS performance in vegetated areas**

**5.1 SUEWS configuration and evaluation**

SUEWS v2020a (Tang et al. 2021) is evaluated using its python wrapper SuPy v2021.3.18 (Sun and Grimmond 2019) with parameters (Table 1) and gap-filled 30/60 min meteorological forcing data (Table 2) based on FLUXNET2015 dataset. Simulations are conducted, with forcing data interpolated to a 5 min timestep (Ward et al. 2016), for three years (Table 3, *Evaluation period*) starting in mid winter. The first year is discarded to allow for model spin-up. The two subsequent years are evaluated when observed latent heat flux are available. In these model runs, the $z_{0m}$ and $z_{dm}$ values used are derived (Appendix B) by LAI state using observations for each LAI season for each (approximately 9 values per site; see Sun et al. (2021) for values). All other parameters (Table 1) are determined for the *parameter derivation period*

indicated in Table 3. At one site (AU-Gin), there are insufficient data for independent evaluation period from the parameter derivation period.

For the periods with 30(/60)-min $Q_E$ EC observations ($Y_{obs}$) available, the 5 min simulated values ($Y_{mod}$)

are averaged to 30(/60)-min to evaluate the j cases between 1 and the number of data points ($N$).The following metrics are used:

1)  hit rate (HR):

$$\text{HR} = \frac{\sum_{j=1}^{N} H(\delta_{Y,j} - |Y_{\text{mod},j} - Y_{\text{obs},j}|)}{N} \tag{30}$$

with Heaviside step function $H$ defined by

$$H(x) = \begin{cases} 0, & x < 0 \\ 1, & x \geq 0 \end{cases} \tag{31}$$

and the threshold $\delta_{Y,j}$ being a value dependent on evaluation variable $Y$.

In particular, $\delta_{Y,j}$ for $Q_E$ is determined as a function of net all-wave radiation $Q^*$ following Hollinger and

Richardson (2005) to be $\delta_{Y,j} = 0.1Q_j^* + 10$ (in W m$^{-2}$) based on measurement uncertainties.

2)  mean absolute error (MAE):

$$\text{MAE} = \frac{\sum_{j=1}^{N} |Y_{\text{mod},j} - Y_{\text{obs},j}|}{N} \tag{32}$$

3)  mean bias error (MBE):

$$\text{MBE} = \frac{\sum_{j=1}^{N} (Y_{\text{mod},j} - Y_{\text{obs},j})}{N} \tag{33}$$

Both the MAE and MBE would ideally be 0 (with units of parameter/variable assessed). Whereas if the

HR=1 it indicates all model predictions fall within the acceptable threshold set, while HR =0 would suggest none are within the acceptance threshold.

A performance score PS as a function for each metric ($x$; i.e. HR, MAE, |MBE|) is used to rank the sites:

$$PS = \frac{1}{N} \sum_{k=1}^{N} (w_x \tilde{x})_k \tag{34}$$

where $\tilde{x}$ is the rescaled ranking score a given metric after being ranked from poorest to best and $w_x$ is a weight associated with the temporal analysis type $k$ which varies from 1 to $N$ (number of component periods; e.g., $N = 24$ for hourly results). Equal weights (1/3) are used in the PS calculations for HR, MAE

and |MBE|.

**5.2 Impacts of model parameters on model performance**

Given the many parameters in SUEWS, first we assess the relative importance of the parameters. We assume in this analysis our derived parameters (Sect. 4) are "perfect", so we can undertake a sensitivity analysis (McCuen 1974, Beven 1975) of the Penman-Monteith equation (i.e. Eqn. 3, denoted by PM

hereafter):

$$\Delta Q_E = PM(AE + \Delta AE, r_a + \Delta r_a, r_s + \Delta r_s) - Q_E \tag{35}$$

where $AE = Q^* - \Delta Q_S$ is the available energy, incorporating parameters influences related to LAI, albedo and OHM. Similarly, multiple parameters influence the resistance terms. In Eqn. 31 the prefix $\Delta$ indicates bias terms. For simplicity, we consider the direct impacts only (i.e., secondary impacts from parameters inter-dependence are ignored). Expanding Eqn. 31 in Taylor series, gives:

$$\Delta Q_E \approx PM'(AE) \cdot \Delta AE + PM'(r_a) \cdot \Delta r_a + PM'(r_s) \cdot \Delta r_s \tag{36}$$

where $PM'(x)$, the first-order derivative of $x$, indicates the sensitivity of modelled $Q_E$. Note "$\approx$" implies the approximation of $\Delta Q_E$ as the sum of bias from the chosen parameters. To examine the influences of different parameters in model performance, we use two non-dimensional metrics derived from Eqn. 32:

1) *sensitivity coefficient* (SC) (McCuen 1974):

$$SC = PM'(x) \cdot \frac{x}{Q_E} = \frac{\partial Q_E}{\partial x} \frac{x}{Q_E} \approx \frac{\Delta Q_E/Q_E}{\Delta x/x} \tag{37}$$

gives the fractional change in $x$ causing a change in $Q_E$, indicating a relative sensitivity of PM to $x$.

For instance, SC= 0.5 suggests a 20% increase in $x$ may increase $Q_E$ by 10% (=20% × 0.5).

2) *attribution fraction* (AF): quantifies the fraction of model bias derived from a given parameter $x$:

$$AF = PM'(x) \cdot \frac{\Delta x}{\Delta Q_E} \tag{38}$$

Ideally, the sum of all AF contributors would equal one, but as we omit inter-dependence of impacts of parameters, this may not occur. However, comparing the different contributors is indicative of their relative importance in modelled $Q_E$.

For both $SC_{AE}$ and $SC_{r_a}$ they are generally a similar type of patterns (Fig. 12a-f) with seasonal and diurnal variations for the three PFT. During warm periods (~summer and ~noon time), with an increase in $\Delta AE$

and $\Delta r_a$ it is found to lead to larger positive bias in modelled $Q_E$; whereas in cooler periods (~winter and

~night time) the $\Delta AE$ and $\Delta r_a$ is found to increase the negative bias.

However, the temporal patterns in $SC_{r_s}$ differ (Fig. 13g-i) from those in $SC_{AE}$ (Fig. 13a-c) and $SC_{r_a}$ (Fig.

13d-f): the $SC_{r_s}$ values are always negative, and consistently larger in magnitude (cf. $SC_{AE}$ and $SC_{r_a}$), implying a particularly strong sensitivity of $Q_E$ to $r_s$. This is consistent with Beven (1975), who found it to dominate the modelled summertime $Q_E$ sensitivity in the PM framework.

The relative (cf. total) bias from the parameters is assessed in modelled $Q_E$ at monthly and hourly temporal scales using the median AF (Fig. 13). $AF_{r_s}$ is larger than both $AF_{AE}$ and $AF_{r_a}$; i.e., $r_s$ imposes a dominant influence in modelled $Q_E$ bias. There is more temporal variability in $AF_{r_s}$ (cf. $AF_{AE}$ and $AF_{r_a}$)

with cooler periods (morning and evening, whole winter) generally have values greater than one, indicating the bias in $r_s$ dominates modelled $Q_E$. As $AF_{r_s}$ is still generally larger than ~0.3 (except for transitional periods in summer, 8:00–9:00 in the morning, when $AF_{r_s} < {\sim}0.3$), $r_s$ remains an important control on modelled $Q_E$. These results together, indicate that it is critical to assign accurate $r_s$ to obtain accurate estimates of $Q_E$.

[Figure]

***Figure 12*** *Temporal variation in median (colour) sensitivity coefficient (SC, eq.38) of (a-c) available*
*energy (AE, sect. 5.1), (d-f) aerodynamic resistance ($r_a$) and (g-i) surface resistance ($r_s$) for (a,d,g)*
*evergreen trees (EveTr), (b,e,h) deciduous trees (DecTr), and (c,f,i) grassland and crops (Grass).*

[Figure]

***Figure 13** As Fig.12, but the attribution fraction (AF, Eq. 39). Note AF$_{rs}$ scale is logarithmic.*

**5.3 Evaluation of SUEWS simulated $Q_E$ with two different sources of $g_s$ parameters**

Given the critical importance of surface resistance to model performance in $Q_E$ (Sect. 5.2), we assess the impact two different sources of $g_s$ parameters (keeping all other site parameters the same, with the values as indicated in section 5.1): (i) site-specific values derived from FLUXNET2015 data (Sect. 4); and (ii)

PFT-specific NOAH values modified for SUEWS (Appendix A). Errors in the other derived parameters (e.g., LAI related parameters, storage heat flux coefficients via available energy) will impact both sets of results but they are assumed to be equal, allowing the impacts of using site- and PFT-specific $g_s$

parameters on SUEWS simulated $Q_E$ to be assessed. Given NOAH is extensively used in NWP systems (e.g., WRF, Skamarock and Klemp, 2008), the result also allows the applicability of NOAH-based $g_s$

parameters at FLUXNET sites to be assessed.

Analysis using two years of $Q_E$ EC flux data (after 1 year spin up) uses three metrics (Sect. 5.1). The overall median results are similar between the two sets of parameters across the 38 sites split into the three PFTs (Fig. 14, red lines). The median HR are between 0.6 and 0.7, median MAE are less than 25 W

m$^{-2}$, and median MBE are ~5 W m$^{-2}$. The Grass site, notably HR and MAE (Fig. 14a, b), performance is very similar suggesting the NOAH-based parameters could be used for these sites at annual scales as a first-order proxy.

[Figure]

**Figure 14** *Simulated Q_E using two sets of parameters (colour, FLUXNET - Table C3, NOAH – Table*
*A1 assigned based on Table 3 PFT class) at 38 sites (boxplots, as Fig. 9, subdivided into three*
*land cover classes: EveTr, DecTr and Grass) evaluated for two years with observed 30/60-min*
*fluxes using three metrics (Sect. 5.1): (a) HR, (b) MAE, and (c) MBE.*

Evaluation using three different time periods (annual, monthly and hourly), shows differences in performance between using the FLUXNET2015 and NOAH-based parameters (Fig. 15). The HR is similar for all three temporal scales (Fig. 15a–c) for the three site types (colour). Both the MAE (Fig.

15d–f) and MBE (Fig. 15g–i) indicate better model performance can be obtained using the

FLUXNET2015-based parameters (i.e. not above the 1:1 line). When using the NOAH parameters (Fig.

15h,i), some monthly MBE are ~40 W m$^{-2}$ larger at EveTr sites for 8 of 156 cases and 5/312 for DecTr sites. Similarly, at the hourly scale the NOAH MBE are on occasions ~30 W m$^{-2}$ larger (4/132 EveTr cases and 6/264 DecTr cases). However, the NOAH results have similar metrics at Grass sites. This suggests at the EveTr and DecTr sites, the NOAH-based $g_s$ parameters may on occasion be less appropriate, suggesting that the individual sites values may be better.

[Figure]

**_Figure 15_** _As Fig 14, but evaluation metrics (colour coded by land cover class) determined for three_
_temporal scales (columns): (a,d,g) whole period (n = 38 sites but different number of samples per_
_site, Table 3), (b,e,h) monthly (n = 456 = 38 sites × 12 months) and (c,f,i) hourly (n = 912 = 38 sites_
_× 24 hours) ,_

**5.4 Evaluation of SUEWS simulated $Q_E$ and key parameters at sites with contrasting model performance**

Given the results in Section 5.3, the performance of individual sites using the FLUXNET2015-derived parameters (Sect. 4) at monthly (Fig. 16) and hourly (Fig. 17) time scales are investigated.

As expected (Sect. 5.2), the HR values are consistently better at all sites during cooler (winter) than warmer (summer) seasons (Fig. 16a–c), and similarly for night rather midday time periods (Fig 17a–c).

Given the consistency in MAE and HR (Fig. 16,17d-f) patterns, the sites identified to be simulated the

'best' (blue) and 'poorest' (orange) are the same (Section 5.1).

However, using the MBE different sites are selected. For example, monthly MBE at AU-ASM stays close to zero throughout the year while at AU-Das it varies between -40 W m$^{-2}$ and 60 W m$^{-2}$ (Fig 16h). The largest intra-month MBE range for an EveTr site is 87.1 W m$^{-2}$, which occurs at FR-LBr. The equivalent range for DecTr sites is larger (96.1 W m$^{-2}$ at AU-DaS) but smaller at Grass sites (69.6 W m$^{-2}$; US-Ne3).

The intra-hourly MBE ranges are smaller than intra-monthly values: with a DecTr and a Grass site having a large range than the largest EveTr site (AU-DaS (61.9 W m$^{-2}$), US-Goo (60.0 W m$^{-2}$), FR-LBr (53.1 W

m$^{-2}$), respectively).

[Figure]

**Figure 16** *Variation in evaluation metrics (Section 5.2) based on 30/60-min Q$_E$ data by month using*
*the derived parameters based on FLUXNET2015 dataset (Tables C1–C3): (a–c) HR, (d–f) MAE,*
*(g–l) MBE by sites grouped into three PFTs (a,d,g) EveTr, (b,e,h) DecTr and (c,f,i) Grass with*
*sites of best (blue)/poorest (orange) performance highlighted while others in grey (indicated by*
*PS: see text and Eqn. 35 for details). Note Southern Hemisphere sites are offset by 6 months*
*(Sect. 5.1), so 'general' seasons are consistent across sites.*

[Figure]

Figure 17 *As Fig. 16, but for diurnal cycles using local standard time.*

To investigate $Q_E$ performance relative to key parameters (LAI, $r_a$, and $g_s$) we select the sites with contrasting results from each PFT to understand the drivers. The hourly and monthly ranked performance (Eqn. 35) are broadly consistent within each PFT type (Fig. 18). Sites with higher hourly scores generally have better monthly scores, except for IT-SRo within the EveTr cohort. It has the highest hourly PS (0.86) but is ranked fourth based on $PS_{monthly}$ (0.64), whereas the highest $PS_{momthly}$ (0.91) is for FI-Hyy which also has the second rank $PS_{hourly}$ (0.83). To select sites for further analysis we rank based on the mean of monthly and hourly PS results. The six sites chosen are the best and poorest sites for the three PFTs (i.e. extremes in Fig. 18, highlighted in Fig. 16 and 17).

[Figure]

**Figure 18** *Performance score (PS, Eqn. 35, higher value better) using FLUXNET2015 derived parameters for sites (Table 3) from three PFT: a) EveTr, b) DecTr and c) Grass.*

Comparing the contrasting site $Q_E$ performance (best cf. poorest) for the three PFTs (Fig. 19 cf. 20; 21 cf.

22; 23 cf..24), we identify the skill of capturing the annual LAI dynamics is crucial to seasonal model performance (Fig. 19–24a). At the "best" sites but BE-Lon (a Grass site, whose performance is more controlled by surface conductance $g_s$ skill and shall be discussed later) the phenology generally has the correct timing while at the poorest onsets of some stages are missed (e.g. Fig 22a). Timing appears to be more critical than magnitude, as although the LAI magnitude at AU-ASM has a large bias in year 2 (0.5

$m^2\ m^{-2}$, Fig. 21a) the phenology timing is well captured. This result for a first rank site (i.e. best performance) implies the rescaling nature of LAI in parameterisation of albedo (Eqn. 6) and surface conductance (Eqn. 14) plays an important role. This indicates the importance of assigning appropriate

LAI parameters, notably those influencing the timing (i.e. temperature thresholds $T_{base,GDD}$ and $T_{base,SDD}$ in

Eqn. 4), in SUEWS modelling $Q_E$ at vegetated sites.

As expected (Sect 5.2), SUEWS performance is critically impacted by surface conductance $g_s$ skill (Fig.

19–24b–e *cf.* j–m): sites and seasons with better model $g_s$ skill (i.e. simulations and observations closer)

show overall better performance (e.g. for Grass sites, BE-Lon vs. US-CRT, cf. Fig. 23c and 24c). $g_s$ is better modelled in warmer (JJA and SON) than cooler seasons (MAM and DJF). At night, $g_s$ is generally underestimated, by a similarly order of magnitude as that in cooler seasons (~3 mm s$^{-1}$). These results, consistent with SUEWS results for two UK urban sites (Ward et al 2016), suggest improvements are needed in the Jarvis type $g_s$ parameterisation during cooler periods. Given the method adopted here of using summertime observations, as used by many other land surface models (e.g. NOAH - Chen and

Dudhia 2001; HTESSEL - Balsamo et al. 2009), it implies that the widely adopted Jarvis type $g_s$

parameterisations and/or related parameter values (cf. Sect. 4.3) maybe biased towards vegetation canopies in warmer periods. The "cool" bias in modelled $g_s$ found here, and in earlier SUEWS work (e.g.,

Ward et al. (2016)), should be considered a more common issue beyond the SUEWS model. Given the needs in long-term climate modelling, systematic biases should be removed suggesting other land surface models that adopt the Jarvis type $g_s$ parameterisation might need revisions as well.

The aerodynamic resistance $r_a$ is modelled well at all sites (Fig. 19–24f–i), with nocturnal biases larger (e.g. underestimate of ~50 s m$^{-1}$ at AT-Neu, Fig. 23f–i). This good performance may be largely attributed to the use of local site - growth stage derived aerodynamic roughness parameters (Appendix B) rather than estimated using a morphometric model (e.g. based on canopy height). To estimate $z_{0m}$ and $z_d$ using

Eqn. 9 and 10 the $f_0$ and $f_d$ parameters can be derived using Sun et al. (2021) for different growing stages with the FLUXNET2015 data when canopy heights are available. The largest intra-PFT variability, occurs for 'Grass' sites (Fig. B1).

[Figure]

**Figure 19** *Fl-Hyy (EveTr, PS=0.92) performance (a) annual LAI. (b–e) $g_s$ (in seasonal ensemble of diurnal cycles: median values in bold lines, while interquartile ranges in shadings), f–i) $r_a$; and j–m) $Q_E$. Note the same colouring of simulations for sites with best/poorest model performance and six-month offset in annual cycles are applied for consistency with Fig. 16 and 17.*

[Figure]

**Figure 20** *As Fig. 19, but for AU-Wom (EveTr, PS=0.18).*

[Figure]

**Figure 21** *As Fig. 19, but for AU-ASM (DecTr, PS=0.96).*

[Figure]

**Figure 22** *As Fig. 19, but for AU-DaS (DecTr, PS=0.04).*

[Figure]

**Figure 23** *As Fig. 19, but for BE-Lon (Grass, PS=0.88).*

[Figure]

**Figure 24** *As Fig. 19, but for US-CRT (Grass, PS=0.04).*

**6 Concluding remarks**

In this work, we derive parameters for SUEWS for fully vegetated land covers that are commonly found in background ('rural') contexts of cities, where SUEWS has been widely used to model urban climates.

To facilitate derivation of SUEWS parameters we provide workflows in Jupyter notebooks (Sun et al.

2021) for leaf area index (LAI), albedo, Objective Hysteresis Model (OHM) coefficients, aerodynamic roughness parameters and surface conductance ($g_s$). We use these to determine parameters at 38 vegetated

FLUXNET sites in North America, Europe and Australia. Using the derived parameters, we assess performance of SUEWS in predicting latent heat flux ($Q_E$) at different temporal scales (monthly and hourly).

It is concluded that:

▪ Where observations are available, we recommend determining local parameters, as derived
parameters vary within PFT (Appendix C). The tools provided here are designed to facilitate this
(Sect. 4).

▪ Given the global availability of MODIS LAI and reanalysis-based air temperature datasets (e.g.,
ERA5), it is feasible to derive site by site LAI parameters for SUEWS (Sect. 4.1).

▪ OHM coefficients for modelling storage heat flux derived here show clear seasonality: summertime
(i.e., days warmer than annual median air temperature) $a_1$ and $a_3$ are smaller than their wintertime
counterparts while the seasonal contrast in $a_2$ is smaller, suggesting seasonally varying values should
be used for long-term (i.e. > one year) simulations.

▪ Surface conductance related parameters derived using summertime upper-boundary-based approach
(Matsumoto et al. 2008), produce parameter related to solar radiation ($G_K$) and optimal air
temperature ($G_T$) with some dependence on geographical locations, which could be used as a proxy to
derive these two parameters.

▪ SUEWS modelled $Q_E$ is particularly sensitive to surface conductance as informed by the attribution
analysis using an analytical framework by McCuen (1974) and consistent with results of Beven
(1975) that surface conductance plays a dominant role in moderating the bias in modelled $Q_E$.

▪ SUEWS configured with NOAH-based parameters has comparable prediction skill in $Q_E$ compared to
site-specific parameters when assessed by hit rate (HR) with medians being ~0.65. However, site-
specific parameters improve SUEWS performance as shown by the mean absolute error (MAE) and
mean bias error (MBE) metrics, becoming increasingly evident at finer temporal scales (monthly and
hourly).

▪ SUEWS with site-specific parameters outperforms in cooler periods (i.e., winter and night) compared
to warmer periods (i.e., summer and day): HR is consistently higher in the former periods than the
latter (0.71 cf. 0.52 in median) while MAE (cooler vs. warmer seasons median: ~12 cf. ~31 W m$^{-2}$).

▪ Correctly predicting LAI timing dynamics has a crucial influence on overall $Q_E$ model performance,
followed by surface conductance $g_s$ that is generally underestimated during cooler periods (more
pronounced at night by ~3 mm s$^{-1}$).

As the first comprehensive study of SUEWS at multiple vegetated sites, we also identify future
development and application needs:

▪ None of the simple LAI schemes in SUEWS account for hydrological impacts on LAI. Vegetation
with shallow roots (e.g. US-SRG in Arizona, US, categorised as GRA according to IGBP, Fig. D2)
are not well modelled when air temperature if the only phenology forcing variable. Hydrological
feedback should be considered in future development of the LAI scheme in SUEWS.

▪ The specific humidity deficit surface conductance parameter relation needs improvement as a plateau-
like trend is observed near the lower end (e.g., Fig. 9c).

▪ A potential source of parameters values for PFT beyond those studied here (i.e. values provided
Appendix C, Sun et al. 2021) could be NOAH-based parameters (Appendix A) but these should be
used with caution, as demonstrated (Section 5).

▪ More careful treatment of snow cover should be incorporated to enhance SUEWS capacity in high-
latitude regions

**Appendix A: NOAH-based equivalent values for surface conductance related parameters for SUEWS**

The NOAH land surface scheme (Chen et al. 1996) uses a similar Jarvis-type parameterisation of surface
conductance $g_s$ as in SUEWS (i.e. Eqn. 13) but with different formulation of $g_s$ sub-components from
SUEWS (Eqn A1-A4 cf. Eqn. 15–18). The NOAH equations, using our notation are:

• Incoming solar radiation ($K_\downarrow$):

$$g_{NOAH}(K_\downarrow) = \frac{\frac{R_{cmin}}{5000} + f}{1 + f} \tag{A1}$$

where $f = 0.55 \frac{K_\downarrow}{R_{gl}} \frac{2}{LAI}$ with $R_{gl}$ an adjustable parameter for $K_\downarrow$.

• Air specific humidity ($q$):

$$g_{NOAH}(q) = \frac{1}{1 + h_s(q_s - q)} \tag{A2}$$

where $h_s$ is the adjustable parameter for specific humidity $q$ and $q_s$ the saturation specific humidity.

• Air temperature ($T_a$):

$$g_{NOAH}(T_a) = 1 - 0.0016(T_{ref} - T_a)^2 \tag{A3}$$

where $T_{ref}$ is the adjustable parameter for air temperature $T_a$.

• Soil moisture ($\theta_{soil}$):

$$g_{NOAH}(\theta_{soil}) = \frac{\theta_{soil} - \theta_{WP}}{\theta_{FC} - \theta_{WP}} \tag{A4}$$

where $\theta_{FC}$ and $\theta_{WP}$ are field capacity and wilting point (see Table A2 for values of different soil types).

To use the NOAH parameters (as given in Table 1 and 2 of Chen and Duhdia 2001), we convert the
NOAH parameters (Eqn. A1- A4) to the SUEWS required parameters (i.e. for Eqn. 15-18). The resulting
SUEWS parameters (Table A1) are used (Sect. 5) to produce results in Fig. 14 and 15 denoted by NOAH.

**Table A1** *NOAH-derived (data: based on Table 1 and 2 of Chen and Duhdia 2001) surface conductance related parameters for SUEWS*

| | $g_{max}$ [mm s$^{-1}$] | $G_K$ [W m$^{-2}$] | $G_T$ [°C] | $T_L$ [°C] | $T_H$ [°C] | $G_{q,base}$ [-] | $G_{q,shape}$ [-] | $G_\theta$ [-] |
|---|---|---|---|---|---|---|---|---|
| EBF | 10.0 | 67 | 24.85 | -0.15 | 49.85 | 0.361 | 0.932 | 0.002 |
| DBF | 10.0 | 67 | 24.85 | -0.15 | 49.85 | 0.348 | 0.924 | 0.002 |
| MF | 10.0 | 66 | 24.85 | -0.15 | 49.85 | 0.352 | 0.927 | 0.002 |
| ENF | 6.7 | 65 | 24.85 | -0.15 | 49.85 | 0.361 | 0.932 | 0.002 |
| GRA | 25.0 | 336 | 24.85 | -0.15 | 49.85 | 0.384 | 0.945 | 0.002 |
| CSH | 3.3 | 291 | 24.85 | -0.15 | 49.85 | 0.372 | 0.938 | 0.002 |
| OSH | 2.5 | 275 | 24.85 | -0.15 | 49.85 | 0.372 | 0.938 | 0.002 |
| SAV | 6.7 | 316 | 24.85 | -0.15 | 49.85 | 0.372 | 0.938 | 0.002 |
| CRO | 25.0 | 336 | 24.85 | -0.15 | 49.85 | 0.384 | 0.945 | 0.002 |
| WET | 6.7 | 316 | 24.85 | -0.15 | 49.85 | 0.338 | 0.919 | 0.002 |
| WSA | 6.7 | 316 | 24.85 | -0.15 | 49.85 | 0.372 | 0.938 | 0.002 |

**Table A2** *Soil field capacity ($\theta_{FC}$) and wilting point ($\theta_{WP}$) used in NOAH (data: based on Table 2 of Chen and Duhdia 2001).*

| | $\theta_{FC}$ [m$^3$ m$^{-3}$] | $\theta_{WP}$ [m$^3$ m$^{-3}$] |
|---|---|---|
| Sand | 0.236 | 0.01 |
| Loamy sand | 0.283 | 0.028 |
| Sandy loam | 0.312 | 0.047 |
| Silt loam | 0.36 | 0.084 |
| Silt | 0.36 | 0.084 |
| Loam | 0.329 | 0.066 |
| Sandy clay loam | 0.314 | 0.067 |
| Silty clay loam | 0.387 | 0.12 |
| Clay loam | 0.382 | 0.103 |
| Sandy clay | 0.338 | 0.1 |
| Silty clay | 0.404 | 0.126 |
| Clay | 0.412 | 0.138 |

Except for the $g_s$ function for air temperature – SUEWS and NOAH adopt the effectively same formulation – other $g_s$ functions may produce different results even using the converted parameter values (Table A1). In particular, the shortwave radiation and specific humidity (Fig. A1), the SUEWS values (blue) are higher for all PFT types than NOAH (red). The role of soil type (Table A2) on the soil moisture deficit function (Fig. A2) results in larger differences at dry and mid-wet extremes.

[Figure]

**Figure A1** *NOAH (red) and SUEWS (blue) surface conductance functions for incoming solar radiation (K↓) and specific humidity deficit (Δq) for different IGBP PFTs.*

[Figure]

**Figure A2.** As Fig A1 but for soil moisture deficit (Δθ) for different soil types with an assumed soil
depth of 2000 mm. Soil hydraulic properties (field capacity θ$_{FC}$ and wilting point θ$_{WP}$) are provided
in Table A2.

**Appendix B: Derivation of roughness length and zero-plane displacement height for momentum**

The aerodynamic roughness parameters for momentum (roughness length $z_{0m}$ and zero-plane displacement height $z_d$) are derived using observed $u_*$ and $u$ under neutral conditions (i.e. $\left|\frac{z_m - z_d}{L}\right| < 0.01$

with an initial estimate of $z_d = 0.7 H_c$) of different vegetation stages based on LAI (see Sect. 4.1 for classification details) by the least-square method for the following relation (Monin and Obukhov, 1954):

$$u = \frac{u_*}{\kappa}\ln\left(\frac{z_m - z_d}{z_{0m}}\right)$$

(B1)

where $\kappa$ is the von Kármán constant (0.4 is used here). In particular, for sites with varying canopy height

$H_c$, $z_{0m}$ and $z_d$ are derived for each of the periods when $H_c$ stayed unchanged and more than 20

observational pairs of $u_*$ and $u$ are available.

Using the derived $z_{0m}$ and $z_d$, $f_0$ and $f_d$ parameters can be obtained (Eqn. 9 and 10). These is considerable intra-PFT variability of both $f_0$ and $f_d$ (Fig. B1). There are also intra-site variations associated with varying

$H_c$. Given the large variability in both $f_0$ and $f_d$, the rule-of-thumb approach would incur large bias in estimated aerodynamic and surface resistances and subsequently the modelled $Q_E$. To reduce such bias, in the evaluation of the other sub-models and parameter determinations in this paper, we use the derive $z_{0m}$

and $z_d$ determined for each vegetation stage and site.

[Figure]

***Figure B1***. *Relations between canopy height ($H_c$) and a) roughness length for momentum ($z_{0m}$, Eqn.*
*B2) and b) displacement height ($z_d$, Eqn. B3) for different vegetation stages based on LAI (see*
*Sect. 4.1 for classification details)*

Modelled wind speed under neutral conditions matches well with observations at 38 study sites with

MAE < 0.3 m s[-1] and MBE close to zero (Fig. B2). Of the three SUEWS PFTs, 'Grass' sites have the poorer performance. This is probably because this PFT includes crops which will change frequently because of crop rotations: cereal, potato, sugar beet at BE-Lon (Moureaux et al., 2006); winter barley, rapeseed, winter wheat, maize and spring barley at DE-Kli (Prescher et al., 2010), maize and soybean at

US-Ne2 and US-Ne3).

[Figure]

**Figure B2** *MAE (blue) and MBE (orange) for modelled wind speed under neutral conditions for three*
*SUEWS PFTs: a) EveTr, b) DecTr and c) Grass.*

**Appendix C: SUEWS parameters derived at selected FLUXNET sites**

**Table C1** *LAI and albedo related parameters at 38 sites (Table 3 gives site information) derived*
*using FLUXNET2015 dataset (Sect. 4.1).*

| | $\alpha_{min}$ [-] | $\alpha_{max}$ [-] | $LAI_{max}$ [m² m⁻²] | $LAI_{min}$ [m² m⁻²] | $GDD_{full}$ [°C day] | $SDD_{full}$ [°C day] | $T_{base,GDD}$ [°C] | $T_{base,SDD}$ [°C] | $\omega_{1,GDD}$ [-] | $\omega_{1,SDD}$ [-] | $\omega_{2,GDD}$ [-] | $\omega_{2,SDD}$ [-] |
|---|---|---|---|---|---|---|---|---|---|---|---|---|
| **AT-Neu** | 0.23 | 0.19 | 2.24 | 0.18 | 1200.58 | -1015.40 | -0.20 | 14.02 | -2.00 | -3.02E-03 | 5.35E-05 | 0.09 |
| **AU-ASM** | 0.12 | 0.09 | 0.30 | 0.20 | 93.17 | -31.48 | 13.90 | 16.42 | -1.32 | -2.21E-03 | 1.43E-05 | 0.97 |
| **AU-DaS** | 0.14 | 0.12 | 1.86 | 0.97 | 350.98 | -78.63 | 25.74 | 24.90 | -2.00 | -5.45E-03 | 5.29E-05 | 1.37 |
| **AU-Gin** | 0.13 | 0.11 | 1.09 | 0.60 | 239.14 | -96.09 | 12.73 | 17.21 | -2.00 | -1.16E-03 | 7.87E-05 | 2.95 |
| **AU-Wom** | 0.10 | 0.09 | 4.51 | 1.19 | 404.14 | -43.42 | 5.50 | 7.86 | -2.00 | -5.17E-03 | 2.10E-03 | 1.17 |
| **BE-Lon** | 0.17 | 0.14 | 1.66 | 0.15 | 264.31 | -467.14 | 8.63 | 14.53 | -2.00 | -2.27E-03 | 4.64E-04 | 3.33 |
| **CA-Gro** | 0.09 | 0.08 | 2.97 | 0.65 | 1000.69 | -300.71 | -0.62 | 13.37 | -2.00 | -5.50E-04 | 2.88E-04 | 1.72 |
| **CA-Oas** | 0.12 | 0.11 | 3.87 | 0.56 | 252.15 | -193.36 | 8.69 | 13.81 | -1.71 | -3.22E-03 | 3.81E-03 | 1.65 |
| **CA-Qfo** | 0.11 | 0.06 | 1.82 | 0.20 | 932.53 | -675.56 | -4.41 | 12.84 | -1.26 | -1.05E-03 | 6.79E-05 | 0.57 |
| **CA-SF2** | 0.12 | 0.11 | 2.74 | 0.50 | 781.58 | -382.78 | 1.43 | 12.35 | -2.00 | -9.12E-04 | 2.08E-04 | 3.03 |
| **CA-SF3** | 0.10 | 0.09 | 2.22 | 0.45 | 724.36 | -598.25 | 1.40 | 14.45 | -2.00 | -7.55E-03 | 1.24E-04 | 0.10 |
| **CA-TP4** | 0.09 | 0.08 | 2.57 | 0.43 | 1042.85 | -599.76 | 4.54 | 18.51 | -2.00 | -3.77E-03 | 1.32E-04 | 0.14 |
| **CH-Cha** | 0.21 | 0.18 | 2.39 | 0.52 | 161.91 | -369.59 | 2.69 | 9.41 | -2.00 | -6.79E-03 | 2.45E-03 | 1.04 |
| **CH-Dav** | 0.07 | 0.06 | 2.04 | 0.20 | 645.94 | -380.00 | 1.95 | 9.97 | -1.36 | -3.69E-04 | 1.14E-04 | 2.83 |
| **CH-Oe1** | 0.22 | 0.18 | 1.98 | 0.31 | 399.90 | -486.85 | 2.78 | 12.45 | -2.00 | -1.39E-03 | 4.58E-04 | 0.29 |
| **DE-Gri** | 0.21 | 0.19 | 2.52 | 0.69 | 369.12 | -383.65 | 3.10 | 12.72 | -2.00 | -7.57E-03 | 6.59E-04 | 0.05 |
| **DE-Hai** | 0.13 | 0.09 | 3.82 | 0.98 | 101.43 | -49.58 | 7.46 | 9.89 | -0.09 | -1.00E-02 | 3.49E-03 | 4.00 |
| **DE-Kli** | 0.19 | 0.14 | 2.24 | 0.30 | 371.62 | -129.27 | 2.81 | 9.29 | -2.00 | -7.34E-03 | 4.20E-04 | 1.46 |
| **DE-Lkb** | 0.22 | 0.14 | 2.68 | 0.51 | 967.50 | -293.93 | -1.32 | 7.59 | -2.00 | -3.45E-03 | 1.45E-04 | 0.52 |
| **DE-Obe** | 0.07 | 0.06 | 2.84 | 0.59 | 349.45 | -306.51 | 1.77 | 7.74 | -2.00 | -7.57E-03 | 1.05E-03 | 1.27 |

| | | | | | | | | | | | | |
|---|---|---|---|---|---|---|---|---|---|---|---|---|
| FI-Hyy | 0.10 | 0.09 | 2.73 | 0.43 | 859.30 | -512.51 | 0.08 | 14.97 | -2.00 | -3.98E-03 | 3.35E-04 | 0.10 |
| FR-LBr | 0.11 | 0.10 | 2.35 | 0.62 | 322.30 | -263.61 | 11.58 | 17.10 | -2.00 | -2.58E-03 | 7.28E-04 | 2.51 |
| IT-Col | 0.13 | 0.11 | 3.12 | 0.54 | 390.41 | -424.85 | 3.60 | 11.45 | -2.00 | -7.40E-04 | 1.22E-03 | 0.85 |
| IT-SRo | 0.09 | 0.07 | 2.73 | 0.92 | 481.73 | -564.19 | 8.39 | 18.06 | -2.00 | -2.68E-03 | 6.75E-04 | 0.07 |
| IT-Tor | 0.23 | 0.18 | 2.08 | 0.20 | 625.92 | -467.45 | 0.70 | 9.16 | -2.00 | -9.33E-04 | 1.58E-04 | 1.45 |
| NL-Loo | 0.10 | 0.09 | 2.01 | 0.49 | 676.48 | -937.96 | 6.27 | 16.31 | -2.00 | -6.60E-03 | 9.67E-05 | 0.02 |
| US-AR1 | 0.15 | 0.14 | 0.96 | 0.27 | 387.82 | -929.25 | 10.88 | 22.53 | -2.00 | -2.31E-03 | 3.71E-05 | 0.06 |
| US-CRT | 0.12 | 0.11 | 2.53 | 0.38 | 23.74 | -264.69 | 25.24 | 17.99 | -2.00 | -1.21E-03 | 4.49E-03 | 1.25 |
| US-Goo | 0.20 | 0.16 | 3.13 | 0.73 | 131.20 | -276.51 | 16.85 | 19.13 | -2.00 | -5.99E-04 | 7.11E-03 | 2.09 |
| US-IB2 | 0.15 | 0.13 | 1.94 | 0.21 | 1403.13 | -940.31 | 1.12 | 19.90 | -1.74 | -7.04E-03 | 4.08E-05 | 0.06 |
| US-Me6 | 0.15 | 0.11 | 1.46 | 0.61 | 22.89 | -309.48 | 6.70 | 4.74 | -2.00 | -5.80E-04 | 2.91E-03 | 0.15 |
| US-MMS | 0.12 | 0.11 | 5.00 | 1.01 | 68.61 | -50.83 | 13.97 | 15.79 | -0.45 | -7.11E-03 | 8.30E-03 | 2.22 |
| US-Ne1 | 0.16 | 0.14 | 2.13 | 0.34 | 475.09 | -321.98 | 14.84 | 22.34 | -1.80 | -7.23E-04 | 2.07E-04 | 1.62 |
| US-Ne2 | 0.16 | 0.13 | 2.08 | 0.30 | 469.13 | -380.78 | 14.65 | 22.42 | -1.83 | -2.03E-03 | 2.13E-04 | 0.41 |
| US-Ne3 | 0.17 | 0.16 | 2.18 | 0.33 | 491.72 | -317.46 | 14.89 | 22.64 | -1.60 | -5.25E-04 | 2.15E-04 | 1.47 |
| US-NR1 | 0.14 | 0.14 | 1.53 | 0.51 | 640.96 | -1239.29 | -0.32 | 11.62 | -2.00 | -1.20E-03 | 9.67E-05 | 0.04 |
| US-Oho | 0.14 | 0.12 | 2.78 | 0.45 | 1540.19 | -981.83 | 1.50 | 21.71 | -1.65 | -5.24E-03 | 9.67E-05 | 0.05 |
| US-Syv | 0.16 | 0.10 | 4.81 | 0.66 | 461.96 | -202.37 | 4.30 | 15.08 | -1.67 | -3.25E-03 | 2.33E-03 | 1.12 |

**Table C2** *As Table C1 but for OHM related parameters (Sect. 4.2)*

| | $a_1$ [-] | | $a_2$ [h] | | $a_3$ [W m$^{-2}$] | |
|---|---|---|---|---|---|---|
| | summer | winter | summer | winter | summer | winter |
| AT-Neu | 0.37 | 0.88 | 0.05 | 0.02 | -12.12 | 11.15 |
| AU-ASM | 0.39 | 0.39 | 0.15 | 0.13 | -26.44 | -35.45 |
| AU-DaS | 0.31 | 0.30 | 0.05 | 0.06 | -29.17 | -21.82 |
| AU-Gin | 0.40 | 0.25 | 0.17 | 0.21 | -11.35 | -25.75 |
| AU-Wom | 0.29 | 0.29 | 0.10 | 0.08 | 3.54 | -18.13 |
| BE-Lon | 0.32 | 0.82 | 0.18 | 0.04 | -3.58 | 8.00 |
| CA-Gro | 0.31 | 0.51 | 0.15 | 0.19 | -9.62 | -8.94 |
| CA-Oas | 0.27 | 0.45 | 0.13 | 0.25 | -6.84 | -0.58 |
| CA-Qfo | 0.30 | 0.43 | 0.22 | 0.25 | -10.70 | -1.50 |
| CA-SF2 | 0.16 | 0.46 | 0.15 | 0.09 | -17.05 | -3.39 |
| CA-SF3 | 0.25 | 0.60 | 0.20 | 0.09 | -22.22 | -8.67 |
| CA-TP4 | 0.28 | 0.42 | 0.17 | 0.13 | -7.94 | 1.07 |
| CH-Cha | 0.25 | 0.67 | 0.05 | 0.00 | -29.09 | -8.37 |
| CH-Dav | 0.56 | 0.65 | 0.13 | 0.11 | -29.23 | -12.08 |
| CH-0e1 | 0.34 | 0.67 | 0.08 | 0.12 | -22.53 | -6.21 |
| DE-Gr1 | 0.44 | 0.76 | 0.04 | 0.02 | -6.13 | 5.14 |
| DE-Hai | 0.15 | 0.21 | -0.01 | 0.15 | -1.31 | 9.81 |
| DE-Kli | 0.45 | 0.77 | 0.12 | 0.07 | -11.62 | 4.91 |
| DE-Lkb | 0.23 | 0.84 | 0.09 | -0.04 | -31.59 | -18.49 |
| DE-0be | 0.33 | 0.53 | -0.06 | 0.02 | -10.33 | -8.95 |
| Fl-Hyy | 0.23 | 0.68 | 0.17 | 0.09 | -8.75 | 7.89 |
| FR-LBr | 0.27 | 0.43 | 0.19 | 0.17 | -17.30 | -8.04 |
| IT-C01 | 0.36 | 0.43 | 0.14 | 0.22 | -4.56 | -10.00 |
| IT-SRo | 0.31 | 0.51 | 0.27 | 0.15 | -5.40 | 13.57 |
| IT-Tor | 0.23 | 0.94 | 0.09 | 0.05 | -18.92 | 4.31 |
| NL-Loo | 0.24 | 0.44 | 0.14 | 0.20 | 1.06 | 2.90 |
| US-ARI | 0.18 | 0.17 | 0.16 | 0.11 | -26.72 | -16.50 |
| US-CRT | 0.32 | 0.63 | 0.11 | 0.06 | -11.14 | -6.23 |
| US-Goo | 0.38 | 0.41 | 0.04 | 0.07 | -16.06 | -20.52 |
| US-lB2 | 0.29 | 0.37 | 0.00 | 0.03 | -4.97 | -6.43 |
| US-Me6 | 0.35 | 0.51 | 0.21 | 0.12 | -24.00 | -0.93 |
| US-MMS | 0.34 | 0.45 | 0.19 | 0.25 | -7.46 | -7.21 |
| US-Ne1 | 0.30 | 0.56 | 0.09 | 0.03 | -13.72 | -9.05 |
| US-Ne2 | 0.29 | 0.51 | 0.06 | 0.04 | -14.54 | -9.19 |
| US-Ne3 | 0.29 | 0.55 | 0.04 | -0.02 | -17.87 | -8.34 |
| US-NRI | 0.27 | 0.22 | 0.05 | 0.06 | -7.97 | -15.95 |
| US-Oho | 0.32 | 0.44 | 0.10 | 0.16 | -13.31 | -12.73 |
| US-syv | 0.33 | 0.33 | 0.06 | 0.13 | -9.21 | -0.93 |

**Table C3** *As Table C1, but for surface conductance related parameters (Sect. 4.3)*

| | $g_{max}$ [mm s$^{-1}$] | $G_K$ [W m$^{-2}$] | $G_T$ [°C] | $T_L$ [°C] | $T_H$ [°C] | $G_{q,base}$ [-] | $G_{q,shape}$ [-] | $G_\theta$ [-] | $\Delta\theta_{WP}$ [mm] |
|---|---|---|---|---|---|---|---|---|---|

| | | | | | | | | | |
|---|---|---|---|---|---|---|---|---|---|
| **AT-Neu** | 39.6193 | 50.00 | 23.67 | -15.74 | 31.59 | 0.50 | 0.90 | 0.04 | 687.75 |
| **AU-ASM** | 15.73 | 288.38 | 26.77 | 8.00 | 34.39 | 0.40 | 0.90 | 0.01 | 409.50 |
| **AU-DaS** | 19.92 | 83.22 | 37.78 | -30.00 | 38.00 | 0.50 | 0.90 | 0.02 | 414.75 |
| **AU-Gin** | 7.44 | 94.79 | 13.23 | -14.59 | 41.54 | 0.41 | 0.90 | 0.06 | 446.25 |
| **AU-Wom** | 30.63 | 70.26 | 14.73 | 2.00 | 41.91 | 0.36 | 0.90 | 0.10 | 535.50 |
| **BE-Lon** | 25.64 | 74.21 | 23.22 | 5.31 | 36.41 | 0.50 | 0.90 | 0.01 | 430.50 |
| **CA-Gro** | 22.96 | 50.00 | 5.33 | -30.00 | 36.00 | 0.40 | 0.90 | 0.01 | 876.75 |
| **CA-Oas** | 26.76 | 50.00 | 14.53 | -30.00 | 30.89 | 0.50 | 0.90 | 0.01 | 603.75 |
| **CA-Qfo** | 15.56 | 50.00 | 11.62 | -0.93 | 46.81 | 0.22 | 0.90 | 0.01 | 404.25 |
| **CA-SF2** | 23.59 | 91.88 | 12.21 | -29.99 | 33.65 | 0.50 | 0.90 | 0.03 | 525.00 |
| **CA-SF3** | 23.95 | 50.00 | -0.01 | -30.00 | 38.43 | 0.36 | 0.90 | 0.07 | 514.50 |
| **CA-TP4** | 23.10 | 50.00 | 19.12 | -9.57 | 50.00 | 0.50 | 0.90 | 0.03 | 446.25 |
| **CH-Cha** | 46.30 | 126.77 | 35.86 | -30.00 | 50.00 | 0.50 | 0.90 | 0.10 | 173.25 |
| **CH-Dav** | 18.14 | 50.00 | 4.55 | -15.93 | 49.96 | 0.50 | 0.90 | 0.07 | 141.75 |
| **CH-Oe1** | 41.74 | 77.78 | 16.52 | -0.99 | 50.00 | 0.50 | 0.90 | 0.01 | 556.50 |
| **DE-Gri** | 25.43 | 142.40 | 20.71 | -23.20 | 36.09 | 0.50 | 0.90 | 0.00 | 750.75 |
| **DE-Hai** | 22.45 | 69.16 | 5.61 | -29.82 | 34.65 | 0.27 | 0.90 | 0.01 | 514.50 |
| **DE-Kli** | 24.50 | 51.37 | 19.63 | -30.00 | 26.26 | 0.50 | 0.90 | 0.01 | 824.25 |
| **DE-Lkb** | 32.34 | 85.39 | -0.12 | -30.00 | 50.00 | 0.50 | 0.90 | 0.02 | 278.25 |
| **DE-Obe** | 14.76 | 50.01 | 4.44 | -28.20 | 34.78 | 0.41 | 0.90 | 0.03 | 283.50 |
| **Fl-Hyy** | 17.23 | 64.75 | 18.94 | 4.00 | 34.44 | 0.40 | 0.90 | 0.01 | 708.75 |
| **FR-LBr** | 24.48 | 50.00 | 14.03 | -8.96 | 33.12 | 0.33 | 0.90 | 0.01 | 1065.75 |
| **IT-Col** | 14.66 | 50.00 | 13.96 | -12.26 | 50.00 | 0.50 | 0.90 | 0.02 | 425.25 |
| **IT-SRo** | 20.57 | 50.00 | 7.66 | -30.00 | 49.99 | 0.47 | 0.90 | 0.01 | 971.25 |
| **IT-Tor** | 42.23 | 73.54 | 23.20 | -15.96 | 49.99 | 0.50 | 0.90 | 0.09 | 294.00 |
| **NL-Loo** | 16.53 | 50.00 | 17.70 | 6.00 | 37.06 | 0.35 | 0.90 | 0.10 | 252.00 |
| **US-AR1** | 27.51 | 181.15 | 16.31 | 6.00 | 34.00 | 0.03 | 0.90 | 0.01 | 315.00 |
| **US-CRT** | 27.43 | 50.00 | 30.54 | -14.00 | 42.02 | 0.50 | 0.90 | 0.10 | 824.25 |
| **US-Goo** | 41.46 | 172.08 | 24.45 | -10.87 | 50.00 | 0.50 | 0.90 | 0.03 | 404.25 |
| **US-IB2** | 48.78 | 50.00 | 29.19 | -7.90 | 36.00 | 0.50 | 0.90 | 0.06 | 483.00 |
| **US-Me6** | 13.70 | 92.88 | 3.44 | -10.00 | 42.71 | 0.11 | 0.90 | 0.01 | 525.00 |
| **US-MMS** | 22.54 | 182.23 | 27.06 | -0.24 | 36.62 | 0.50 | 0.90 | 0.01 | 845.25 |
| **US-Ne1** | 52.14 | 50.00 | 29.84 | -30.00 | 34.43 | 0.50 | 0.90 | 0.04 | 519.75 |
| **US-Ne2** | 53.45 | 59.97 | 32.04 | -30.00 | 34.02 | 0.50 | 0.90 | 0.10 | 383.25 |
| **US-Ne3** | 44.55 | 53.41 | 30.84 | -30.00 | 50.00 | 0.50 | 0.90 | 0.07 | 645.75 |
| **US-NRI** | 15.91 | 52.94 | 5.51 | -11.90 | 33.80 | 0.43 | 0.90 | 0.02 | 509.25 |
| **US-Oho** | 40.01 | 56.33 | 31.08 | -29.99 | 36.53 | 0.50 | 0.90 | 0.10 | 220.50 |
| **US-Syv** | 16.85 | 130.28 | 18.77 | -4.00 | 40.65 | 0.50 | 0.90 | 0.01 | 456.75 |

**Appendix D: Typical Intra-annual LAI dynamics under contrasting meteorological controls**

Given sufficient $CO_2$, vegetation phenology indicated by LAI dynamics, is predominantly controlled by input energy and water (Fang et al. 2019). Two variables that capture this seasonal variability are air temperature and precipitation. Different intra-annual LAI dynamics are evident between sites, with contrasting meteorological controls:

a) Thermally dominant (US-MMS; Fig. D1): Intra-annual cumulative precipitation at US-MMS steadily increases throughout the year (Fig. D1a), implying a fairly even distribution of water supply; while air temperature gradually increases from the mid winter (beginning of a year), peaks in August and decrease with the start of the next winter (Fig. D1b). The LAI pattern at US-MMS responds to the air temperature, notably the growing degrees days (GDD) and then autumn senescence (SDD). This inverse "U"-shape typifies sites with thermally-dominant LAI dynamics. These types of sites are well parameterised by the current LAI scheme in SUEWS (Sect. 2.2.1).

[Figure]

**Figure D1** *Median (line) and interquartile range (shading) daily variation at US-MMS a DBF site*
*during the period 2002–2015 of (a) precipitation (cumulative), (b) air temperature (7-day moving*
*average) and (c) LAI (7-day moving average).*

b) Rainfall and thermal controls (US-SRG; Fig. D2): at this grassland site in Arizona, US the intra-annual precipitation has clear dry and wet seasons. The monsoon wet season after the peak air temperature in

July through September (Fig. D2a), which has warmest air temperatures, Unlike US-MMS (Fig. D2b), the peak air temperature is more distinct (for a shorter period). A clear relation between the onset of rainfall and LAI enhancement can be seen but the GDD and SDD relation differs from US-MMS and it not captured by the current models in SUEWS. The rainfall and enhanced LAI and $Q_E$ are associated with cooler daily air temperatures. Sites where the LAI dynamics are not captured are not explored further in this paper.

[Figure]

***Figure D2*** *As Fig. D1, but for US-SRG (GRA according to IGBP; time span: 2008–2015; DOI:*
*10.18140/FLX/1440114)*

**Code and data availability**

All source codes, input and output data are archived at Zenodo in Sun et al. (2021).

**Author contribution**

HO, TS and SG contributed to data preparation, parameter derivation, running simulations and writing the paper. HO led the initial and TS the revised versions of this work. All other authors (DB, AB, JC, ZD, HI, and JM) provided data for analysis of this work at different stages (some not used in this paper), interpreted the results, and reviewed the manuscript.

**Competing interest**

The authors declare that they have no conflict of interest.

**Acknowledgments**

This work is funded by NERC-COSMA project (NE/S005889/1), Newton Fund Met Office CSSP-China (SG), NERC Independent Research Fellowship (NE/P018637/1), and National Natural Science

Foundation of China (41875013; Zhiqiu Gao and Zexia Duan).

---

## Author Response (AR2)

We thank the editor and reviewer for the re-evaluation of our work and appreciate the helpful suggestions.

**Editor**

**Specific comments:**

1. Line 31 Abstract: Could you make it clear what is being compared between urban and rural areas in the first sentence of the abstract. Its obvious later but this is the first line of the abstract.

Revised as follows:

To compare the impact of surface-atmosphere exchanges from rural and urban areas, fully vegetated areas (e.g. deciduous trees, evergreen trees and grass) commonly found adjacent to cities need to be modelled.

2. Line 78: sentence is very long. Consider splitting into two.

Revised as follows:

Parameters for different types of urban areas (e.g. land cover differences) and regions (e.g. high/mid-latitude) have been derived. However, both limited observations and lack of a standard workflow for deriving parameters remains a constraint. This is evident in the availability of conductance and storage heat flux related parameters (e.g. Järvi et al. 2011, 2014; Ward et al. 2016).

3. Line 90: I assume you could specify that you mean model development or evaluations?

Here we mean FLUXNET can be used for both development and evaluation. Revised as:

FLUXNET (Baldocchi et al., 2001) is a global network of sites that monitor surface-atmosphere exchanges (e.g. carbon, water, and energy using the eddy covariance technique for the turbulent fluxes). These data provide unprecedented possibilities to advance process-based land surface modelling, through both development (e.g. Stöckli et al. 2008) and evaluation (e.g. Zhang et al. 2017).

4. Line 260: Sorry I didn't complete understand criteria 3 "model capacity" – does this mean there is something about the sites that means they are not suitable to be modelled?

Yes, at sites LAI dynamics could not be captured by SUEWS with the current parameterisation. Revised as:

3) model capacity (38/56): the SUEWS v2020a LAI scheme is forced with only air temperature and not other variables (e.g., rainfall) which may strongly influence phenology at some sites (Appendix D). Hence, these sites are excluded.

5. Line 850: Could you add the Zenodo links for the source code model data to this paper for completeness, obviously maintaining the citation to Sun et al. (2021).

Added and revised the related text as follows:

All source codes, input and output data are archived at Sun et al. (2021) which can be accessed at: https://zenodo.org/record/5519919.

**Reviewer**

**General Comments:**

1. Fig. 5 and 10: The lines are not appropriate. They indicate some kind of course between the different parameters. To use markers in this case would be more appropriate.

In Fig. 5 and 10, parallel coordinates (https://en.wikipedia.org/wiki/Parallel\_coordinates, accessed on 26 Feb 2022) are used to link high dimensional datasets. The polylines link labelled features with different scales, removing lines make these linkages much less evident. Caption is also revised as:

Variation in LAI related parameters (12 labelled vertical lines, Sect. 2.2.1) within three land cover classes (colour) showing median (thick line), interquartile range (IQR, 25th and 75th percentiles, dashed lines), site-specific values (thin lines).

 Figure captions: Various figure captions refer to other figure captions. This decreases the readability of the manuscript. Sometimes they are even misleading, i.e. in Fig. 15 it is referred to Fig. 14, though figure 15 shows a scatterplot while Fig. 14 shows a boxplot. This is not the same and should be revised.

Thanks for the suggestion.

The caption of Fig. 15 has now been revised as:

Figure 15: Relation between NOAH and FLUXNET of (rows) three evaluation metrics for (columns) three temporal scales (all, n = 38 sites but different number of samples per site, Table 3; monthly, n = 456 = 38 sites × 12 months; and hourly, n = 912 = 38 sites × 24 hours). Data points are colour coded by land cover class.